# *Beyond In-Context Learning:* Enhancing Long-form Generation of Large Language Models via Task-Inherent Attribute Guidelines

## Abstract

In-context learning (ICL) is an important yet not fully understood ability of pre-trained large language models (LLMs). It can greatly enhance task performance using a few examples, termed *demonstrations*, without fine-tuning. Although effective in question answering, ICL often underperforms in long-form generation tasks such as summarization. Under appropriately realistic assumptions, we empirically and theoretically show that ICL demonstrations alone are insufficient to teach LLMs the task's language and format distributions for generation.

We argue for explicit exposure to the task distributions and hypothesize that defining them by prompting enhances model performance. To this end, we present LongGuide, which efficiently generates two parallel streams of guidelines capturing task language and format properties: *(i) Metric Guidelines* (MGs) that instruct models to optimize self-evaluated metrics; and *(ii) Output Constraint Guidelines* (OCGs) that constrain generation at both token and sentence levels. LongGuide automatically selects the best combination of guidelines, improving both strong open- and closed-source LLMs by over 5% in both zero- and few-shot settings. We show that LongGuide is generalizable, learnable by weak models to enhance strong ones, and integrates synergistically with automatic prompt optimizers.

## 1 Introduction

In recent years, pre-trained large language models (LLMs) have demonstrated impressive instruction-based performance through zero- and few-shot learning capabilities (Brown et al., 2020; Chowdhery et al., 2022; OpenAI, 2022; Touvron et al., 2023; Jiang et al., 2023; Team et al., 2023). Notably, few-shot learning, termed as in-context learning (ICL), has proven highly effective and widely used to calibrate LLMs for applications (Dong et al., 2022). Formally, let $\mathcal{V}$ be the vocabulary of the LM. For a task $T$ at test time, the goal is to generate a token sequence $y \in \mathcal{Y} \subseteq \mathcal{V}^*$, given another input token sequence $x \in \mathcal{X} \subseteq \mathcal{V}^*$. Then, ICL generation using an LLM $\mathcal{M}$ is the generation conditioned on $x$ with $k$ task demonstrations $\{(x_1, y_1), ..., (x_k, y_k)\}$ concatenated into $d_f = [x_1, y_1, ..., x_k, y_k] \in \mathcal{D} \subseteq \mathcal{V}^*$. The probability distribution induced from $\mathcal{M} : \mathcal{V}^* \to \mathbb{R}$ is:

$$P_{\mathcal{M}}(y|d_f, x) := \prod_{t=1}^{|y|} \mathcal{M}_{y^t}([x_1, y_1, ..., x_k, y_k, x, y^{<t}]) \tag{1}$$

where $y = [y^1, \ldots, y^{|y|}]$ with $y^t \in \mathcal{V}$. Several prior studies attempt to explain the ICL capabilities of LLMs, advocating for the sufficiency of well-chosen $d_f$ as implicitly teaching the $\mathcal{M}$ to perform the tasks, especially classification ones (Saunshi et al., 2020; Xie et al., 2021; Wang et al., 2024). Central to their theoretical analyses is a strong assumption that the language model $\mathcal{M}$ fully captures the underlying distribution of the task's language; i.e., $P_{\mathcal{M}}(X) = P_T(X)$ where $P_T$ is the task-specific data distribution.

However, this assumption is often not met, especially with domain-specific terminologies (Cheng et al., 2024) (also see §B.1 for a case study), raising concerns about the actual sufficiency of ICL. Moreover, recent studies empirically show that ICL underperforms in long-form generation tasks involving multi-sentence or -paragraph answers (Sun et al., 2023a; Huang et al., 2024), highlighting

significant gaps in our understandings of the causes of these limitations and how to effectively instruct LLMs for these tasks. These challenges remain unsolved to date.

In this work, we first study the proficiency of ICL for long-form generation tasks. We empirically and theoretically highlight that if a language model fails to capture the task's *text properties (language and format)*, providing demonstrations alone with such properties cannot entirely resolve this (§2). This is because the model does not consistently apply them to all generated responses. Maintaining such properties in responses is crucial for accurately solving the task. Therefore, we argue that providing explicit task guidelines that comprehensively capture these text properties is essential for improving LLM performance. Figure 1 illustrates such an example where instructing LLMs explicitly by guidelines carrying certain properties (e.g., conciseness, #sentences) of the task output distribution improves both alignment with ground truth and generation quality.

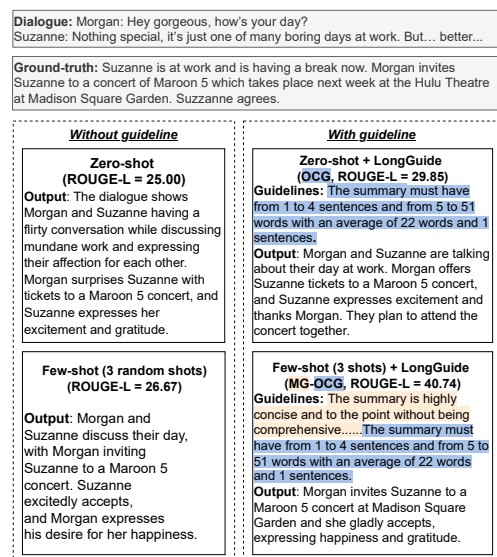

We then propose LongGuide (§3), a guideline-learning algorithm that efficiently generates two types of guidelines concurrently from limited task training data as supplementary instructions to enhance LLMs: (i) Metric Guidelines (MGs) that steer models to optimize self-evaluation guided metrics, inspired by prior studies in machine translation (Ranzato et al., 2015) and LLM self-evaluation (Ren et al., 2023); and (ii) Output Constraint Guidelines (OCGs) that impose constraints on generated outputs at the sentence and token levels, drawing on controllable generation research (Fan et al., 2018a). LongGuide is related to previous studies in task instruction construction (Wang et al., 2022b) and task understanding through task definitions (Yin et al., 2023). However, it differs by offering "post-hoc" instructions that guide LLMs to enhance responses based on learned quality and quantitative criteria.

Figure 1: ChatGPT results on a SAMSum example (Gliwa et al., 2019). With LongGuide, the generated output aligns better with ground truth, and the quality is also improved by removing verbose details: "The dialogue..." (ZS), "...discuss their day" (FS). See Appx.-Figure 14 for full texts.

LongGuide automatically identifies optimal guidelines, significantly enhancing distribution alignment and generation quality across seven generation tasks and one real-life chat LLM benchmark, including summarization, text simplification, translation, dialogue generation, and table-to-text generation. Its guidelines can enhance ICL performance through demonstrations (§5.2), improve non-instruct LLMs (§C.1), boost stronger models when learned by weaker ones (§C.2), and can be further optimized for usage using prompt optimization algorithms (§C.3). Notably, LongGuide is approximately at least 3.75 times more cost-efficient than prompt optimization algorithms (§F.3-Table 20) as it requires only four prompt variants to verify on the validation set while delivering superior performance.

## 2 ICL ALONE IS INSUFFICIENT FOR LONG-FORM GENERATION

A long-form generation dataset with $n$ samples for a task $T$ is defined as $D = \{(x_i^t, y_i^t)\}_{i=1}^n$ where $x_i^t$ and $y_i^t$ are input contexts and ground-truth sentence- or paragraph-long responses (Fan et al., 2019). For such tasks, preserving language and format properties of task-specific data during generation is essential for aligning outputs with ground truth. This is unlike classification, where outputs are predefined. We now demonstrate that ICL fails to enable LLMs to implicitly capture and maintain these properties during generation.

**Setup.** We first select metrics as properties commonly used for dialogue summarization. We follow Fu et al. (2023) to choose six: **(1) Semantic Coverage (COV)**; **(2) Factuality (FAC)**; **(3) Consistency (CON)**; **(4) Informativeness (INF)**; **(5) Coherence (COH)**; **(6) Relevance (REL)**. We also measure **(7) # tokens (NT)** and **(8) # sentences (NS)** of ICL responses, as these format metrics can significantly

| ICL w/ 5 demos | (1) COV | (2) FAC | (3) CON | (4) INF | (5) COH | (6) REL | (7) NT (mean) | (7) NT (std) |
|---|---|---|---|---|---|---|---|---|
| *Expected outcome* | *100%* | *100%* | *100%* | *100%* | *100%* | *100%* | *17.00* | *0.00* |
| Mistral-7B-v0.3 | 12% | 27% | 28% | 8% | 20% | 35% | 87.74 | 144.91 |
| Llama-3.1-8B | 12% | 42% | 50% | 4% | 32% | 47% | 271.81 | 379.48 |
| Qwen2.5-7B | 43% | **90%** | **85%** | **40%** | 78% | **96%** | 281.38 | 264.59 |
| Mistral-7B-it-v0.2 | 38% | 80% | 78% | 17% | 75% | 88% | 50.25 | 55.54 |
| Llama-3.1-8B-it | **44%** | 86% | 82% | 26% | **81%** | 87% | **34.72** | **45.29** |

Table 1: % of responses scored 5 on the (1)-(6) metrics, and the (mean, std) of the (7) #tokens of the responses. Qwen scored high on metrics (1)–(6) because it copies the input dialogue as the summarization outcome.

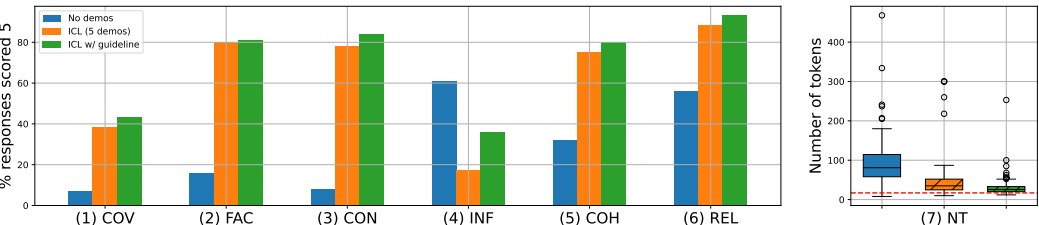

Figure 2: ICL property transfer experiments with Mistral-7B-it-v0.2. See Appx.-Figure 19 for a full example.

impact model performance (Fan et al., 2018a). *For each metric, we select the demonstrations having the same score and evaluate whether the ICL-generated responses maintain that score.*

Our experiments are performed on 100 random **SAMSum** samples (Gliwa et al., 2019) for each metric. We use **ChatGPT (gpt-3.5-turbo-1106)** (OpenAI, 2022) with **Self-consistency** (Wang et al., 2022a) to evaluate metrics (1)–(6) on a scale of 1–5, as it is an effective evaluator (Wang et al., 2023a). NLTK (Bird & Loper, 2004) assesses metrics (7)–(8). For ICL experiments, we examine five instruct and non-instruct models: **Mistral-7B-v0.3** (Jiang et al., 2023), **Llama-3.1-8B** (Dubey et al., 2024), **Qwen-2.5-7B** (Qwen Team, 2024), ans **Mistral-7B-it-v.02** (Jiang et al., 2023) and **Llama-3.1-8B-it** (Dubey et al., 2024). For metrics (1)–(6), we select demonstrations having a perfect score of 5, and for metrics (7)–(8) having 17 response tokens, spanning 2 sentences. For each metric, we further examine whether a **simple guideline**: "The output must maintain...{property}." (w/ guideline) can help instruct models (Mistral) maintain that property better during generation.

**Findings.** We present the $k = 5$ demonstration results in Table 1 and the case of Mistral-7B-it-v.02 in Figure 2 with metrics (1)–(7)[1]. We derive three surprising findings. Firstly, the ICL models do not achieve a 100% score of 5 on any metric and instruct models generally outperform non-instruct models. The highest percentage of score 5 on average is on COH and REL, where ICL models already excel while for critical summarization metrics such as INF and COV, they achieve only up to 20% to 40%. Notably, although all demonstrations contain 17 output tokens, fewer than 5% answers achieve this property. Secondly, increasing # demonstrations does not rectify this issue; the same trends persist across 3, 5, and 10. Finally, by adding a simple guideline shown in Figure 2, the percentages of answers maintaining the metrics are mostly improved, especially (7) and (8), verifying that adding guidelines is indeed helpful for instruct models to maintain these properties. Additionally, without ICL (and without instruction in our consideration), the model is entirely unable to solve the task.

**Theoretical intuitions.** Recall that $P_{\mathcal{M}}$ and $P_T$ are the probability functions of $\mathcal{M}$ and the task data, we prove the Remark A.1 which establishes that when $\mathcal{M}$ does not capture the true task data distribution (i.e., $P_{\mathcal{M}} \neq P_T$), demonstrations alone fail recover the desired alignment in the limit. As a result, certain task language and format properties, even when well-presented in demonstrations, may not be implicitly captured and preserved during LLM generation, as observed in Figure 2. We term this unsolved issue the **text property transfer (PT)** challenge: ensuring that $\mathcal{M}$ captures and preserves specific desired text properties observed in a limited set of labeled data to responses.

We hypothesize that explicitly guiding the model to optimize essential textual language and format properties at their levels matching the task data mitigates the mismatch between $P_{\mathcal{M}}$ and $P_T$, leading to improved performance. This hypothesis is formalized as Hypothesis A.1, and we present our complete theoretical intuition in §A.

---

[1]We also tested demonstration counts of $k = 3, 10$, all follow similar trends, see Appendix C.3's Figure 9.

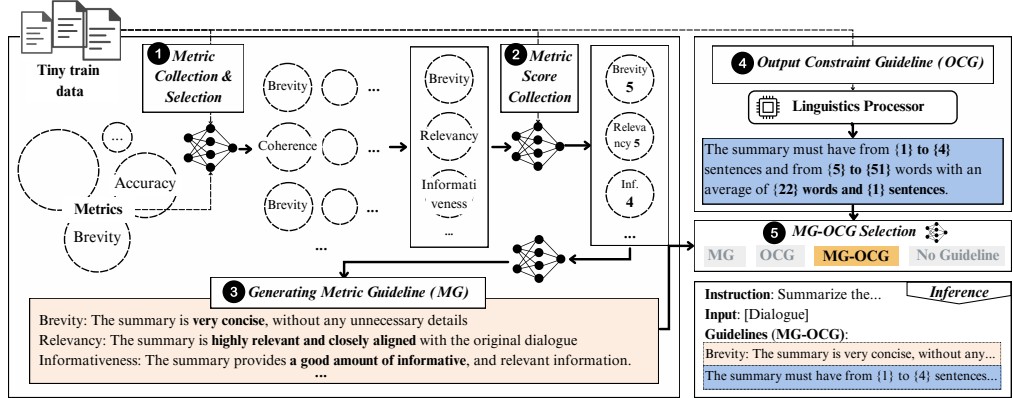

Figure 3: Overview of LongGuide. Orange and blue boxes denote the learned metric guideline (MG) and output constraint guideline (OCG).

# 3   LONGGUIDE: AN EFFICIENT GUIDELINE GENERATION ALGORITHM

**Motivations.**   As we have seen, providing textual **guidelines** instructing LLMs to optimize certain text property metrics can enhance them on responses, possibly because LLMs are optimizers (Yang et al., 2024). Inspired by §2, we propose LongGuide (Figure 3 and Algorithm 1), an algorithm that efficiently generates guidelines for LLMs to optimize self-evaluated text properties during generation. Specifically, Steps 1–3 focus on generating the Metric Guideline (MG) capturing the intrinsic language properties of the task via reference-free metrics. In parallel, Step 4 analyzes the answer format of the task and translates it to Output Constraint Guideline (OCG). The best combination of MG and OCG is selected for inference (Step 5). To ensure LongGuide's generalizability to new tasks, we assume access to at most 50 training samples: $D^{train} = \{(x_i^t, y_i^t)\}_{i=1}^n$.

**Step 1: Metric Collection & Selection.**   To learn a task's language properties, this step reasons to select appropriate language evaluation metrics for self-evaluation. For this purpose, we first construct a pool of evaluation metrics, $S$, applicable to any text generation task. $S$ consists of 27 distinct metrics from 4 main sources (Appx.-Table 13 for details). Specifically, we collect 3 metrics from ABC's of Communication (Wagner, 1963), 12 metrics from (Fu et al., 2023) for dialogue generation, summarization, data2text generation, and machine translation, and propose 12 more metrics for a broader evaluation coverage. We do not collect LM-based metrics, such as FactScore (Min et al., 2023), because it is challenging for LLMs to define and self-evaluate them. Additionally, we do not gather definitions of collected metrics, as their interpretations may vary across different tasks.

With $D^{train}$ and $S$, we then perform $K$ iterations to select the metrics. At each iteration, we randomly sample a batch of data from $D^{train}$ and instruct $\mathcal{M}$ to generate the top-5 most important metrics in $S$ for evaluating batch data properties via Chain-of-Thought prompting (Wei et al., 2022). We implement the top-5 constraint to avoid excessive metrics being selected. The final set of selected metrics, denoted by $M$, consists of the metrics chosen across all iterations *sorted in alphabetic order*.

**Step 2: Metric Score Collection via Self-evaluation.**   This step focuses on evaluating the selected metrics from $M$ on $D^{train}$ to capture the task properties. Motivated by prior studies (Wang et al., 2023a; Ren et al., 2023), we utilize $\mathcal{M}$ to score the metrics on a scale of 1–5. Specifically, for each train sample, $\mathcal{M}$ scores its ground-truth answer on all $\mathcal{M}$'s metrics via Self-Consistency (Wang et al., 2022a). The final metrics' scores, denoted as $scores_M$, are the average of scores over all train samples. Note that we separate this step from Step 1's metric selection because we want to evaluate each chosen metric on $D^{train}$ instead of the samples that led $\mathcal{M}$ to select it.

**Step 3: Generating Metric Guideline (MG).**   This step aims to generate a textual metric guideline (MG) that guides $\mathcal{M}$ to align generation outputs with task-specific properties from $scores_M$. MG is formed by concatenating metrics' definitions generated by $\mathcal{M}$ and tailored by $scores_M$ via the LLM instruction "Based on these scores on a scale of 5...define the expected quality of the output for each

---

**Algorithm 1** LongGuide

---

**Input:** $\mathcal{M}$ and its generation func. $G_{\mathcal{M}}$, train data $D^{train} = \{(x_i^t, y_i^t)\}_{i=1}^n$, linguistic processor $L$.
**Input:** Task instruction $I$, instruction to select metrics $I_M$, score metrics $I_{score}$, generate MG $I_{MG}$.
    • **Step 1: Metric Collection & Selection**
1: Collect the set of widely-used evaluation metrics $S$
2: $M = []$                                                 ▷ *the set of selected metrics*
3: **for** $K$ training iterations **do**
4:     Sample a batch $B$ from $D^{train}$
5:     $S_{sub} := G_{\mathcal{M}}([I_M, B, S])$            ▷ *top-5 metrics selected from $S$ for best evaluating $B$*
6:     $M = M \cup S_{sub}$
7: $M = M.sort()$
    • **Step 2: Metric Score Collection via Self-evaluation**
8: $s_{M_1} = \cdots = s_{M_m} = 0$           ▷ *the self-evaluated average scores of selected metrics*
9: **for** i, $(x, y)$ in enumerate($D^{train}$) **do**
10:     $\{s_{M_1}^i, ..., s_{M_m}^i\} := G_{\mathcal{M}}([I_{score}, x, y, M])$             ▷ *self-evaluation*
11:     Update $s_{M_j} = s_{M_j} + (s_{M_j}^i - s_{M_j})/(i+1)$ for all j in range(m)
12: $score_M = [s_{M_1}, \cdots, s_{M_m}]$
    • **Step 3: Generating Metric Guideline**
13: $\{d_{M_1}, ..., d_{M_m}\} := G_{\mathcal{M}}([I_{MG}, scores_M, M])$      ▷ *generate metrics' definitions w.r.t scores*
14: MG = joined with newline ($\{d_{M_1}, ..., d_{M_m}\}$)
    • **Step 4: Output Constraint Guideline**
15: Using $L$ to compute $(min_s, max_s, avg_s)$ of #sentences in $y_i^t$, and $(min_t, max_t, avg_t)$ of #tokens
16: OCG = "The response must have from $\{min_s\}$ to $\{max_s\}$ sentences and from $\{min_t\}$ to $\{max_t\}$ words with an average of $\{avg_t\}$ words and $\{avg_s\}$ sentences."
    • **Step 5: MG–OCG selection**
17: $G = \{$w/o guideline, $MG, OCG, MG$ & $OCG\}$
18: $LongGuide = \arg\max_{g \in G}(performance(\mathcal{M}|I, g, D^{train}))$     ▷ *automatic guideline selection*
**Output:** LongGuide

---

metric in natural language". We use these moderated definitions instead of raw $scores_M$ because LLMs better capture contextual nuances through descriptions rather than numerical scores (Singh & Strouse, 2024). Figure 3 illustrates an instance where "Informativeness" in the task "dialogue sum." achieving $4/5$ score from Step 2 is defined as "...good amount of informative...".

**Step 4: Output Constraint Guideline (OCG).** Research on controllable generation has extensively proposed constraints including ones on the length, which are broadly applicable, as well as linguistic or keyword, which are more task-specific (Fan et al., 2018a; He et al., 2022). In this step, we aim to establish a robust set of output constraints that apply universally to long-form generation tasks. We focus on six key constraints related to two distributions: the number of sentences and tokens in ground-truth answers. These constraints include minimum, maximum, and average counts, serving as basic statistics for length and expected values. The Output Constraint Guideline (OCG) instructs $\mathcal{M}$ to adhere to these statistics during generation.

**Step 5: MG–OCG selection.** For each task, models possess varying levels of inherent knowledge, resulting in different improvements gained by using MG and OCG (Table 4). This step determines the optimal MG and OCG combination by evaluating model performance on $D^{train}$ across configurations. The best-performing guideline configuration is then selected as the final output of LongGuide.

## 4 EXPERIMENTS

**Benchmarks.** We benchmark LongGuide on seven widely evaluated long-form generation tasks from four main categories, *summarization, text simplification, machine translation and generation*, and one real-life chat LLM benchmark. These tasks are **SAMSum** (Gliwa et al., 2019), **CNN/Daily Mail (3.0.0)** (See et al., 2017) and **XL-SUM** (Hasan et al., 2021) for summarization, **SWiPE** (Laban et al., 2023) for text simplification, **IWSLT-2017 en-ja** (Cettolo et al., 2017) for machine translation, **Synthetic-Persona-Chat** (Jandaghi et al., 2023) for dialogue generation, **CommonGen-Challenge**

| Method | SAMSum | | | | CNN | | | | SWiPE | | | |
|---|---|---|---|---|---|---|---|---|---|---|---|---|
| | R-L↑ | B-1↑ | BS↑ | Avg.JS↓ | R-L↑ | B-1↑ | BS↑ | Avg.JS↓ | R-L↑ | B-1↑ | BS↑ | Avg.JS↓ |
| Zero-shot (ZS) | 22.20 | 20.05 | 58.98 | 0.1014 | 19.23 | 20.43 | 60.59 | 0.1262 | 36.60 | 39.01 | **71.18** | 0.0565 |
| + OCG | 27.55 | 28.64 | 60.38 | 0.0402 | **22.46** | **27.82** | **61.37** | **0.0718** | 32.48 | 32.88 | 67.32 | 0.0650 |
| + MG | 27.81 | **28.81** | 60.06 | 0.0388 | 18.35 | 19.66 | 59.79 | 0.1413 | **38.21** | **40.83** | 70.87 | **0.0550** |
| + MG-OCG | **28.35** | 28.79 | **60.66** | **0.0375** | 22.05 | 26.97 | 61.18 | 0.0789 | 35.47 | 36.95 | 68.77 | 0.0554 |
| + LongGuide | **28.35** | 28.79 | **60.66** | **0.0375** | **22.46** | **27.82** | **61.37** | **0.0718** | **38.21** | **40.83** | 70.87 | **0.0550** |
| Few-shot (FS) | 27.13 | 27.21 | 61.70 | 0.0502 | 17.56 | 20.55 | 57.74 | 0.0844 | 39.47 | 39.76 | 70.56 | 0.0469 |
| + OCG | 27.84 | 29.91 | 61.08 | 0.0336 | 15.20 | 17.58 | 58.12 | 0.0922 | 29.54 | 30.32 | 68.82 | 0.0596 |
| + MG | 27.50 | 30.15 | 62.24 | 0.0352 | 18.13 | 20.94 | 57.89 | 0.0830 | **41.36** | **41.22** | **71.14** | **0.0450** |
| + MG-OCG | **30.65** | **31.72** | **62.73** | **0.0318** | 19.19 | 22.30 | 57.95 | 0.0814 | 38.56 | 37.87 | 68.54 | 0.0529 |
| + LongGuide | **30.65** | **31.72** | **62.73** | **0.0318** | 19.19 | 22.30 | 57.95 | 0.0814 | **41.36** | **41.22** | **71.14** | **0.0450** |

Table 2: Mistral performance verifying LongGuide considerably mitigates the text property transfer (PT) problem (§2): (1) the trends of ROUGE-L (R-L), BLEU-1 (B-1), BERTScore (BS), and Jensen–Shannon divergence (Avg. JS) show strong correlations, supporting our hypothesis Hypothesis A.1; (2) LongGuide substantially enhances Avg. JS scores, thereby mitigating the PT problem.

(Lin et al., 2020) for data-to-text generation, and (a subset of) **AlpacaEval2** (Dubois et al., 2024). We also benchmark the **reasoning tasks** in §D.2. See §E for data preprocessing details.

**Baselines and evaluations.** Since LongGuide is the first method to self-learn guidelines as additional instructions for generation, we compare it with the **zero-/few-shot prompting** baselines in this section, and **many-shot prompting** in §D.1. We also evaluate it against three of the strongest prompt optimization algorithms to date: **APO** (Pryzant et al., 2023) in this section, and **EvolPrompt** (Guo et al., 2024) and **adv-ICL** (Long et al., 2024) in §C.3, both of which optimize the input prompt on the $D^{train}$. We also compare LongGuide with **"General Guidelines"** in §5.2 where we ask the models to reason over demonstrations to generate task guidelines. For models, we empirically examine both strong open- and closed-source LLMs: **Mistral-7B-it v0.2** (Jiang et al., 2023) as an open-source model and **ChatGPT (gpt-3.5-turbo-1106)** (OpenAI, 2022) as a closed-source model. For evaluations, we use **ROUGE-L** (Lin, 2004) (recall-based) following Bai et al. (2024) (also for LongGuide's Step 5), and **GPT-4o-Judge** (OpenAI, 2024) as our main evaluation metrics. For GPT-4o-Judge, we evaluate how aligned the generated answer is with the reference answer and its quality on five criteria: *(i) Format consistency*; *(ii) Content completeness*; *(iii) Factuality*; *(iv) Style adherence*; *(v) Generation quality* on a scale of 1–10, following Zheng et al. (2023) (see §F.1 for full prompt). We also report **BLEU-1** (Papineni et al., 2002) (precision-based), **BERTScore** (Zhang et al., 2020) (meaning-based), and **Human evaluation** verifying the metric optimization and generation quality in §5.1. The results are averaged over **three runs**, with 95% CI of the t-test.

## 4.1 FINDINGS

We address three key research questions: (1) Does LongGuide address property transfer (PT), and if so, is it correlated with improved model performance? (2) How does it enhance model performance? (3) Are the enhancements attributed to MG, OCG, or both?

**LongGuide enhances PT which correlates with improved performance.** LongGuide effectively addresses the PT problem identified in §2. Our experiments are conducted on 3 benchmarks SAMSum, CNN, and SWiPE with Mistral under the zero-shot and few-shot settings. For each task, we first obtain the set of selected text properties from LongGuide that the model needs to optimize. We then measure the average of Jensen-Shannon divergence (Lin, 1991) between their score distributions (judged by ChatGPT) between the generated answers and the ground truth answers, across all selected properties, denoted as $Avg.JS$.

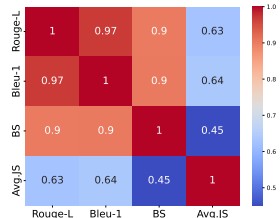

Figure 4: Pairwise Pearson correlation coefficient of metrics.

Table 2 presents the results. LongGuide significantly lowers the $Avg.JS$ scores compared to the baselines, demonstrating the effectiveness of guidelines for enhancing property transfer. Furthermore, our findings corroborate our hypothesis: across benchmarks, $Avg.JS$ exhibits moderate to strong positive correlations with the performance metrics (ROUGE-L, BLEU-1, BERTScore) measured by

| | Method | Sum. | | | Simplification | Translation | Dialogue Gen. | Table2Text |
|---|---|---|---|---|---|---|---|---|
| | | SAMSum | CNN (3.0.0) | XL-Sum | SWiPE | IWSLT17 en-ja | Syn. Persona | Comm.-Chall. |
| | #shots (ran.) | 3 | 3 | 5 | 3 | 3 | 5 | 5 |
| Mistral-it (0.2) | Zero-shot | 22.20 / 7.43 | 19.23 / 7.38 | 9.19 / 5.96 | 36.60 / 7.21 | 13.12 / 2.82 | 12.76 / 2.68 | 10.12 / 5.14 |
| | + APO | 23.77 / 7.31 | 19.53 / 7.40 | 12.06 / 5.85 | 36.92 / 7.21 | 14.45 / 2.91 | 10.66 / 2.41 | 11.21 / 4.68 |
| | + *LongGuide* | 28.35 / 7.73 | 22.46 / 7.45 | 14.38 / 6.29 | 38.21 / 7.32 | 16.53 / 3.45 | 14.69 / 4.45 | 25.20 / 6.81 |
| | *% gain (+)* | 6.15 / 0.30 | 3.23 / 0.07 | 5.19 / 0.33 | 1.61 / 0.11 | 3.41 / 0.63 | 1.93 / 1.77 | 15.08 / 1.67 |
| | Few-shot | 27.13 / 7.66 | 17.56 / 5.84 | 9.79 / 4.46 | 39.47 / 7.12 | 12.69 / 2.66 | 3.56 / 1.00 | 3.98 / 1.34 |
| | + APO | 26.23 / 7.44 | 18.18 / 5.89 | 11.99 / 4.55 | 39.55 / 7.11 | 14.08 / 2.92 | 4.26 / 1.05 | 5.45 / 2.05 |
| | + *LongGuide* | 30.65 / 7.72 | 19.19 / 5.99 | 15.23 / 5.06 | 41.36 / 7.24 | 16.62 / 3.40 | 5.25 / 3.93 | 25.05 / 6.65 |
| | *% gain (+)* | 3.52 / 0.06 | 1.63 / 0.15 | 5.44 / 0.40 | 1.89 / 0.12 | 3.66 / 0.74 | 1.69 / 2.93 | 21.07 / 5.31 |
| ChatGPT | Zero-shot | 23.83 / 7.43 | 20.12 / 7.44 | 10.80 / 5.96 | 45.09 / 7.28 | 36.13 / 7.62 | 19.46 / 6.04 | 24.21 / 6.53 |
| | + APO | 25.05 / 7.45 | 20.34 / 7.39 | 12.19 / 6.07 | 46.32 / 7.51 | 37.74 / 7.44 | 19.91 / 6.12 | 23.63 / 6.53 |
| | + *LongGuide* | 30.47 / 7.59 | 22.19 / 7.67 | 20.93 / 6.36 | 45.09 / 7.28 | 41.22 / 8.11 | 22.98 / 6.41 | 34.41 / 7.23 |
| | *% gain (+)* | 6.64 / 0.16 | 2.07 / 0.23 | 10.13 / 0.40 | 0.00 / 0.00 | 5.09 / 0.49 | 3.52 / 0.37 | 10.20 / 0.70 |
| | Few-shot | 22.21 / 7.32 | 14.51 / 4.38 | 11.42 / 5.95 | 33.72 / 5.07 | 31.93 / 7.25 | 16.10 / 4.67 | 22.08 / 4.19 |
| | + APO | 24.22 / 7.28 | 15.20 / 4.01 | 14.07 / 6.19 | 34.46 / 5.13 | 33.72 / 7.31 | 17.68 / 4.55 | 25.09 / 6.12 |
| | + *LongGuide* | 31.46 / 7.72 | 18.17 / 4.42 | 19.95 / 6.36 | 37.60 / 5.25 | 38.43 / 7.91 | 22.36 / 5.26 | 38.21 / 7.21 |
| | *% gain (+)* | 9.25 / 0.40 | 3.66 / 0.04 | 8.53 / 0.41 | 3.88 / 0.18 | 6.50 / 0.66 | 6.53 / 0.59 | 16.13 / 3.02 |

Table 3: ROUGE-L / GPT-4o-Judge results on seven long-form generation tasks. LongGuide remarkably outperforms baselines on most tasks and substantially enhances LLMs. BLUE-1 scores are reported in Appx.-11.

the Pearson correlation coefficient (Pearson, 1895) (Figure 4). In §D.6, we present the density plots for all metrics measured on the results with and without LongGuide.

**LongGuide significantly boosts instruction-based performance of LLMs.** Table 3 details our main experiment results on downstream tasks. Firstly, for baselines, zero-shot performance is interestingly higher than the few-shot for both models, and the gaps are huge for Synthetic Persona and CommonGen-Challenge. We hypothesize that models were partly exposed to task data during training, causing few-shot demonstrations to push the prompts out of distribution, leading to frequent refusals to answer. Meanwhile, LongGuide helps

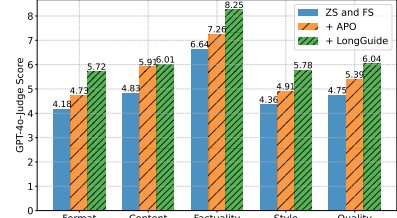

Figure 5: GPT-4-Judge scores over criteria.

models overcome this issue. Secondly, LongGuide substantially improves zero- and few-shot baselines by 6% on ROUGE-L and 0.8 on GPT-4o-Judge on average: improvement for few-shot prompting is surprisingly higher than in zero-shot, possibly because improving a stronger baseline is harder than a weaker one. Notably, LongGuide outperforms APO in most benchmarks, especially under the zero-shot setting, demonstrating that our strategy of optimizing text property tasks is markedly more effective than APO optimizing only ROUGE-L on limited data. Thirdly, we observe that LongGuide achieves the highest improvements on CommonGen-Challenge with 15.62% and lowest on SWiPE with 1.85% on ROUGE-L. These improvements are mainly because the answers generated by the baselines are often far longer than the ground truth. LongGuide rectifies this issue by controlling the output length and quality, leading to substantial performance gains. Fourthly, among the two models, LongGuide interestingly improves Mistral by 5.39%, while ChatGPT, regarded as stronger, improves by a larger margin, 6.58%. This suggests that LongGuide has the potential to benefit stronger models in the future. Among five GPT-4o-Judge criteria in Figure 5, LongGuide notably improves Format, Style, and Factuality, confirming its effectiveness in aligning model generation with ground-truth distributions. Finally, the significant gains in Quality, together with the ROUGE-L scores from Table 3 further verify that LongGuide also strongly enhances the generation quality.

**Where do the enhancements come from?** To identify the primary source of performance gains, we present the results of LLMs with LongGuide's components in Table 4. Firstly, MG-OCG combination (w/ MG-OCG) is the most useful for LLMs, observed to be the best 15 times, followed by OCG (w/ OCG) with 10, and MG (w/ MG) twice. While these statistics underscore the effectiveness of MG-OCG, OCG particularly proves itself highly effective in tasks such as summarization, translation, and table-to-text generation. Secondly, individual MG or OCG strengthens the prompting baselines, with OCG showing a slight edge. This is because while MG focuses on the language properties of answers, it does not directly control the output format, sometimes causing longer/shorter answers than the ground truths. Exceptionally, on SWiPE, OCG harms all models, whereas MG shows notably strong effectiveness with Mistral. Manual investigations reveal that ground-truth answers in SWiPE

| | Method | SAMSum | CNN (3.0.0) | XL-Sum | SWiPE | IWSLT17 en-ja | Synthetic Persona | CommGen-Chall. |
|---|---|---|---|---|---|---|---|---|
| **Mistral-7B-it (0.2)** | Zero-shot (ZS) | $22.20_{\pm0.43}$ | $19.23_{\pm0.34}$ | $9.19_{\pm0.03}$ | $36.60_{\pm0.59}$ | $13.12_{\pm1.39}$ | $12.76_{\pm1.54}$ | $10.12_{\pm0.02}$ |
| | + OCG | $27.55_{\pm0.98}$↑ | $\mathbf{22.46}_{\pm0.64}$↑ | $\mathbf{14.38}_{\pm0.15}$↑ | $32.48_{\pm1.91}$↓ | $\mathbf{16.53}_{\pm0.59}$↑ | $14.35_{\pm0.47}$↑ | $24.16_{\pm0.11}$↑ |
| | + MG | $27.81_{\pm1.17}$↑ | $18.35_{\pm0.60}$↓ | $9.37_{\pm0.25}$↑ | $\mathbf{38.21}_{\pm1.72}$↑ | $8.71_{\pm0.53}$↓ | $12.53_{\pm0.58}$↓ | $21.54_{\pm7.50}$↑ |
| | + MG-OCG | $\mathbf{28.35}_{\pm1.66}$↑ | $22.05_{\pm0.84}$↑ | $13.64_{\pm0.38}$↑ | $35.47_{\pm2.89}$↓ | $15.76_{\pm1.85}$↑ | $\mathbf{14.69}_{\pm1.08}$↑ | $\mathbf{25.20}_{\pm1.89}$↑ |
| | MG-OCG sel. | MG-OCG | MG-OCG | OCG | OCG | MG | OCG | MG-OCG |
| | Few-shot (FS) | $27.13_{\pm0.26}$ | $17.56_{\pm0.63}$ | $9.79_{\pm0.18}$ | $39.47_{\pm0.45}$ | $12.69_{\pm1.82}$ | $3.56_{\pm0.36}$ | $3.98_{\pm0.17}$ |
| | + OCG | $27.84_{\pm0.88}$↑ | $15.20_{\pm5.28}$↓ | $12.22_{\pm1.19}$↑ | $29.54_{\pm1.90}$↓ | $\mathbf{16.62}_{\pm0.81}$↑ | $5.06_{\pm1.05}$↑ | $\mathbf{25.05}_{\pm0.76}$↑ |
| | + MG | $27.50_{\pm2.08}$↑ | $18.13_{\pm5.28}$↑ | $11.80_{\pm2.06}$↑ | $\mathbf{41.36}_{\pm1.37}$↑ | $8.67_{\pm0.62}$↓ | $4.32_{\pm0.39}$↑ | $14.58_{\pm2.24}$↑ |
| | + MG-OCG | $\mathbf{30.65}_{\pm0.88}$↑ | $\mathbf{19.19}_{\pm0.49}$↑ | $\mathbf{15.23}_{\pm0.33}$↑ | $38.56_{\pm1.39}$↓ | $15.83_{\pm0.95}$↑ | $\mathbf{5.25}_{\pm0.94}$↑ | $5.94_{\pm1.00}$↑ |
| | MG-OCG Sel. | MG-OCG | MG-OCG | MG-OCG | MG | OCG | MG-OCG | OCG |
| **ChatGPT (1106)** | Zero-shot (ZS) | $23.83_{\pm0.54}$ | $20.12_{\pm0.27}$ | $10.80_{\pm0.18}$ | $\mathbf{45.09}_{\pm1.45}$ | $36.13_{\pm0.87}$ | $19.46_{\pm0.40}$ | $24.21_{\pm0.37}$ |
| | + OCG | $29.19_{\pm0.77}$↑ | $\mathbf{22.39}_{\pm0.82}$↑ | $\mathbf{20.93}_{\pm0.52}$↑ | $37.76_{\pm1.44}$↓ | $38.86_{\pm1.11}$↑ | $\mathbf{22.98}_{\pm2.65}$↑ | $\mathbf{34.41}_{\pm1.01}$↑ |
| | + MG | $25.38_{\pm0.79}$↑ | $20.37_{\pm0.41}$↑ | $10.42_{\pm1.15}$↓ | $45.06_{\pm2.96}$↓ | $37.88_{\pm2.42}$↑ | $19.91_{\pm0.59}$↑ | $17.23_{\pm2.57}$ |
| | + MG-OCG | $\mathbf{30.47}_{\pm1.57}$↑ | $22.19_{\pm0.65}$↑ | $20.02_{\pm0.89}$↑ | $41.38_{\pm4.91}$↓ | $\mathbf{41.22}_{\pm0.46}$↑ | $20.95_{\pm1.91}$↑ | $31.57_{\pm0.99}$↑ |
| | MG-OCG Sel. | MG-OCG | MG-OCG | OCG | ZS | MG-OCG | MG-OCG | OCG |
| | Few-shot (FS) | $22.21_{\pm2.35}$ | $14.51_{\pm0.80}$ | $11.42_{\pm0.13}$ | $33.72_{\pm2.61}$ | $31.93_{\pm1.88}$ | $16.10_{\pm2.61}$ | $22.08_{\pm0.63}$ |
| | + OCG | $30.00_{\pm1.07}$↑ | $\mathbf{18.17}_{\pm1.32}$↑ | $\mathbf{19.95}_{\pm1.38}$↑ | $16.68_{\pm1.29}$↓ | $38.57_{\pm1.81}$↑ | $\mathbf{22.36}_{\pm0.89}$↑ | $38.12_{\pm1.99}$↑ |
| | + MG | $29.43_{\pm0.83}$↑ | $15.45_{\pm2.16}$↑ | $12.49_{\pm0.59}$↑ | $19.36_{\pm1.40}$↓ | $\mathbf{39.45}_{\pm3.55}$↑ | $18.64_{\pm0.49}$↑ | $22.18_{\pm7.50}$↑ |
| | + MG-OCG | $\mathbf{31.46}_{\pm1.34}$↑ | $14.84_{\pm2.58}$↓ | $18.58_{\pm0.44}$↑ | $\mathbf{37.60}_{\pm2.85}$↑ | $38.43_{\pm2.37}$↑ | $19.47_{\pm1.20}$↑ | $\mathbf{38.21}_{\pm3.70}$↑ |
| | MG-OCG Sel. | MG-OCG | OCG | OCG | MG-OCG | MG-OCG | OCG | MG-OCG |

Table 4: ROUGE-L results with 95% CI from t-test. The gains of LongGuide's components vary across different models and tasks. The "MG-OCG selection" results are reported in Appx.-Table 14.

exhibit high variances in #sentences and #tokens which explains why OCG is not effective for this benchmark. Thirdly, an interesting case is ChatGPT with few-shot prompting on SWiPE, where individual MG and OCG impair performance but their combination enhances it. This shows evidence that MG and OCG complement each other. As discussed above, due to the uneven nature of answers in SWiPE, using MG or OCG alone may not work well for multiple samples, as MG and OCG only provide expected statistics. However, combining them could enhance performance by allowing them to complement each other. A such complement SWiPE example is outlined in Appx.-Figure 16.

## 5 DISCUSSION

We address five key questions about the usefulness, applicability, and generalizability of LongGuide. Its additional properties are provided in §C along with more method analyses in §D.

### 5.1 HUMAN EVALUATION: DOES LONGGUIDE ENHANCE GENERATION QUALITY?

We perform a human evaluation to quantify LongGuide's effect in helping LLMs optimize the selected metrics and enhance generation quality, as no automatic methods can address this need to date. For this purpose, we randomly select 50 zero-shot generated samples from the SAMSum and Synthetic Persona (since MG-OCG is the best for these datasets, Table 4) by ChatGPT and Mistral. Three English-native undergraduates are hired to rate whether ZS + LongGuide improves ZS on each of the selected MG and OCG metrics. Due to resource constraints, we evaluate 5 random MG metrics.

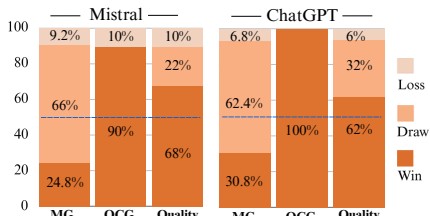

Figure 6: Win/Draw/Loss rates of w/ versus w/o LongGuide.

As shown in Figure 6, we notice that ZS + LongGuide outperforms ZS on 27.8% MG metrics on average, draws on 64.2%, and loses on only 8%. Specifically, among the MG metrics, "Brevity" shows the highest winning rate of 73% while "Relevance" obtains the lowest winning rate of 12%, possibly because ZS models can already generate highly relevant outcomes. Meanwhile, on the OCG metrics, LongGuide achieves a superior win of 95% on average. Finally, regarding the generation quality, our annotators prefer LongGuide output by up to 92%. These indicate that LongGuide not only aligns the outputs with the ground truths but also enhances the generation quality. The fine-grained scores of MG metrics are provided in §D.9, and we measure the annotators' agreement by Krippendorff's alpha (Krippendorff, 2022), obtaining $\alpha$=68.9%.

### 5.2 LONGGUIDE LEARNS FROM DEMONSTRATIONS TO BOOST ICL PERFORMANCE

Here, we revisit the question posed in §2 and verify that LongGuide learned from demonstrations substantially increases ICL performance. Our experiments using Mistral cover CNN, IWSLT17 en-ja, and CommGen-Chall datasets. Our experiments involve averaging the performance under zero- and few-shot settings. For **Baseline**, no guideline is utilized. For **Long-Guide on Demos**, we train LongGuide on demonstrations used in Table 3, in contrast to

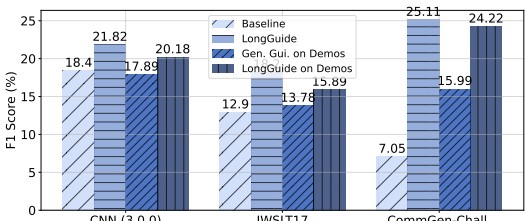

Figure 7: LongGuide learned from demonstrations substantially enhances Mistral performance (ROUGE-L).

the $D^{train}$ for the case of LongGuide. We add one more baseline, **General Guidelines (Gen. Gui.) on Demos**, where we ask the models to generate general task guidelines from demonstrations.

Figure 7 summarizes the results with details provided in Appx.-Table 15. Specifically, LongGuide trained on $D^{train}$ outperforms it on demonstrations, suggesting its possible scalability with more training data. Moreover, Gen. Gui. slightly worsens the Baseline on CNN. Manual examination reveals that Gen. Gui. produces surface-level task guidelines such as "Ensure the summary concisely conveys the key points of the source material.", missing task-specific constraints which LongGuide can effectively capture. Both LongGuide and LongGuide on Demos notably surpass the Baseline, and Gen. Gui., highlighting the benefits of capturing task-specific properties.

## 5.3 LONGGUIDE ON REAL-LIFE CHAT LLM BENCHMARK

We evaluate the effectiveness of LongGuide in aligning LLMs with desired real-world chats. Our experiments are conducted on a subset of 203 random samples from the AlpacaEval2 benchmark (Dubois et al., 2024) with ChatGPT (1106). Since AlpacaEval2 lacks training data, we select 5 random samples from the public Alpaca-GPT4 instruction-tuning dataset (alpaca-gpt4), despite it being relatively out-of-distribution (OOD) compared to AlpacaEval2.

Table 5 presents our findings. Few-shot demonstrations and OCG negatively impact performance, likely due to the OOD nature of Alpaca-GPT4 compared to AlpacaEval2. In contrast, with just 5 Alpaca-GPT4 samples, MG metrics, and LongGuide enhance performance by capturing certain response properties from GPT-4 (OpenAI, 2023), nearly doubling the zero-shot points.

| Methods | LC Win Rate | Win Rate |
|---|---|---|
| Zero-shot (ZS) | 11.08% | 3.17% |
| + OCG | 4.73% | 2.44% |
| + MG | **19.13%** | **7.07%** |
| + MG-OCG | 8.42% | 3.90% |
| **+ LongGuide** | **19.13%** | **7.07%** |
| Few-shot (FS) | 8.08% | 2.68% |
| + OCG | 7.73% | 3.45% |
| + MG | **12.65%** | **4.88%** |
| + MG-OCG | 12.63% | 4.88% |
| **+ LongGuide** | **12.65%** | **4.88%** |

Table 5: AlpacaEval2 experiments.

## 5.4 HOW DOES LLM HANDLE LONGGUIDE, AND CONTEXT GIVEN LONGGUIDE?

To analyze LongGuide's impact on LLMs, we perform a simple attention analysis to investigate (1) how LLMs attend to LongGuide and (2) utilize the input context when conditioning on LongGuide. Specifically, for (1), we calculate the average attention scores across all heads and layers for each guideline token. For (2), we evaluate the entropy of the attention scores overall context tokens. We experiment with Mistral on 100 SAMSum random samples. We learn two key findings.

Firstly, Mistral shows substantial attention to the guidelines. By using MG, 37.81% of attention is on guideline tokens. For OCG, it is 22.56%, and MG-OCG, 37.87%. Notably, the average attention on OCG tokens is higher than on context, while MG

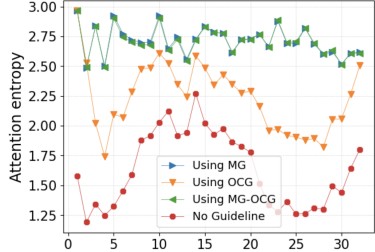

Figure 8: Entropy of attention over the input context across 32 Mistral layers.

and MG-OCG receive a fair amount, confirming mode attention on guidelines (Appx.-Table 16). Secondly, from Figure 8, Mistral exhibits more selective context attention when conditioned on guidelines. The largest entropy gap occurs in the first layer, where with guidelines, the model sparsely processes the context but, without them, is biased towards focusing narrowly on specific context parts. In the final layer, the model distributes attention more evenly with guidelines than without. Generally, MG stabilizes context use across layers, while OCG shows greater variance, likely because it does not directly control generation quality, therefore, the model bias almost exists as origin, as we can

see the trends of using OCG and no guidelines are relatively similar. These findings indicate that guidelines potentially improve context utilization and mitigate token bias.

## 5.5 ABLATION STUDIES OF LONGGUIDE'S STEPS

From Table 4, we identify the unique contributions of each step within LongGuide. Specifically, omitting Step 1 transforms it into OCG, whereas excluding Step 3 yields MG, and skipping Step 4 becomes MG-OCG. Here, we investigate LongGuide effectiveness when skipping Step 2, Metrics' scores collection: for selected metrics from Step 1, we directly task the models to optimize them for the generated answers. We experiment with Mistral on SAMSum, SWiPE, and CommGen-Chall. datasets because for these datasets, the best guideline combination includes MG.

| Methods | SAMSum | SWiPE | CommGen-Chall. |
|---|---|---|---|
| Zero-shot (ZS) | 22.20 / 7.43 | 36.60 / 7.21 | 10.12 / 5.14 |
| *+ LongGuide* | **28.35** / **7.73** | **38.21** / **7.32** | **25.20** / **6.81** |
| + LongGuide w/o step 2 | 26.99 / 7.49 | 36.90 / 7.22 | 25.03 / 6.66 |
| Few-shot (FS) | 27.13 / 7.66 | 39.47 / 7.12 | 3.98 / 1.34 |
| *+ LongGuide* | **30.65** / **7.72** | **41.36** / **7.24** | **25.05** / **6.65** |
| + LongGuide w/o step 2 | 30.37 / 7.70 | 35.54 / 6.28 | 24.15 / 5.82 |

Table 6: Mistral ROUGE-L / GPT-4o-Judge main ablation study with LongGuide when Step 2 is skipped.

The results in Table 6 show that without Step 2, the model performs worse, particularly for SAMSum and SWiPE in the zero-shot setting. Analysis of generation outputs reveals two failures when omitting Step 2: (1) metric conflicts where models struggle to optimize competing objectives like brevity and coverage jointly, and (2) task misalignment: without score-based validation, models optimize surface-level metric components forgetting deeper task requirements. A detailed case study in Appx.-Figure 18 demonstrates how Step 2's absence leads to verbose outputs that technically increases coverage but reduces overall quality.

## 6 RELATED WORK

**Automatic prompt design for long-form generation.** Long-form generation tasks are essential and have been studied extensively (Li et al., 2024). With LLM advancements, adapting these models for such tasks using prompt-based methods is critical yet challenging. Previous studies (Bang et al., 2023; Yang et al., 2023a; Hadi et al., 2023; Zhou et al., 2023b; Pan et al., 2024) highlight the limited efficacy of LLMs in producing outputs that resemble ground truths, as evaluated by ROUGE-L (Lin, 2004). Our approach autonomously composes supplementary contexts, integrating text evaluation metrics and format constraints. In addition, studies regarding enhancing instructions for LLMs (Wang et al., 2022b; Yin et al., 2023; Wang et al., 2023b), automatic prompt optimization (Zhou et al., 2023a; Pryzant et al., 2023), and demonstration selection (Yang et al., 2023b; Qin et al., 2023) are also related areas that can be developed in parallel and combined with our work (§C.3).

**Controllable generation with LLMs.** Controllable generation during fine-tuning has been studied extensively (Fan et al., 2018a; Lakew et al., 2019; Martin et al., 2020; He et al., 2022). More recently, researchers have explored prompting methods to control LLM generation. For instance, Sun et al. (2023b) found that LLMs struggle to meet fine-grained hard constraints, while Fonseca & Cohen (2024) proposed controlling stylistic features like keywords and narrative during generation, leading to improved LLM summarization outcomes. Although (Lu et al., 2023; Fonseca & Cohen, 2024) are closely related to our OCG, our approach goes beyond summarization and open-ended only features, as discussed in §3. We focus on universally applicable features across multiple tasks.

## 7 CONCLUSION

In this paper, we demonstrate that in-context learning (ICL) falls short in implicitly ensuring that large language models (LLMs) consistently preserve essential language and format properties in long-form generation tasks. To address this challenge, we introduce LongGuide, an efficient algorithm that automatically learns the critical language and format properties from task-specific data, converting them into textual guidelines for LLMs. Our results show that LongGuide significantly improves LLM performance across seven generation tasks and is highly generalizable, offering strong potential for various downstream applications with minimal data. This work paves the way for adaptive, task-specific prompt generation, advancing LLM adaptation.

## GENERALIZABILITY AND CUSTOMIZATION OF LONGGUIDE

LongGuide facilitates flexible generalization that allows customization and extension of guidelines MG and OCG for specific tasks, which we strongly recommend. For instance, in summarization, MG can focus on only 4-5 standard metrics from $S$ while integrating summary-specific metrics like "Summary Structure" and "Retention of Core Supporting Evidence." Simultaneously, OCG can impose stricter constraints on topics, keywords, grammar, or tones (Fan et al., 2018a; Lakew et al., 2019; Martin et al., 2020). Although LongGuide is primarily presented for general long-form generation, we strongly advise for these customizations to enhance its effectiveness.

## LIMITATIONS

Our study has several limitations. Firstly, our theoretical analysis focuses solely on the task language distribution which is $P_{\mathcal{M}}(X)$ or $P_{\mathcal{M}}(X|D_f)$ instead of the actual output distribution, which is $\arg\max_{y\in\mathcal{Y}} P_{\mathcal{M}}(Y = y \mid X)$ or $\arg\max_{y\in\mathcal{Y}} P_{\mathcal{M}}(Y = y \mid D_f, X)$. In our study, while leveraging the task language distribution allows us to hypothesize and highlight the limitations of demonstrations, shifting focus to the actual output distribution could yield more insights.

Secondly, LongGuide's learned guidelines are based on task-level and average statistics rather than sample-based details. We designed our framework at the task level to address limited data constraints, as we found that sample-based learning under these conditions leads to high errors. While task-level guidelines already demonstrate significant improvements for LLMs, sample-based guidelines could offer more tailored guidance, potentially leading to optimal results. Moreover, this average guidance approach may be ineffective for tasks with high variance in the statistics that LongGuide learns. In such cases, the final step of LongGuide can prevent performance decline by likely choosing no guideline. For example, we found this applies to Code2Text (Richardson et al., 2017) & StoryGeneration (Fan et al., 2018b).

Thirdly, LongGuide relies on models having a certain level of task knowledge to perform self-evaluation effectively, and LongGuide necessitates LLMs with strong instruction-following capabilities. However, we anticipate that cutting-edge AI language models will overcome this limitation both now and in the near future.

Lastly, the guidelines learned by LongGuide may not be useful for the tasks the models are trained on. This is because these guidelines might introduce out-of-distribution context relative to the training data, thereby reducing the effectiveness of the testing inference. For instance, while we see notable enhancements on the CommonGen-Challenge dataset (Lin et al., 2020), it's intriguing that we don't observe any improvements on the WebNLG (Gardent et al., 2017) and E2E NLG (Puzikov & Gurevych, 2018) datasets, despite their expected similarity. Given the popularity of these datasets, we suspect the models we tested may have been previously trained on them.

## ETHICAL CONSIDERATIONS

This method could be misused to optimize prompts for harmful purposes such as generating misinformation, hate speech, or privacy violations. While our method is not intended for such uses, it is impossible to completely prevent misuse. Although our method could enhance the efficiency and efficacy of bad actors, we do not anticipate that LongGuide is inherently more effective in these negative contexts than in positive applications. Finally, we employ annotators at an hourly rate of $20, which exceeds the local minimum wage requirements.

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

# A    THEORETICAL INTUITIONS

We now present a theoretical intuition to explain the observed phenomena. Recall that $P_\mathcal{M}$ and $P_T$ are the probability functions of $\mathcal{M}$ and the task data, then:

**Remark A.1.** *Under mild assumptions and $P_\mathcal{M} \neq P_T$, there exists $x \in \mathcal{X}$ such that $P_\mathcal{M}(X = x|D_f) \neq P_T(X = x)$.*

The proof is provided in §A.1. In essence, Remark A.1 asserts that when $\mathcal{M}$ fails to capture the true task data distribution, $d_f$ cannot recover the desired alignment in the limit. As a result, certain task language and format properties, even when well-presented in demonstrations, may not be implicitly captured and preserved during LLM generation. We term this unsolved issue the **text property transfer (PT)** challenge: ensuring that $\mathcal{M}$ captures and preserves specific desired text properties observed in a limited set of labeled data to responses.

**Definition A.1** (Text Property). For task $T$, a text property is defined by a feature map $f : \mathcal{Y} \to \mathbb{R}$ that measures a language or format property of its responses $y_i^t$ (e.g., coherence or length).

*We hypothesize that explicitly guiding the model to optimize essential text properties at levels matching the task data mitigates the mismatch identified in Remark A.1 between $P_\mathcal{M}$ and $P_T$, leading to improved performance.*

**Hypothesis A.1.** *Task $T$ can be decomposed into well-chosen text properties $\{f_1, \ldots, f_r\}$ with weights $\{\alpha_1, \ldots, \alpha_r\}$ such that by jointly optimizing them during the generation process, we can approximately optimize $T$'s objective: $\arg\min_{\theta \in \Theta} \mathcal{L} \approx \arg\min_{\theta \in \Theta} \sum_{i=1}^{r} \alpha_i \mathcal{L}_i$, where $\mathcal{L}$ is the task objective and $\mathcal{L}_i$ are property-specific objectives.*

We validate Hypothesis A.1 empirically in §4.1. It implies that if a prompting method $P_o$ + LongGuide (§3) matches or exceeds $P_o$ in metrics $f_i$ and text quality, it also does so in task performance.

**Remark A.2.** *Let $P_o$ be a prompting method and LongGuide be an instruction method defined in §3. If for each text property task $T_i$, $P_o$ + LongGuide matches or exceeds $P_o$ in both the metric $f_i$ and text generation quality, then $P_o$ + LongGuide matches or exceeds $P_o$ in task performance.*

## A.1    PROOF OF REMARK A.1

**Assumption A.1.** *There exists $x \in \mathcal{X}$ for which $P_\mathcal{M}(X = x) \neq P_T(X = x)$.*

This assumption is intuitive and realistic, recognizing that LLMs cannot fully capture the vast and nuanced complexity of real-world language beyond their training data. It contradicts the common assumption $P_\mathcal{M}(X) = P_T(X)$ made by prior studies (Xie et al., 2021; Wang et al., 2024). A simple empirical evidence is provided in §B.1. We also assume:

**Assumption A.2.** *Two probability functions are functionally zero equivalent if they act on the same input space and any arbitrary event causes both functions to be zero or non-zero. We assume that $P_T$ and $P_\mathcal{M}$ are functionally zero equivalent, i.e., $P_\mathcal{M}(X = x) = 0 \Leftrightarrow P_T(X = x) = 0 \,\forall x \in \mathcal{X}$.*

Note that Assumption A.2 is a relaxed version of the common assumption $P_\mathcal{M}(X) = P_T(X)$, and does not conflict with Assumption A.1.

*Proof of Remark A.1.* We prove this theorem by contradiction. Suppose the negation of Remark A.1 is true, i.e., there exists a $D_1 \in \mathcal{D}$ such that $\forall X \in \mathcal{X}, P_\mathcal{M}(X|D_1) = P_T(X)$ *(S1)*.

Now, let us consider the event $X \cap D_1^c$ where $D_1^c$ is the conjugate of event $D_1$, or $D_1^c = \mathcal{D} \backslash D_1$. We have $P_\mathcal{M}(X \cap D_1^c|D_1) = 0$. From the assumption *(S1)*, we derive $P_T(X \cap D_1^c) = 0$. From the Assumption A.2, since $P_\mathcal{M}$ and $P_T$ are functionally zero equivalent, we have $P_\mathcal{M}(X \cap D_1^c) = 0$.

Similarly, we can consider the event $X^c \cap D^c$ where $X^c$ is the conjugate of $X$, we arrive at $P_\mathcal{M}(X^c \cap D_1^c) = 0$.

Since the two $X \cap D_1^c$ and $X^c \cap D_1^c$ form a disjoint union of $D_1^c$, we derive $P_\mathcal{M}(D_1^c) = P_\mathcal{M}(X \cap D_1^c) + P_\mathcal{M}(X^c \cap D_1^c) = 0 + 0 = 0$. Since $D_1$ and $D_1^c$ form a disjoint union of $\mathcal{D}$, we have $P_\mathcal{M}(D_1) = 1$.

From the negation statement *(S1)*, we have $P_{\mathcal{M}}(X|D_1) = P_T(X) \ \forall X \in \mathcal{X}$. Since $X \cap D_1$ and $X \cap D_1^c$ form a disjoint union of $X$, we have $P_{\mathcal{M}}(X) = P_{\mathcal{M}}(X \cap D_1) + P_{\mathcal{M}}(X \cap D_1^c) = P_{\mathcal{M}}(X \cap D_1) + 0 = P_{\mathcal{M}}(X \cap D_1)$. We also have $P_{\mathcal{M}}(X|D_1) = \frac{P_{\mathcal{M}}(X \cap D_1)}{P_{\mathcal{M}}(D_1)}$ from Bayes's theorem, meaning that $P_{\mathcal{M}}(X|D_1) = P_{\mathcal{M}}(X \cap D_1) = P_{\mathcal{M}}(X)$ (since $P_{\mathcal{M}}(D_1) = 1$). Meanwhile, from the negation statement *(S1)*, we have $P_{\mathcal{M}}(X|D_1) = P_T(X)$, thus $P_{\mathcal{M}}(X) = P_T(X)$ for all $X \in \mathcal{X}$, which contradicts to our Assumption A.1. Therefore, our negation statement *(S1)* is false, leading to Remark A.1 is true. $\qquad\square$

## A.2 Proof of Remark A.2

*Proof of Remark A.2.* For simplicity, let us denote $\mathcal{L}(P)$ and $\mathcal{L}_i(P)$ as the losses of the language model $\mathcal{M}$ on task $T$ and text property task $T_i$, respectively when using the prompting method $P$.

From the assumption of Remark A.2 stating that for each text property task $T_i$, $P_o$ + LongGuide matches or exceeds $P_o$ in $f_i$, we have:

$$f_i(P_o + LongGuide) \geq f_i(P_o) \forall i \in \{1, \ldots, r\} \tag{2}$$

which is also equivalent to:

$$\alpha_i \mathcal{L}_i(P_o + LongGuide) \leq \alpha_i \mathcal{L}_i(P_o) \forall i \in \{1, \ldots, r\} \tag{3}$$

In other words, we obtain:

$$\sum_{i=1}^{r} \alpha_i \mathcal{L}_i(P_o + LongGuide) \leq \sum_{i=1}^{r} \alpha_i \mathcal{L}_i(P_o) \tag{4}$$

From Hypothesis A.1, we have $\sum_{i=1}^{r} \alpha_i \mathcal{L}_i(P_o + LongGuide) = \mathcal{L}(P_o + LongGuide)$ and $\sum_{i=1}^{r} \alpha_i \mathcal{L}_i(P_o) = \mathcal{L}(P_o)$. Then, Equation (4) is equivalent to $\mathcal{L}(P_o + LongGuide) \leq \mathcal{L}(P_o)$, which also means that $P_o + LongGuide$ is a more optimal prompting method than $P_o$.

$\qquad\square$

## B Empirical Case Studies Supporting §2

### B.1 Empirical Case Study for Assumption A.1

Here, we provide empirical evidence for Assumption A.1. It posits that for a given long-form generation test task $T$, the language model $\mathcal{M}$ does not entirely encapsulate $T$'s true language distribution. This assumption aligns with real-world scenarios since LLMs are not infinitely powerful: there exist task instances where they are out-of-distribution (OOD) for $\mathcal{M}$.

For example, let us consider Remark A.1 as the task $T$ and $\mathcal{M}$ be ChatGPT OpenAI (2022). The true language distribution is the specific vocabulary, technical jargon, and nuanced writing style typically found in mathematics. The full dialogue can be found at `https://chatgpt.com/share/0189761f-202b-46e6-bab0-39a80bd5340e`.

> Input: Given two probability functions $P_1$ and $P_2$. Suppose that there exists at least X such that $P_1(X) \neq P_2(X)$. Prove that for any D, there exists at least X' such that $P_1(X'|D) \neq P_2(X')$.

> ChatGPT output (30 Jul 24): To prove the statement, we need to understand the conditions under which the equality of the probability functions $P_1$ and $P_2$ given a condition $D$ breaks down.
> Let's start by stating the given conditions more clearly:
> 1. We have two probability functions $P_1$ and $P_2$. 2. There exists at least one event $X$ such that $P_1(X) \neq P_2(X)$.
> We need to show that for any event $D$, there exists at least one event $X'$ such that $P_1(X'|D) \neq P_2(X'|D)$.
> Let's break it down into steps...

Based on the above example, we can see that ChatGPT does not focus on proving $P_1(X'|D) \neq P_2(X')$. Instead, it misunderstood as proving $P_1(X'|D) \neq P_2(X'|D)$, showing that the ChatGPT's language distribution when solving the task is different from the true task language distribution.

### B.2 EMPIRICAL CASE STUDY FOR REMARK A.1

Empirical evidence supporting Remark A.1 is not possible to establish since it is an extremal result concerning existence and non-existence. Essentially, Remark A.1 says that if at the beginning, the two distributions of the task and language model are not the same ("first not the same") but functionally zero equivalent, then for any demonstrations, the two distributions of the task and language model conditioned on those demonstrations are not the same ("second not the same").

It's important to note that the data point causing the "first not the same" can differ from the data point causing the "second not the sam", and this "second not the same" data point needs to be examined by all possible demonstrations. This makes it difficult to empirically verify the theorem since the demonstration space is vast.

## C LONGGUIDE'S EXTRA PRELIMINARY PROPERTIES

### C.1 LONGGUIDE CAN IMPROVE NON-INSTRUCT MODELS

Using guidelines learned by LongGuide, we add more instructions to models. Therefore, we aim to examine whether non-instruct models can benefit from these guidelines. Our final conclusion is yes, LongGuide has strong potential to enhance non-instruct models.

**Setups.** Since non-instruct models might struggle to follow our instructions to generate the guidelines §7, we utilize the guidelines learned by an instruct model instead. We run our experiments with **Mistral-7B-v0.1**[2](Jiang et al., 2023) using the guidelines learned by Mistral-7B-Instruct-v0.2.

**Findings.** The results are provided in Table 7. We observe that Long-Guide improves more than half of the experiments, showing its potential effectiveness in enhancing even non-instruct models, especially for the translation task.

| Methods | CNN (3.0.0) | IWSLT17 | CommGen-Chall. |
|---------|-------------|---------|----------------|
| Zero-shot (ZS) | $7.60_{\pm 0.58}$ | $2.99_{\pm 0.83}$ | $\mathbf{10.96}_{\pm 0.36}$ |
| + OCG | $6.60_{\pm 0.74}\downarrow$ | $3.70_{\pm 0.29}\uparrow$ | $10.12_{\pm 0.56}\downarrow$ |
| + MG | $\mathbf{9.04}_{\pm 1.02}\uparrow$ | $\mathbf{5.39}_{\pm 0.93}\uparrow$ | $8.55_{\pm 0.74}\downarrow$ |
| + MG-OCG | $8.38_{\pm 0.91}\uparrow$ | $4.59_{\pm 0.97}\uparrow$ | $7.99_{\pm 0.70}\downarrow$ |
| + LongGuide | $\mathbf{9.04}_{\pm 1.02}\uparrow$ | $\mathbf{5.39}_{\pm 0.93}\uparrow$ | $\mathbf{10.96}_{\pm 0.36}$ |
| Few-shot (FS) | $3.14_{\pm 0.32}$ | $3.44_{\pm 0.83}$ | $4.67_{\pm 0.33}$ |
| + OCG | $2.24_{\pm 0.21}\downarrow$ | $3.86_{\pm 0.61}\uparrow$ | $8.11_{\pm 0.63}\uparrow$ |
| + MG | $\mathbf{3.24}_{\pm 0.26}\uparrow$ | $6.65_{\pm 0.97}\uparrow$ | $\mathbf{10.71}_{\pm 0.80}\uparrow$ |
| + MG-OCG | $2.99_{\pm 0.29}\downarrow$ | $\mathbf{7.88}_{\pm 0.91}\uparrow$ | $9.39_{\pm 0.89}\uparrow$ |
| + LongGuide | $2.24_{\pm 0.21}\downarrow$ | $\mathbf{7.88}_{\pm 0.91}\uparrow$ | $\mathbf{10.71}_{\pm 0.80}\uparrow$ |

Table 7: ROUGE-L performance of **Mistral-7B-v0.1** using Long-Guide learned by **Mistral-7B-Instruct-v0.2**. We observe that Long-Guide improves more than half of the experiments, showing its potential effectiveness in enhancing even non-instruct models, especially for the translation task.

### C.2 LONGGUIDE CAN BE TRANSFERABLE FROM WEAKER TO STRONGER MODELS

---

[2]https://huggingface.co/mistralai/Mistral-7B-v0.1

We find that the guidelines learned by LongGuide are transferable from weaker to stronger models. A weaker model can learn the guidelines at a low cost, which can then be used to enhance the performance of stronger models. This is particularly advantageous because powerful models are often closed-source and expensive to query, whereas open-source models are weaker but free to use.

| Methods | CNN (3.0.0) | IWSLT17 en-ja | CommGen-Chall. |
|---|---|---|---|
| ChatGPT Zero-shot (ZS) | $20.12_{\pm 0.27}$ | $36.13_{\pm 0.87}$ | $24.21_{\pm 0.37}$ |
| ChatGPT ZS w/ Mistral's MG | $21.41_{\pm 0.62}\uparrow$ | $39.66_{\pm 2.47}\uparrow$ | $29.95_{\pm 23.66}\uparrow$ |
| ChatGPT Few-shot (FS) | $14.51_{\pm 0.80}$ | $31.93_{\pm 1.88}$ | $22.08_{\pm 0.63}$ |
| ChatGPT FS w/ Mistral's MG | $13.96_{\pm 11.50}\downarrow$ | $32.34_{\pm 13.79}\uparrow$ | $33.34_{\pm 13.56}\uparrow$ |
| Mistral Zero-shot (ZS) | $19.23_{\pm 0.34}$ | $13.12_{\pm 1.39}$ | $10.12_{\pm 0.02}$ |
| Mistral w/ ChatGPT's MG | $19.67_{\pm 0.71}\uparrow$ | $7.98_{\pm 1.49}\downarrow$ | $6.29_{\pm 1.06}\downarrow$ |
| Mistral Few-shot (FS) | $17.56_{\pm 0.63}$ | $12.69_{\pm 1.82}$ | $3.89_{\pm 0.17}$ |
| Mistral FS w/ ChatGPT's MG | $19.00_{\pm 7.82}\uparrow$ | $11.86_{\pm 2.79}\downarrow$ | $3.61_{\pm 0.38}\downarrow$ |

Table 8: LongGuide can be transferable from weaker to stronger models, evaluated by ROUGE-L.

**Setups.** We demonstrate this through experiments on CNN (3.0.0), IWSLT17 en-ja, and CommGen-Chall, representing all the tasks. We used the MG generated by Mistral for experiments on ChatGPT and vice versa under both zero-shot and few-shot settings.

**Findings.** Table 8 outlines the results. We observe that Mistral's MG generally improves ChatGPT performance, but not vice versa. Explaining these phenomena, firstly, the OCG is transferable across models because it is independent of any specific model. Secondly, the MG, while it helps models capture task distributions, an MG learned from a stronger model may not benefit a weaker model, as the weaker model may misinterpret it. In contrast, the stronger model, with better text comprehension, can generalize task distributions from MG even when MG is poor and/or not well expressive generated by the weaker model.

### C.3 LongGuide can be Compared and Combined with Automatic Prompt Optimization Algorithms

The MG and OCG learned by LongGuide may not be fully optimized for LLMs. Hence, it's intuitive to suggest that LLMs could achieve even greater performance by adopting optimal guidelines. In this section, we illustrate that the guidelines learned by LongGuide can be further refined through discrete prompt optimization algorithms. This capability is advantageous for LongGuide, enabling its concurrent development and integration with automatic prompt optimization algorithms.

**Setup.** We employ two strong prompt optimizers, APO (Pryzant et al., 2023) and adv-ICL (Long et al., 2024), in our experiments. We also compare LongGuide with EvolPrompt (Guo et al., 2024) in this section. Here is our methodology: we integrated the guidelines generated by LongGuide into the prompt, including the input instruction and demonstrations. Subsequently, we applied the prompt optimizers to refine the input instruction, demonstrations, and guidelines. Our experiments were conducted using Mistral on datasets including CNN, IWSLT 2017 en-ja, and CommonGen-Challenge. Following our findings in Table 4, the guideline being optimized for CNN and IWSLT 2017 en-ja is OCG, while for CommonGen-Challenge is MG-OCG.

**Findings.** Our results are detailed in Table 9. In summary, when further optimizing the OCG using APO and adv-ICL for CNN and IWSLT 2017, we observed a slight improvement. This could be attributed to the OCG already being concise and straightforward, making it easier for models to grasp. However, for the CommonGen-Challenge dataset, which utilizes the MG-OCG guideline with more detail, APO and adv-ICL have a greater amount of material to optimize within the prompts. This led to a substantial improvement in performance compared to the other datasets.

| Methods | CNN (3.0.0) | IWSLT17 | CommGen-Chall. |
|---|---|---|---|
| Zero-shot (ZS) | $19.23_{\pm 0.34}$ | $13.12_{\pm 1.39}$ | $10.12_{\pm 0.02}$ |
| + APO | $19.53_{\pm 2.08}$ | $14.45_{\pm 1.84}$ | $11.21_{\pm 2.02}$ |
| + EvolPrompt | $20.16_{\pm 3.44}$ | $15.04_{\pm 2.12}$ | $14.06_{\pm 3.02}$ |
| + adv-ICL | $18.87_{\pm 2.69}$ | $15.01_{\pm 1.72}$ | $13.12_{\pm 2.21}$ |
| + LongGuide | $22.46_{\pm 0.64}$ | $16.53_{\pm 0.59}$ | $25.20_{\pm 1.89}$ |
| + LongGuide + APO | $\mathbf{22.76}_{\pm 1.04}\uparrow$ | $\mathbf{17.13}_{\pm 1.05}\uparrow$ | $\mathbf{27.01}_{\pm 1.01}\uparrow$ |
| + LongGuide + adv-ICL | $21.97_{\pm 3.21}\downarrow$ | $16.90_{\pm 2.15}\uparrow$ | $26.18_{\pm 3.47}\uparrow$ |

Table 9: Guidelines learned by LongGuide are further optimized by discrete prompt optimization frameworks bringing even better performance, with Mistral, evaluated by ROUGE-L.

| | | Sum. | | | Simplification | Translation | Dialogue Gen. | Table2Text | |
|---|---|---|---|---|---|---|---|---|---|
| | Method | SAMSum | CNN (3.0.0) | XL-Sum | SWiPE | IWSLT17 en-ja | Syn. Persona | Comm.-Chall. | *Avg.* |
| | #shots (ran.) | 3 | 3 | 5 | 3 | 3 | 5 | 5 | |
| Mistral-it (0.2) | Zero-shot (ZS) | 22.20 / 20.05 | 19.23 / 20.43 | 9.19 / 8.82 | 36.60 / 39.01 | 13.12 / 13.72 | 12.76 / 11.79 | 10.12 / 6.19 | *17.38* |
| | + APO | 23.77 / 22.02 | 19.53 / 21.46 | 12.06 / 11.50 | 36.92 / 39.41 | 14.45 / 15.49 | 10.66 / 10.05 | 11.21 / 7.12 | *18.26* |
| | *+ LongGuide* | **28.35 / 28.79** | **22.46 / 27.82** | **14.38 / 14.13** | **38.21 / 40.83** | **16.53 / 18.81** | **14.69 / 12.86** | **25.20 / 24.03** | *23.37* |
| | *% gain (+)* | 6.15 / 8.74 | 3.23 / 7.39 | 5.19 / 5.31 | 1.61 / 1.82 | 3.41 / 5.09 | 1.93 / 1.07 | 15.08 / 17.84 | *5.99* |
| | Few-shot (FS) | 27.13 / 27.21 | 17.56 / 20.55 | 9.79 / 8.32 | 39.47 / 39.76 | 12.69 / 13.78 | 3.56 / 2.67 | 3.98 / 1.94 | *16.32* |
| | + APO | 26.23 / 25.88 | 18.18 / 21.32 | 11.99 / 11.71 | 39.55 / 39.56 | 14.08 / 14.70 | 4.26 / 2.91 | 5.45 / 3.76 | *17.12* |
| | *+ LongGuide* | **30.65 / 31.72** | **19.19 / 22.30** | **15.23 / 14.02** | **41.36 / 41.22** | **16.62 / 17.92** | **5.25 / 4.46** | **25.05 / 21.90** | *21.92* |
| | *% gain (+)* | 3.52 / 4.51 | 1.63 / 1.75 | 5.44 / 5.70 | 1.89 / 1.46 | 3.66 / 4.14 | 1.69 / 1.79 | 21.07 / 19.96 | *5.61* |
| ChatGPT | Zero-shot (ZS) | 23.83 / 20.23 | 20.12 / 24.11 | 10.80 / 11.46 | 45.09 / 43.28 | 36.13 / 38.32 | 19.46 / 19.75 | 24.21 / 24.04 | *25.77* |
| | + APO | 25.05 / 22.90 | 20.34 / 21.88 | 12.19 / 12.52 | **46.32 / 44.89** | 37.74 / 39.01 | 19.91 / 19.80 | 23.63 / 24.18 | *26.45* |
| | *+ LongGuide* | **30.47 / 28.37** | **22.19 / 30.79** | **20.93 / 22.61** | 45.09 / 43.28 | **41.22 / 43.79** | **22.98 / 23.79** | **34.41 / 36.84** | *31.91* |
| | *% gain (+)* | 6.64 / 8.14 | 2.07 / 6.68 | 10.13 / 11.15 | 0.00 / 0.00 | 5.09 / 5.47 | 3.52 / 4.04 | 10.20 / 12.80 | *6.13* |
| | Few-shot (FS) | 22.21 / 25.37 | 14.51 / 17.52 | 11.42 / 10.83 | 33.72 / 32.69 | 31.93 / 32.68 | 16.10 / 18.10 | 22.08 / 23.52 | *22.34* |
| | + APO | 24.22 / 22.77 | 15.20 / 17.04 | 14.07 / 15.69 | 34.46 / 33.18 | 33.72 / 35.50 | 17.68 / 17.77 | 25.09 / 24.70 | *23.65* |
| | *+ LongGuide* | **31.46 / 30.04** | **18.17 / 18.52** | **19.95 / 22.49** | **37.60 / 35.66** | **38.43 / 42.84** | **22.36 / 20.31** | **38.21 / 37.64** | *29.55* |
| | *% gain (+)* | 9.25 / 4.67 | 3.66 / 1.00 | 8.53 / 11.66 | 3.88 / 2.97 | 6.50 / 10.16 | 6.53 / 2.21 | 16.13 / 14.12 | *7.21* |

Table 11: Supplemetary ROUGE-L / BLEU-1 results on seven long-form generation tasks showing that the trends of ROUGE-L and BLEU-1 scores are nearly identical.

# D    SUPPLEMENTARY RESULTS AND DISCUSSIONS

## D.1    ADDITIONAL BASELINES: USING MORE SHOTS FOR ICL

We supplement the results for CNN (3.0.0), SWiPE, and Comm.-Chall. in Table 10 where we use 10 shots for CNN, 50 shots for SWiPE, and Comm.-Chall up to the window size limit of gpt-3.5-turbo-1106 evaluated by ROUGE-L / GPT-4o-Judge scores.

We observe that while supplementing more shots to ChatGPT improves model's performance, LongGuide further boosts the ICL performance significantly for all three benchmarks.

| #shot | CNN (3.0.0) | SWiPE | Comm.-Chall. |
|---|---|---|---|
| 3-5 shots | 14.51 / 4.38 | 33.72 / 5.07 | 22.08 / 4.19 |
| + LongGuide | **18.17 / 4.42** | **37.60 / 5.25** | **38.21 / 7.21** |
| 10-50 shots | 20.55 / 6.67 | 44.04 / 6.07 | 28.18 / 4.85 |
| + LongGuide | **21.69 / 6.82** | **46.17 / 6.67** | **42.55 / 7.72** |

Table 10: Performance comparison of models with and without LongGuide across different datasets and shot settings.

## D.2    LONGGUIDE ON REASONING TASKS

We conduct experiments comparing LongGuide to various baselines on reasoning tasks. We select Mistral as our LLM, and GSM8K (Cobbe et al., 2021) and SVAMP (Patel et al., 2021) as benchmarks for evaluation. For each benchmark, we randomly sampled 200 instances from the test set for assessment and 50 instances from the train set to train the prompt optimizers and LongGuide.

The results are averaged over three runs, and outlined in Table 12. LongGuide slightly outperforms the Zero-shot and Few-shot baselines but falls short compared to prompt optimizers. Nonetheless, the findings confirm that additional instructions for LLMs can potentially improve the init model, leading to further enhanced reasoning performance with prompt optimization.

| Methods | GSM8k | SVAMP |
|---|---|---|
| Zero-shot (ZS) | 39.66 | 60.33 |
| + APO | 41.83 | 62.33 |
| + adv-ICL | 42.66 | 62.83 |
| + LongGuide | 40.83 | 63.33 |
| Few-shot (FS) | 32.33 | 61.66 |
| + APO | 34.33 | 63.00 |
| + adv-ICL | 35.00 | 62.66 |
| + LongGuide | 34.83 | 62.83 |

Table 12: Performance of LongGuide with Mistral on reasoning tasks.

## D.3    SUPPLEMENTARY RESULTS FOR §2

## D.4    UNDERSTANDING MG AND OCG: HOW DO THEY WORK (TOGETHER)?

**Metric Guideline (MG) (Step 1-3).**    To understand how models select and evaluate metrics, we analyze the specific metrics chosen for each task, their selection frequencies, and their average scores (Appx.-Table 17 and figs. 12 and 13 respectively). Overall,

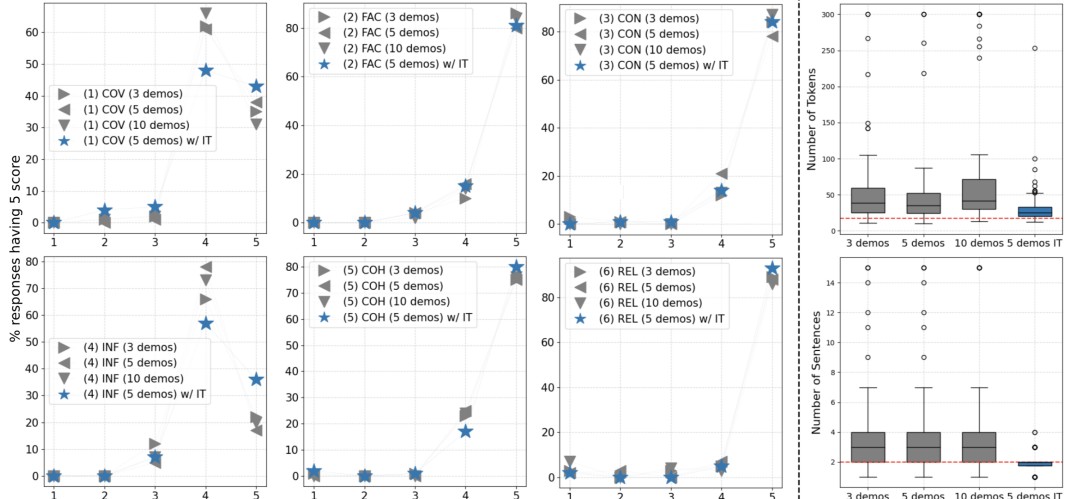

Figure 9: Property maintenance experiments with ICL full results. IT is the adding a simple guideline baseline.

each of the 27 metrics is selected and evaluated in at least one task. Among them, common linguistic metrics such as "Clarity" are frequently selected, while task-specific metrics like "Creativity" are less frequently chosen. By examining the scores of selected metrics, we find that common linguistic metrics generally achieve high scores, as anticipated. However, task-specific metrics like "Creativity" exhibit varying scores across tasks, indicating their differing importance and relevance. Additionally, we also find that within MG can conflict with each other, such as "Conciseness" and "Informativeness" (see Appx.-Figure 17 for an example). This underscores the importance of LongGuide's Step 2 in weighting the metrics.

**Output Constraint Guideline (OCG) (Step 4).**   We find that both the token and sentence constraints are crucial for LLMs (Appx.-D.12), with the sentence being more beneficial. We hypothesize that LLMs have better control over the number of sentences than tokens, as counting sentences is intuitively simpler than tokens. This can be observed in our experiment in §2.

**MG and OCG are complementary and non-interchangeable.**   MG and OCG complement each other rather than conflict, as partially discussed in §4.1. This is because MG language metrics primarily concern the characteristics of responses rather than their structural aspects such as sentence and token count, which is the main focus of the OCG. In addition, the MG and OCG are not interchangeable.  One might question whether adopting conciseness and brevity metrics could sufficiently alter the OCG, or if the OCG could effectively encompass the MG guideline. Our answer is no. While MG can steer LLMs towards brevity in responses, it lacks precise quantification for conciseness. Modern LLMs, often trained to generate verbose responses, may struggle to meet human conciseness without explicit statistics. Meanwhile, the OCG supplies them in the form of bins and means, yet these statistics alone do not directly address linguistic qualities. We provide examples as evidence supporting our arguments in Appx.-Figures 15 and 16.

### D.5   COLLECTED METRICS IN LONGGUIDE'S STEP 1 (§3)

Table 13 presents our 27 metrics collected for LongGuide's Step 1.

### D.6   JS DIVERGENCE OVER ALL LONGGUIDE METRICS WITH SAMSUM (§4.1)

Figure 10 presents density plots of MG and OCG metrics selected by Mistral under the few-shot (FS) setting, measured on ground-truth, FS, and FS w/ LongGuide answers. For Jensen–Shannon divergence, the lower is better.

| Source | Metrics | # |
|---|---|---|
| The ABC's of Communication (Wagner, 1963) | Accuracy, Brevity, Clarity | 3 |
| BARTScore (Yuan et al., 2021) | Relevance, Coherence | 2 |
| GPTScore (Fu et al., 2023) | Semantic Coverage, Factuality, Fluency, Informativeness, Consistency, Engagement, Specificity, Correctness, Understandability, Diversity | 10 |
| We propose | Completeness, Conciseness, Neutrality, Naturalness, Readability, Creativity, Rationalness, Truthfulness, Respect of Chronology, Non-repetitiveness, Indicativeness, Resolution | 12 |
| **Total** | 27 | 27 |

Table 13: Metrics collected for LongGuide's metric guideline (MG).

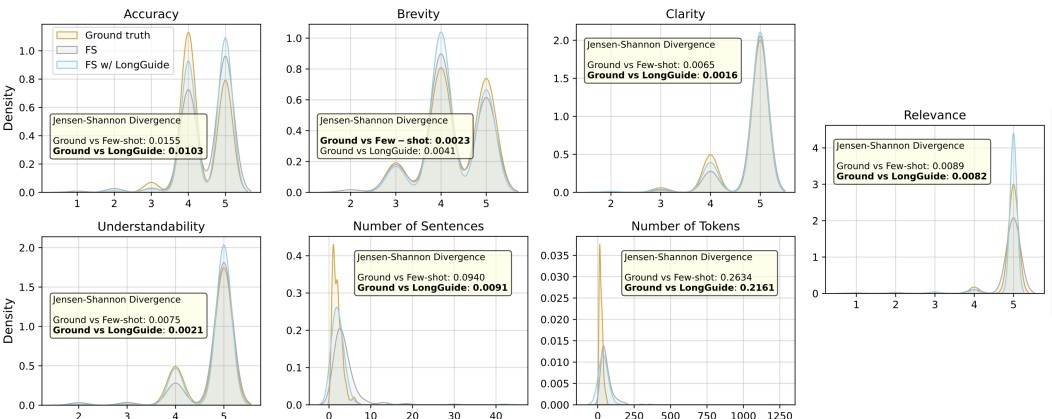

Figure 10: Density plots of MG and OCG metrics selected by Mistral under the few-shot (FS) setting, measured on ground-truth, FS, and FS w/ LongGuide answers. For Jensen–Shannon divergence, **lower is better**.

| | Method | Summarization | | | Simplification | Translation | Dialogue Generation | Table2Text |
|---|---|---|---|---|---|---|---|---|
| | | SAMSum | CNN (3.0.0) | XL-Sum | SWiPE | IWSLT17 en-ja | Synthetic Persona | CommGen-Chall. |
| | #shots (random) | 3 | 3 | 5 | 3 | 5 | 5 | 5 |
| Mistral-7B-it | Zero-shot (ZS) | 21.25 | 18.96 | 8.88 | 36.21 | 14.05 | 12.93 | 9.12 |
| | + OCG | 27.43 | **21.92** | **14.22** | 31.19 | **16.93** | 12.99 | 20.67 |
| | + MG | 27.68 | 18.02 | 10.26 | **36.74** | 11.06 | 13.74 | 19.98 |
| | + MG-OCG | **28.34** | 21.63 | 13.90 | 35.12 | 15.49 | **14.14** | **20.87** |
| | *MG-OCG Sel.* | *MG-OCG* | *OCG* | *OCG* | *MG* | *OCG* | *MG-OCG* | *MG-OCG* |
| | Few-shot (FS) | 25.55 | 17.30 | 9.85 | 39.29 | 13.52 | 6.19 | 4.01 |
| | + OCG | 27.31 | 16.45 | 12.47 | 29.85 | **17.58** | 6.45 | **20.50** |
| | + MG | 27.88 | 18.47 | 12.01 | **41.07** | 14.09 | 6.47 | 11.16 |
| | + MG-OCG | **30.01** | **19.87** | **14.89** | 39.40 | 17.02 | **8.06** | 5.18 |
| | *MG-OCG Sel.* | *MG-OCG* | *MG-OCG* | *MG-OCG* | *MG* | *OCG* | *MG-OCG* | *OCG* |
| ChatGPT | Zero-shot (ZS) | 24.21 | 19.54 | 10.78 | **45.11** | 36.22 | 19.68 | 24.23 |
| | + OCG | 28.81 | 21.88 | **20.66** | 37.58 | 38.45 | **23.09** | **35.04** |
| | + MG | 25.12 | 20.02 | 10.42 | 45.09 | 37.72 | 19.81 | 18.50 |
| | + MG-OCG | **29.79** | **21.99** | 19.91 | 42.72 | **41.50** | 20.82 | 30.09 |
| | *MG-OCG Sel.* | *MG-OCG* | *MG-OCG* | *OCG* | *ZS* | *MG-OCG* | *MG-OCG* | *OCG* |
| | Few-shot (FS) | 27.44 | 13.77 | 12.11 | 33.30 | 28.76 | 17.12 | 24.12 |
| | + OCG | 29.98 | **17.55** | **19.26** | 16.22 | 35.73 | **21.50** | 36.51 |
| | + MG | 28.89 | 14.03 | 12.75 | 19.14 | 36.09 | 19.12 | 21.99 |
| | + MG-OCG | **30.65** | 13.12 | 18.64 | **37.24** | **36.22** | 18.99 | **38.33** |
| | *MG-OCG Sel.* | *MG-OCG* | *OCG* | *OCG* | *MG-OCG* | *MG-OCG* | *OCG* | *MG-OCG* |

Table 14: MG-OCG selection results on $D^{train}$ set for the main experiments in Table 3, evaluated by ROUGE-L.

### D.7 Step 5 CD-MG Selection Results of LongGuide (§4.1)

The numerical MG-OCG selection results on $D^{train}$ are presented in Table 14, as also noted in Table 4. Overall, the performance of LongGuide on $D^{train}$ closely mirrors its performance on the testing tasks in Table 4. The only discrepancy is for the IWSLT17 en-ja task with ChatGPT using few-shot prompting: the optimal guideline combination on $D^{train}$ is MG-OCG (see Table 14), whereas the best on the testing set is MG (see Table 4).

### D.8 LongGuide can Generalize from Demonstrations (§5.2)

| Methods | CNN (3.0.0) | IWSLT17 en-ja | CommGen-Chall. |
|---|---|---|---|
| Zero-shot (ZS) | $19.23_{\pm0.34}$ | $13.12_{\pm1.39}$ | $10.12_{\pm0.02}$ |
| + OCG trained on $D^{train}$ | $\mathbf{22.46}_{\pm0.64}$ | $16.53_{\pm0.59}$ | $24.16_{\pm0.11}$ |
| + MG trained on $D^{train}$ | $18.35_{\pm0.60}$ | $8.71_{\pm0.53}$ | $21.54_{\pm7.50}$ |
| + MG-OCG trained on $D^{train}$ | $22.05_{\pm0.84}$ | $15.76_{\pm1.85}$ | $\mathbf{25.20}_{\pm1.89}$ |
| + LongGuide trained on $D^{train}$ | $\mathbf{22.46}_{\pm0.64}$ | $16.53_{\pm0.59}$ | $\mathbf{25.20}_{\pm1.89}$ |
| + OCG trained on Demos | $20.46_{\pm0.10}$ | $\mathbf{17.27}_{\pm1.83}$ | $23.97_{\pm0.47}$ |
| + MG trained on Demos | $18.33_{\pm0.25}$ | $8.63_{\pm1.08}$ | $18.98_{\pm0.52}$ |
| + MG-OCG trained on Demos | $19.16_{\pm0.37}$ | $14.00_{\pm3.42}$ | $24.46_{\pm2.43}$ |
| + LongGuide trained on Demos | $20.46_{\pm0.10}$ | $14.00_{\pm2.42}$ | $24.46_{\pm2.43}$ |
| Few-shot (FS) | $17.56_{\pm0.63}$ | $12.69_{\pm1.82}$ | $3.98_{\pm0.17}$ |
| + OCG trained on $D^{train}$ | $19.17_{\pm1.27}$ | $\mathbf{19.86}_{\pm2.93}$ | $25.05_{\pm0.76}$ |
| + MG trained on $D^{train}$ | $17.18_{\pm2.01}$ | $12.82_{\pm0.15}$ | $21.79_{\pm5.20}$ |
| + MG-OCG trained on $D^{train}$ | $\mathbf{21.18}_{\pm1.07}$ | $18.70_{\pm0.73}$ | $25.43_{\pm5.28}$ |
| + LongGuide trained on $D^{train}$ | $\mathbf{21.18}_{\pm1.07}$ | $\mathbf{19.86}_{\pm2.93}$ | $25.05_{\pm0.76}$ |
| + OCG trained on Demos | $16.88_{\pm1.44}$ | $19.40_{\pm1.39}$ | $\mathbf{28.28}_{\pm0.69}$ |
| + MG trained on Demos | $15.59_{\pm0.59}$ | $12.07_{\pm2.68}$ | $23.99_{\pm4.66}$ |
| + MG-OCG trained on Demos | $19.89_{\pm0.39}$ | $17.78_{\pm3.23}$ | $27.41_{\pm0.87}$ |
| + LongGuide trained on Demos | $19.89_{\pm0.39}$ | $17.78_{\pm18.43}$ | $23.99_{\pm4.66}$ |

Table 15: LongGuide learns the guidelines from only demonstrations with Mistral, evaluated by ROUGE-L.

Table 15 presents the numerical results of Figure 7 in §5.2. Even with only 3-5 exemplars as demonstrations, LongGuide effectively derives MG and OCG guidelines, benefiting the model. In this case, $D^{train}$ is the set of demonstrations, and the rest of LongGuide's steps remain unchanged.

### D.9 Human Evaluation Fine-grained Results (§5.1)

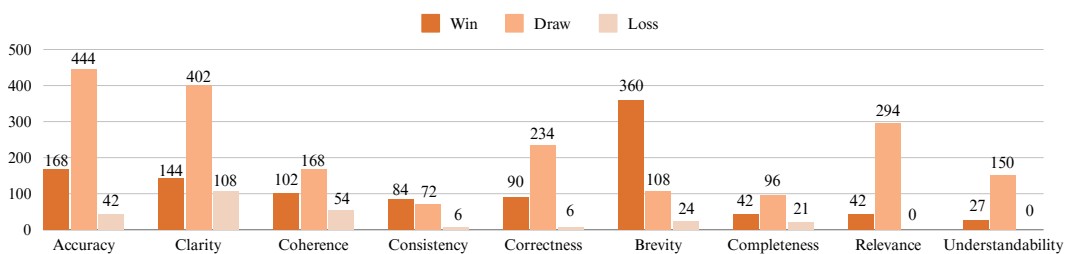

Figure 11: Fine-grained human evaluation results on evaluated MG metrics.

Figure 11 presents our fine-grained human evaluation results. Overall, LongGuide shows the best in terms of "Accuracy" and "Clarity", with a significant number of winning ratings. This suggests that the generated text is factually correct and easy to understand. Meanwhile, LongGuide shows more mixed results in terms of "Clarity" and "Coherence". While there is still a high winning rating, the proportion of draw and loss ratings is also relatively high, possibly because improving "Brevity" can somehow reduce the "Clarity".

|            | MG     | Context (MG) | OCG    | Context (OCG) | MG-OCG | Context (MG-OCG) |
|------------|--------|--------------|--------|---------------|--------|------------------|
| Per token  | 0.0019 | 0.0064       | 0.0133 | 0.0077        | 0.0017 | 0.0064           |
| All        | 37.81% | 62.19%       | 22.56% | 77.44%        | 37.87% | 62.13%           |

Table 16: Attention score over guideline and context tokens of Mistral.

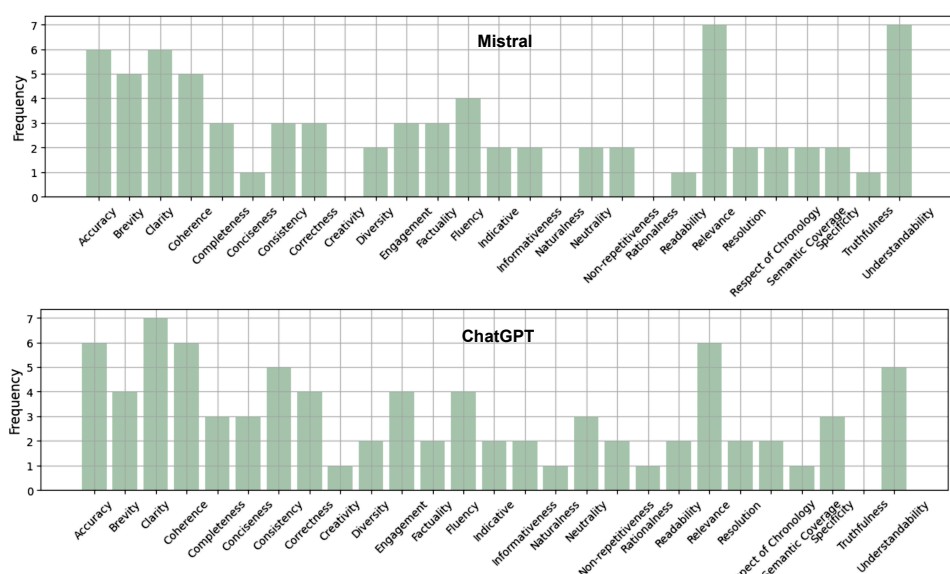

Figure 12: Frequency of metrics selected as the metric guideline.

## D.10 ATTENTION ANALYSIS FOR GUIDELINE TOKENS (§5.4)

Table 16 shows our simple attention analysis.

| Task | Model | Selected Metrics |
|------|-------|------------------|
| SAMSum | Mistral | ['Accuracy', 'Brevity', 'Clarity', 'Relevance', 'Understandability'] |
|  | ChatGPT | ['Accuracy', 'Brevity', 'Clarity', 'Relevance', 'Understandability'] |
| CNN | Mistral | ['Accuracy', 'Brevity', 'Clarity', 'Coherence', 'Completeness', 'Engagement', 'Readability', 'Relevance', 'Truthfulness', 'Understandability'] |
|  | ChatGPT | ['Accuracy', 'Brevity', 'Clarity', 'Coherence', 'Completeness', 'Conciseness', 'Engagement', 'Neutrality', 'Readability', 'Relevance', 'Specificity'] |
| XLSum | Mistral | ['Accuracy', 'Brevity', 'Clarity', 'Coherence', 'Completeness', 'Consistency', 'Correctness', 'Diversity', 'Engagement', 'Factuality', 'Fluency', 'Indicative', 'Informativeness', 'Neutrality', 'Non-repetitiveness', 'Relevance', 'Resolution', 'Respect of Chronology', 'Semantic Coverage', 'Specificity', 'Understandability'] |
|  | ChatGPT | ['Accuracy', 'Brevity', 'Clarity', 'Coherence', 'Completeness', 'Consistency', 'Correctness', 'Diversity', 'Engagement', 'Factuality', 'Fluency', 'Indicative', 'Informativeness', 'Neutrality', 'Non-repetitiveness', 'Rationalness', 'Relevance', 'Resolution', 'Respect of Chronology', 'Semantic Coverage', 'Specificity', 'Understandability'] |
| SWiPE | Mistral | ['Accuracy', 'Brevity', 'Clarity', 'Relevance', 'Understandability'] |
|  | ChatGPT | ['Accuracy', 'Brevity', 'Clarity', 'Coherence', 'Conciseness', 'Consistency', 'Correctness', 'Readability', 'Understandability'] |
| IWSLT17 en-ja | Mistral | ['Accuracy', 'Clarity', 'Coherence', 'Consistency', 'Correctness', 'Factuality', 'Fluency', 'Relevance', 'Understandability'] |
|  | ChatGPT | ['Accuracy', 'Clarity', 'Coherence', 'Consistency', 'Correctness', 'Factuality', 'Fluency', 'Relevance', 'Understandability'] |
| Synthetic Persona | Mistral | ['Accuracy', 'Brevity', 'Clarity', 'Coherence', 'Completeness', 'Consistency', 'Correctness', 'Diversity', 'Engagement', 'Factuality', 'Fluency', 'Indicative', 'Informativeness', 'Neutrality', 'Non-repetitiveness', 'Relevance', 'Resolution', 'Respect of Chronology', 'Semantic Coverage', 'Specificity', 'Understandability'] |
|  | ChatGPT | ['Accuracy', 'Clarity', 'Coherence', 'Consistency', 'Correctness', 'Diversity', 'Engagement', 'Fluency', 'Indicative', 'Informativeness', 'Neutrality', 'Non-repetitiveness', 'Relevance', 'Resolution', 'Respect of Chronology', 'Specificity', 'Understandability'] |
| CommGen-Chall. | Mistral | ['Coherence', 'Conciseness', 'Fluency', 'Relevance', 'Understandability'] |
|  | ChatGPT | ['Clarity', 'Coherence', 'Completeness', 'Conciseness', 'Consistency', 'Creativity', 'Engagement', 'Fluency', 'Naturalness', 'Relevance'] |

Table 17: Selected metrics by tasks by Mistral and ChatGPT.

## D.11 WHICH METRICS WERE SELECTED THE MOST FOR MG? (§D.4)

To better understand how models select and evaluate metrics, we analyze the specific metrics chosen for each task (Table 17), their selection frequencies (Figure 12), and their average scores (Figure 13). We observe that both models consistently choose key metrics like "Accuracy," "Clarity," "Relevance," and "Understandability," which are important for many language tasks. They also adjust their metric choices based on the tasks. For example, specific tasks like CNN and XL-Sum include additional metrics such as "Engagement" and "Semantic Coverage." This suggests that the models select metrics reasonably, based on the needs of the task, rather than showing a preference for certain metrics. Overall, the variety and suitability of the selected metrics show that the process is fair and appropriate for the tasks.

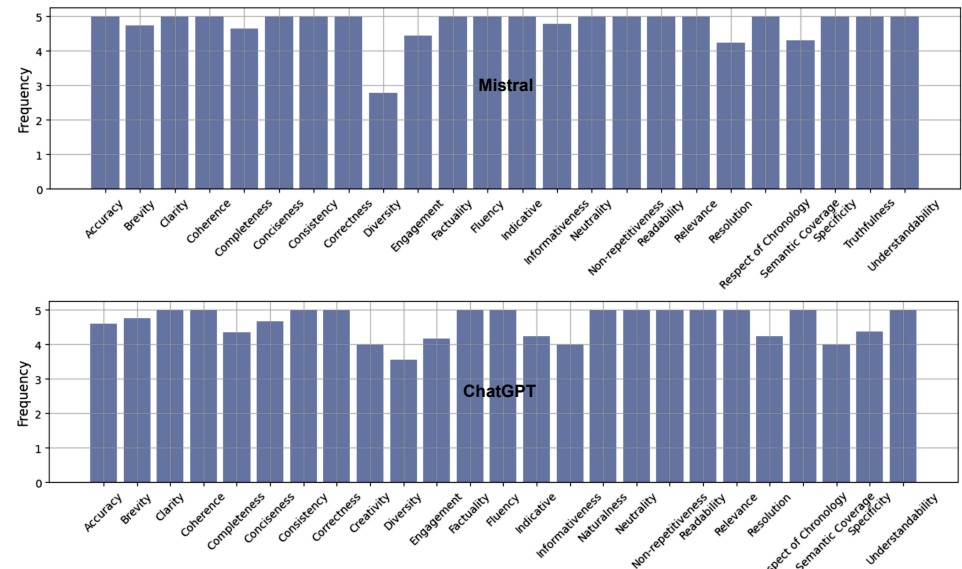

Figure 13: Average scores of metrics as the metric guideline.

## D.12 EXTRA ABLATION STUDIES: WITHOUT OCG'S TOKEN OR SENTENCE CONSTRAINT (§D.4)

| Methods | CNN (3.0.0) | IWSLT17 en-ja | CommGen-Chall. |
|---|---|---|---|
| Zero-shot (ZS) | $19.23_{\pm0.34}$ | $13.12_{\pm1.39}$ | $10.12_{\pm0.02}$ |
| + LongGuide | $\mathbf{22.46}_{\pm0.64}$ | $\mathbf{16.53}_{\pm0.59}$ | $\mathbf{25.20}_{\pm1.89}$ |
| + LongGuide w/o Token Constraint | $21.54_{\pm0.52}\downarrow$ | $14.09_{\pm1.07}\downarrow$ | $21.49_{\pm2.15}\downarrow$ |
| + LongGuide w/o Sentence Constraint | $20.92_{\pm0.23}\downarrow$ | $10.02_{\pm4.17}\downarrow$ | $13.32_{\pm0.73}\downarrow$ |
| Few-shot (FS) | $17.56_{\pm0.63}$ | $12.69_{\pm1.82}$ | $3.98_{\pm0.17}$ |
| + LongGuide | $\mathbf{21.18}_{\pm1.07}$ | $\mathbf{19.86}_{\pm2.93}$ | $\mathbf{25.05}_{\pm0.76}$ |
| + LongGuide w/o Token Constraint | $20.30_{\pm1.46}\downarrow$ | $19.75_{\pm1.47}\downarrow$ | $20.30_{\pm1.46}\downarrow$ |
| + LongGuide w/o Sentence Constraint | $15.89_{\pm2.26}\downarrow$ | $12.57_{\pm2.99}\downarrow$ | $12.20_{\pm3.91}\downarrow$ |

Table 18: Mistral results when omitting OCG's Token or Sentence Information, showing the importance of OCG's Token and Sentence information, evaluated by ROUGE-L.

Since OCG's token information and sentence information are the two types of information emphasized in OCG, we further investigate the importance of each type of information. The empirical experiments are conducted with Mistral on CNN, IWSLT-2017 en-ja, and CommonGen-Challenge. We present the results in Table 18. We observe that skipping OCG's token information or sentence information would hurt the performance. Specifically, the results drop more significantly when sentence information is omitted, and even fall below the Zero-shot score in CNN Few-shot with LongGuide and IWSLT17 en-ja Few-shot with LongGuide. The performance drops significantly in the CommonGen-Challenge Few-shot case, with a fall of 55.20%. Due to the volatility of the token count in a sentence, it is hard to estimate the other information with only one type of information given. Therefore, both types of information should be provided to better capture the text distribution.

## E IMPLEMENTATION DETAILS

**Task benchmark preprocessing.** We chose the newest versions of the above datasets. For each dataset except Synthetic-Persona-Chat, we sample 200 samples from the test set for our evaluation, following Bai et al. (2024), and 50 random samples from the train set for $D^{train}$. For Synthetic-Persona-Chat, we randomly sample 25 dialogues from its test set for our evaluation (678 utterances in total) and 3 dialogues from its train set where 50 random utterances are selected for $D^{train}$.

**Prompting baselines' hyperparameters.** We present the implementation and hyperparameters' details for our proposed LongGuide as well as prompting baselines below.

- **LongGuide.** We set the batch size is 5 and number of iterations is also 5 for LongGuide's step 1. For steps 2, 3, and 4, no hyperparameter involves. For the evaluations by Self-consistency (Wang et al., 2022a), we sample 3 results.

- **APO (Pryzant et al., 2023).** We set the number of optimization iterations is 5. We use 1 sample with the lowest ROUGE-L score as the error sample for generating gradients, following Long et al. (2024). At each iteration, 5 textual gradients are generated, and 5 new prompts are sampled from textual gradients. Finally, 1 paraphrase of the input prompt is sampled at each optimization iteration.

- **adv-ICL (Long et al., 2024).** We use 3 iterations with a batch size of 5 as suggested by Long et al. (2024). At each iteration, the number of new prompts sampled is 5.

**Models' hyperparameters.** The models' hyperparameters are presented below.

- **GPT-4o.** We use *gpt-4o* for our experiments. We use a window size of 2048 and Nucleus Sampling (Holtzman et al., 2019) as our decoding strategy with a $p$ value of 1 and temperature of 0.1. We use the system role as "You are a helpful assistant!".

- **ChatGPT.** We use *gpt-3.5-turbo-1106* for our experiments. We use a window size of 1500 and Nucleus Sampling (Holtzman et al., 2019) as our decoding strategy with a $p$ value of 1. We use the system role as "You are a helpful assistant!".

- **Mistral-7B-it-v0.2.** We use a window size of 1500, and Sampling decoding strategy (Holtzman et al., 2019) ($do\_sampling = True$). We load the model from Huggingface Transformers library (Wolf et al., 2020) with the model id is "mistralai/Mistral-7B-Instruct-v0.2". We do not set any explicit system role.

# F   PROMPTS AND PROMPTING ANALYSIS

## F.1   GPT-4O-JUDGE'S PROMPT

Our GPT-4o-Judge prompt evaluating the generated response and the reference is heavily motivated by Zheng et al. (2023).

Please act as an impartial judge and evaluate how well an assistant's answer aligns with the reference answer and the quality of the assistant's answer. You will be given a user prompt, a reference answer and an assistant's answer. Your evaluation must consider the following criteria:
- Format consistency: ensuring the generated response matches the length and structure of the reference.
- Content completeness: evaluating whether all key points present in the reference are included in the assistant's answer.
- Factuality: checking for factual correctness of the assistant's answer.
- Style adherence: ensuring that the tone, style, and level of detail of the of the assistant's answer match the reference.
- Assistant's answer quality: assessing how well the response satisfies the user's requirements.
Begin your evaluation by providing a short explanation for each. Be as objective as possible. After providing your explanation, please rate the response on all the criterion on a scale of 1 to 10 by strictly following this format:
[The Start of Explanation]
...
[The End of Explanation]
[The Start of Ratings]
{
"Format": 1-10,
"Content": 1-10,
"Factuality": 1-10,
"Style": 1-10,
"Quality": 1-10,
}
[The End of Ratings]
[User Prompt]
user_prompt
[The Start of Reference Answer]
answer_ref
[The End of Reference Answer]
[The Start of Assistant's Answer]
answer_a
[The End of Assistant's Answer]

## F.2 CHATGPT PROPERTY SCORER PROMPT

> You are an expert in evaluating the quality of a text generation task. You possess a nuanced understanding of various critical aspects. Brevity is paramount for you, ensuring concise expression without sacrificing essential information. Clarity is essential for comprehension, ensuring that your text is easily understood by the intended audience. Relevance ensures that the generated content aligns closely with the given context or prompt. Neutrality is crucial, maintaining an impartial tone devoid of bias. Coherence ties together ideas seamlessly, fostering a logical flow within your text. Completeness guarantees that all relevant points are addressed adequately. Specificity enhances precision, providing detailed and accurate information. Respect of chronology ensures temporal coherence, maintaining the chronological order of events. Accuracy demands factual correctness, avoiding errors or misinformation. Non-repetitiveness prevents redundancy, ensuring freshness in your expression. Indicative language aids in signaling key points or conclusions. Lastly, resolution ensures that your text concludes satisfactorily, resolving any questions or issues raised throughout.
> Input: {dialogue}
> Output: {generated_summary}
> Your task is to evaluate the following criteria in a scale of 1-5, with 1 is worst and 5 is best.
> {
> "Semantic Coverage": 1-5,
> "Factuality": 1-5,
> "Consistency": 1-5,
> "Informativeness": 1-5,
> "Coherence": 1-5,
> "Relevance": 1-5
> }
> The definitions of the criteria are:
> Semantic Coverage (COV): The extent to which a dialogue summary captures the main ideas and topics discussed in the conversation.
> Factuality (FAC): The accuracy and truthfulness of the information presented in the dialogue summary, reflecting fidelity to the original conversation.
> Consistency (CON): The degree to which the summary maintains logical and contextual coherence throughout, avoiding contradictory or conflicting information.
> Informativeness (INF): The richness and depth of information conveyed in the dialogue summary, including key details and relevant context.
> Coherence (COH): The overall clarity and organization of the summary, ensuring smooth transitions between ideas and coherence in the narrative flow.
> Relevance (REL): The pertinence of the information included in the dialogue summary to the intended purpose or topic, ensuring alignment with the user's interests or needs.
> Your output must be in Python dictionary format.

## F.3 LONGGUIDE'S PROMPTS

**Prompting templates for LongGuide.** Let $Q, C, I, D_f$ be the input query, context, instruction, and demonstration token sequence respectively (§1, §2), and $G^{best}$ is the learned guideline(s), the prompt for $\mathcal{M}$ is formatted: ``{I}\n{D_f}\n{C}\n{Q}\n{G^{best}}''.

| Models | Method | Summarization | | | Simplification | Translation | Dialogue Generation | Table2Text |
|---|---|---|---|---|---|---|---|---|
| | | SAMSum | CNN (3.0.0) | XL-Sum | SWiPE | IWSLT17 en-ja | Synthetic Persona | CommGen-Chall. |
| | #shots (random) | 3 | 3 | 5 | 3 | 5 | 5 | 5 |
| Mistral | #tokens consumed | 642 | 1110 | 811 | 1020 | 915 | 855 | 939 |
| | US$ consumed | 0 | 0 | 0 | 0 | 0 | 0 | 0 |
| ChatGPT | #tokens consumed | 1866 | 7683 | 4863 | 2380 | 1370 | 1344 | 1272 |
| | US$ consumed | insignificant | insignificant | insignificant | insignificant | insignificant | insignificant | insignificant |

Table 19: Total number of tokens consumed and US$ consumed for models to learn the metric guideline (MG) and output constraint guideline (OCG).

**Prompting costs.** Table 19 presents the total number of tokens consumed for models to learn the metric guidelines and output constraint guideline (OCG) for both models with the hyperparameters

of LongGuide specified in §E. We observe that the number of tokens needed to learn the guidelines is insignificant, demonstrating that LongGuide is a cost-effective solution and potentially beneficial for a wide range of applications.

Table 20 presents the prompting cost comparision between LongGuide and other PO algorithms. We compare the number of new prompts sampled by each algorithm for validation set verification, as these prompts are the primary cost bottleneck in PO algorithms. We observe that LongGuide is approximately at least **3.75** times cheaper than adv-ICL in both settings and **18.75** times cheaper than APO. For SAMSum, the validation of one prompt using 50 samples involves approximately 22K tokens, which incurs a cost of 0.02 USD as of November 19, 2024.

**Prompt for step 1, metric selection.**    Below is the prompt we use for step 1 selecting metrics for a given task.

> Select top-5 metrics that are the most important from the list below to evaluate a special way of {TASK_NAME}. {str(PRE_DEFINED_ASSESSMEN_METRICS)}.
> Here are some demonstrations of the task {TASK_NAME}: {DEMONSTRATION_STRING}.
> Output your list of metrics in Python list format without any explanation: [...].

**Prompt for step 2, metric score collection.**    Below is the prompt we use for step 2 for evaluating selected metrics on the task.

> You are given an input and an output of a {TASK_NAME} task.
> Input: {input}
> Output: {output}
> Your task is to evaluate the following criteria on a scale of 1-5, with 1 being worst and 5 being best.
> {EVALUATION_FORMAT}
> The definitions of the criteria are: {METRICS_DEFINITIONS}
> Your output must be in Python dictionary format without explanation.

**Prompt for step 2, collecting metrics' definitions.**    Below is the prompt we use for step 2 collecting METRICS_DEFINITIONS for step 2.

> Define the list of following metrics in details as the quality of the output expected for the {TASK_NAME} task.
> {metrics}
> Give me the list in bullet points.

**Prompt for step 3, generating metric guideline (MG).**    Below is the prompt we use for step 3, generating the metric guideline (MG).

> Now you are given the following metrics: {metrics_string} for the {TASK_NAME} task.
> Based on these scores on a scale of 5 for the quality of the output: {str(metrics_collected_scores)}, define the expected quality of the output for each metric in natural language. Give me the list in bullet points.

## G    EXAMPLES

| | Method | #Prompts Sampled | Cost |
|---|---|---|---|
| **ZS** | adv-ICL | (3 iterations) × (1 instruction) × (5 variants) | 15 × prompt validation cost |
| | APO | (5 iterations) × (15 prompts sampled) × (1 instruction) | 75 × prompt validation cost |
| | LongGuide | 4 prompts (MG, OCG, MG-OCG, No guideline) | **4** × prompt validation cost |
| **FS** | adv-ICL | (3 iterations) × (3 demonstrations + 1 instruction) × (5 variants) | 60 × prompt validation cost |
| | APO | (5 iterations) × (15 prompts sampled) × (3 demonstrations + 1 instruction) | 300 × prompt validation cost |
| | LongGuide | 4 prompts (MG, OCG, MG-OCG, No guideline) | **4** × prompt validation cost |

Table 20: Prompting cost comparison between PO methods and LongGuide based on # new prompts sampled to test over the validation set.

---

**Zero-shot + OCG** (ROUGE-L = 29.85)

Summarize the summary from the following dialogue.

Morgan: Hey gorgeous, how's your day?
Suzanne: Nothing special, it's just one of many boring days at work. But… better now though!
Morgan: Are you working at all? 😊
Suzanne: I'm trying 😊 But you aren't helping me, at all
Suzanne: I'm just taking a well-deserved break 😊
Morgan: I miss you Suzie
Suzanne: I miss you too Morgan
Morgan: Do you feel like going to a concert next week? Maroon 5 is playing at the Hulu Theater at Madison Square Garden.
Morgan: As it happens, I've got two tickets 😊
Morgan: Do you want to go?
Suzanne: Really? OMG! That's wonderful!
Suzanne: Thank you sweetheart!
Morgan: Oh, nothing. I just want you to be happy 😊

The summary must have from 1 to 4 sentences and from 5 to 51 words with an average of 22 words and 1 sentences.

*ChatGPT: Morgan and Suzanne are talking about their day at work. Morgan offers Suzanne tickets to a Maroon 5 concert, and Suzanne expresses excitement and thanks Morgan. They plan to attend the concert together.*

---

**Few-shot (3 shots) + MG-OCG** (ROUGE-L = 40.74)

Summarize the summary from the following dialogue.

Your generated summary must strictly fulfill the following task metrics
- Accuracy: The summary is completely and precisely capturing the essence and key points of the original dialogue with no omissions or errors.
- Clarity: The summary is clear, easy to understand, and free from ambiguity or confusion.
- Coherence: The summary maintains a logical and consistent flow of ideas and concepts, making it easy to follow.
- Completeness: The summary includes all essential information from the original dialogue without missing any critical details.
- Understandability: The summary is straightforward and accessible, making it comprehendible to a broad audience.
- Brevity: The summary is succinct and to the point, conveying the most important ideas using the least possible number of words.
- Relevance: The summary focuses on the main points and ideas of the original dialogue that are important and significant to the task or context.

The summary must have from 1 to 4 sentences and from 5 to 51 words with an average of 22 words and 1 sentences.

Input: [Demonstration 1 dialogue]
Output: [Demonstration 1 summary]

Input: [Demonstration 2 dialogue]
Output: [Demonstration 2 summary]

Input: [Demonstration 3 dialogue]
Output: [Demonstration 3 summary]

Input:
Morgan: Hey gorgeous, how's your day?
Suzanne: Nothing special, it's just one of many boring days at work. But… better now though!
Morgan: Are you working at all? 😊
Suzanne: I'm trying 😊 But you aren't helping me, at all
Suzanne: I'm just taking a well-deserved break 😊
Morgan: I miss you Suzie
Suzanne: I miss you too Morgan
Morgan: Do you feel like going to a concert next week? Maroon 5 is playing at the Hulu Theater at Madison Square Garden.
Morgan: As it happens, I've got two tickets 😊
Morgan: Do you want to go?
Suzanne: Really? OMG! That's wonderful!
Suzanne: Thank you sweetheart!
Morgan: Oh, nothing. I just want you to be happy 😊

Output:

*ChatGPT: Morgan invites Suzanne to a Maroon 5 concert at Madison Square Garden and she gladly accepts, expressing happiness and gratitude*

Figure 14: Full example of ChatGPT results on a SAMSum example (Gliwa et al., 2019) w/ LongGuide guidelines (Zero-shot + LongGuide and Few-shot + LongGuide) from Figure 1

**Ground-truth Answer (Summary)**

Rob and Eve will meet on Sunday morning to go to the shops. Eve has something to do at about 3

---

**Zero-shot (ROUGE-L = 30.77)**

Summarize the summary from the following dialogue.

Rob: Are we meeting up 2morrow?
Eve: How about Sunday
Eve: stores are open so we can go then
Rob: Ok I just don't know what time they close
Eve: we have ot go in the moring
Eve: I have some stuff to do around 3
Rob: ok

*ChatGPT*: *Rob and Eve discuss meeting up, with Eve suggesting they go on Sunday since stores are open. Rob agrees but is concerned about store closing times. Eve suggests going in the morning as she has plans in the afternoon. Rob agrees to the plan.*

---

**Zero-shot + MG  (ROUGE-L = 36.07)**

Summarize the summary from the following dialogue.

Rob: Are we meeting up 2morrow?
Eve: How about Sunday
Eve: stores are open so we can go then
Rob: Ok I just don't know what time they close
Eve: we have ot go in the moring
Eve: I have some stuff to do around 3
Rob: ok

Your generated summary must strictly fulfill the following task metrics
- Accuracy: The summary is completely and precisely capturing the essence and key points of the original dialogue with no omissions or errors.
- Clarity: The summary is clear, easy to understand, and free from ambiguity or confusion.
- Coherence: The summary maintains a logical and consistent flow of ideas and concepts, making it easy to follow.
- Completeness: The summary includes all essential information from the original dialogue without missing any critical details.
- Understandability: The summary is straightforward and accessible, making it comprehendible to a broad audience.
- Brevity: The summary is succinct and to the point, conveying the most important ideas using the least possible number of words.
- Relevance: The summary focuses on the main points and ideas of the original dialogue that are important and significant to the task or context.

*ChatGPT*: *Rob and Eve discuss meeting up, considering Sunday due to stores being open. They plan to go in the morning as Eve has plans later. Rob expresses uncertainty about store closing times, to which Eve suggests going in the morning.*

---

**Zero-shot + OCG  (ROUGE-L = 36.73)**

Summarize the summary from the following dialogue.

Rob: Are we meeting up 2morrow?
Eve: How about Sunday
Eve: stores are open so we can go then
Rob: Ok I just don't know what time they close
Eve: we have ot go in the moring
Eve: I have some stuff to do around 3
Rob: ok

The summary must have from 1 to 4 sentences and from 5 to 51 words with an average of 22 words and 1 sentences.

*ChatGPT*: *Rob and Eve plan to meet on Sunday when stores are open, but Rob is unsure about the closing time, so they decide to go in the morning.*

---

**Zero-shot + MG-OCG  (ROUGE-L = 44.00)**

Summarize the summary from the following dialogue.

Rob: Are we meeting up 2morrow?
Eve: How about Sunday
Eve: stores are open so we can go then
Rob: Ok I just don't know what time they close
Eve: we have ot go in the moring
Eve: I have some stuff to do around 3
Rob: ok

Your generated summary must strictly fulfill the following task metrics
- Accuracy: The summary is completely and precisely capturing the essence and key points of the original dialogue with no omissions or errors.
- Clarity: The summary is clear, easy to understand, and free from ambiguity or confusion.
- Coherence: The summary maintains a logical and consistent flow of ideas and concepts, making it easy to follow.
- Completeness: The summary includes all essential information from the original dialogue without missing any critical details.
- Understandability: The summary is straightforward and accessible, making it comprehendible to a broad audience.
- Brevity: The summary is succinct and to the point, conveying the most important ideas using the least possible number of words.
- Relevance: The summary focuses on the main points and ideas of the original dialogue that are important and significant to the task or context.

The summary must have from 1 to 4 sentences and from 5 to 51 words with an average of 22 words and 1 sentences.

*ChatGPT*: *Rob and Eve plan to meet on Sunday to take advantage of store hours; they aim to go in the morning, allowing Eve time for personal activities around 3.*

Figure 15: A sample from SAMSum dataset where MG and OCG supplement each other and are not inter-changeable to increase the performance in final answer.

**Ground-truth Answer (Simplified text)**

Dr. Yehuda Nir (31 March 1930 – 19 July 2014) was an Polish-born Israeli-American psychiatrist. He was known for surviving the holocaust. Nir served as a chief of child psychiatry at Memorial Sloan-Kettering Cancer Center from 1979 to 1986. He was born in Lvov, Poland. Nir died in Manhattan, New York, aged 84. **(53 tokens)**

---

**Zero-shot (ROUGE-L = 21.43)**

Simplify the following text

Yehuda Nir (March 31, 1930 – July 19, 2014) was a Polish-born American Holocaust survivor, psychiatrist and author of The Lost Childhood. Nir posed as a Roman Catholic and learned Latin to escape Nazi persecution in Poland during World War II. Nir's ordeal led him to a career as a psychiatrist, specializing in the treatment of post-traumatic stress disorder and severely ill children. He immigrated to the United in 1959 to complete medical residencies in New York City and Philadelphia. He served as the chief of child psychiatry of Memorial Sloan-Kettering Cancer Center from 1979 until 1986.
Nir was born Juliusz Gruenfeld in Lvov, Poland, (present-day Ukraine) on March 31, 1930. He later changed his name to "Nir" after World War II since "Gruenfeld" has German origins. Nir means plowed fields in Hebrew.
Nir released a memoir of his experience during the Holocaust, "The Lost Childhood" in 1989. A second edition was reprinted by Scholastic Press in 2002. The Lost Childhood is now used as part of the high school curriculum throughout the United States. He also published four self-help books focusing on relationships, such as "Not Quite Paradise: Making Marriage Work" and "Loving Men for All the Right Reasons."
Yehuda Nir died at his home in Manhattan, New York City, on July 19, 2014, at the age of 84. His funeral was held at Riverside Memorial Chapel on the Upper West Side with burial on the Mount of Olives in Jerusalem.

*ChatGPT: Yehuda Nir, born as Juliusz Gruenfeld in Lvov, Poland (now Ukraine) in 1930, survived the Holocaust by posing as a Roman Catholic and learning Latin. He later changed his name to "Nir" as a reference to plowed fields in Hebrew. Nir became a psychiatrist, focusing on treating post-traumatic stress disorder and severely ill children. He moved to the U.S. in 1959 to finish medical residencies in New York and Philadelphia. Nir served as the head of child psychiatry at Memorial Sloan-Kettering Cancer Center from 1979 to 1986.*
*Nir published a memoir of his Holocaust experiences, titled "The Lost Childhood," in 1989. The book has been included in high school curriculums in the U.S. since then, and a second edition was published by Scholastic Press in 2002. In addition, he authored four self-help books about relationships, such as "Not Quite Paradise: Making Marriage Work" and "Loving Men for All the Right Reasons."*
*Nir passed away on July 19, 2014, at the age of 84, in his Manhattan home. His funeral took place at Riverside Memorial Chapel on the Upper West Side, while he was buried on the Mount of Olives in Jerusalem.* **(192 tokens)**

---

**Zero-shot + MG  (ROUGE-L = 16.09)**

Simplify the following text

Yehuda Nir (March 31, 1930 – July 19, 2014) was a Polish-born American Holocaust survivor, psychiatrist and author of The Lost Childhood. Nir posed as a Roman Catholic and learned Latin to escape Nazi persecution in Poland during World War II. Nir's ordeal led him to a career as a psychiatrist, specializing in the treatment of post-traumatic stress disorder and severely ill children. He immigrated to the United in 1959 to complete medical residencies in New York City and Philadelphia. He served as the chief of child psychiatry of Memorial Sloan-Kettering Cancer Center from 1979 until 1986.
Nir was born Juliusz Gruenfeld in Lvov, Poland, (present-day Ukraine) on March 31, 1930. He later changed his name to "Nir" after World War II since "Gruenfeld" has German origins. Nir means plowed fields in Hebrew.
Nir released a memoir of his experience during the Holocaust, "The Lost Childhood" in 1989. A second edition was reprinted by Scholastic Press in 2002. The Lost Childhood is now used as part of the high school curriculum throughout the United States. He also published four self-help books focusing on relationships, including "Not Quite Paradise: Making Marriage Work" and "Loving Men for All the Right Reasons."
Yehuda Nir died at his home in Manhattan, New York City, on July 19, 2014, at the age of 84. His funeral was held at Riverside Memorial Chapel on the Upper West Side with burial on the Mount of Olives in Jerusalem.

The generated simplification must fulfill the following task metrics
- Accuracy: The text is expected to be highly accurate, with minimal errors or inaccuracies.
- Brevity: The text is expected to be concise and to the point, avoiding unnecessary repetition or verbosity.
- Clarity: The text is expected to be clear and easy to understand, conveying the intended message effectively.
- Coherence: The text is expected to be logically connected, with smooth transitions between ideas and concepts.
- Conciseness: The text is expected to be succinct and focused, without unnecessary elaboration or redundant information.
- Naturalness: The text is expected to read naturally, resembling fluent human language in terms of flow and expression.
- Readability: The text is expected to be easy to read and comprehend, with good overall readability.
- Simplicity: The text is expected to be simple and straightforward, catering to a broad audience and avoiding complexity.
- Understandability: The text is expected to be easily understood by the intended audience, with clear and unambiguous language.

*ChatGPT: Yehuda Nir, born Juliusz Gruenfeld in Poland in 1930, changed his name after World War II to the Hebrew term "Nir" meaning "plowed fields." Escaping Nazi persecution as a Roman Catholic with learned Latin, Nir became a psychiatrist specializing in PTSD and severely ill children. He immigrated to the US in 1959 for medical residencies in New York and Philadelphia. Nir published a memoir of his Holocaust experiences, "The Lost Childhood," in 1989 and a second edition in 2002. Now used in US high schools, this memoir highlights his harrowing journey. Additionally, he wrote self-help books on relationships, including "Not Quite Paradise" and "Loving Men." Nir died at 84 in Manhattan in 2014 and was interred in Jerusalem.* **(118 tokens)**

---

**Zero-shot + OCG (ROUGE-L = 15.79)**

Simplify the following text

Yehuda Nir (March 31, 1930 – July 19, 2014) was a Polish-born American Holocaust survivor, psychiatrist and author of The Lost Childhood. Nir posed as a Roman Catholic and learned Latin to escape Nazi persecution in Poland during World War II. Nir's ordeal led him to a career as a psychiatrist, specializing in the treatment of post-traumatic stress disorder and severely ill children. He immigrated to the United in 1959 to complete medical residencies in New York City and Philadelphia. He served as the chief of child psychiatry of Memorial Sloan-Kettering Cancer Center from 1979 until 1986.
Nir was born Juliusz Gruenfeld in Lvov, Poland, (present-day Ukraine) on March 31, 1930. He later changed his name to "Nir" after World War II since "Gruenfeld" has German origins. Nir means plowed fields in Hebrew.
Nir released a memoir of his experience during the Holocaust, "The Lost Childhood" in 1989. A second edition was reprinted by Scholastic Press in 2002. The Lost Childhood is now used as part of the high school curriculum throughout the United States. He also published four self-help books focusing on relationships, including "Not Quite Paradise: Making Marriage Work" and "Loving Men for All the Right Reasons."
Yehuda Nir died at his home in Manhattan, New York City, on July 19, 2014, at the age of 84. His funeral was held at Riverside Memorial Chapel on the Upper West Side with burial on the Mount of Olives in Jerusalem.

The simplified text must have from 1 to 17 sentences and from 16 to 426 words with an average of 80 words and 4 sentences.

*ChatGPT: Yehuda Nir, born Juliusz Gruenfeld in Poland in 1930, survived the Holocaust by posing as a Roman Catholic and learned Latin. He later became a psychiatrist, focusing on PTSD and severely ill children in the US. Nir published "The Lost Childhood" about his experiences and self-help books on relationships. He died in 2014, with a funeral in New York and burial in Jerusalem.*
*Born in Lvov, Poland, in 1930 as Juliusz Gruenfeld, Yehuda Nir posed as a Catholic during the Holocaust and used his knowledge of Latin to hide. After immigration to the US in 1959, he specialized in child psychiatry and PTSD treatment.*
*Nir published his memoir, "The Lost Childhood," in 1989 and it is now used in US high schools. Additionally, he wrote four self-help books, including "Not Quite Paradise" and "Loving Men for All the Right Reasons". Yehuda Nir, aged 84, passed away in Manhattan, New York, in 2014. His funeral was held at Riverside Memorial Chapel and he was buried on the Mount of Olives in Jerusalem.* **(171 tokens)**

---

**Zero-shot + MG-OCG (ROUGE-L = 27.82)**

Simplify the following text

Yehuda Nir (March 31, 1930 – July 19, 2014) was a Polish-born American Holocaust survivor, psychiatrist and author of The Lost Childhood. Nir posed as a Roman Catholic and learned Latin to escape Nazi persecution in Poland during World War II. Nir's ordeal led him to a career as a psychiatrist, specializing in the treatment of post-traumatic stress disorder and severely ill children. He immigrated to the United in 1959 to complete medical residencies in New York City and Philadelphia. He served as the chief of child psychiatry of Memorial Sloan-Kettering Cancer Center from 1979 until 1986.
Nir was born Juliusz Gruenfeld in Lvov, Poland, (present-day Ukraine) on March 31, 1930. He later changed his name to "Nir" after World War II since "Gruenfeld" has German origins. Nir means plowed fields in Hebrew.
Nir released a memoir of his experience during the Holocaust, "The Lost Childhood" in 1989. A second edition was reprinted by Scholastic Press in 2002. The Lost Childhood is now used as part of the high school curriculum throughout the United States. He also published four self-help books focusing on relationships, including "Not Quite Paradise: Making Marriage Work" and "Loving Men for All the Right Reasons."
Yehuda Nir died at his home in Manhattan, New York City, on July 19, 2014, at the age of 84. His funeral was held at Riverside Memorial Chapel on the Upper West Side with burial on the Mount of Olives in Jerusalem.

The generated simplification must fulfill the following task metrics
- Accuracy: The text is expected to be highly accurate, with minimal errors or inaccuracies.
- Brevity: The text is expected to be concise and to the point, avoiding unnecessary repetition or verbosity.
- Clarity: The text is expected to be clear and easy to understand, conveying the intended message effectively.
- Coherence: The text is expected to be logically connected, with smooth transitions between ideas and concepts.
- Conciseness: The text is expected to be succinct and focused, without unnecessary elaboration or redundant information.
- Naturalness: The text is expected to read naturally, resembling fluent human language in terms of flow and expression.
- Readability: The text is expected to be easy to read and comprehend, with good overall readability.
- Simplicity: The text is expected to be simple and straightforward, catering to a broad audience and avoiding complexity.
- Understandability: The text is expected to be easily understood by the intended audience, with clear and unambiguous language.

The simplified text must have from 1 to 17 sentences and from 16 to 426 words with an average of 80 words and 4 sentences

*ChatGPT: Dr. Yehuda Nir, born Juliusz Gruenfeld in 1930 in Lvov, Poland, survived the Holocaust by posing as a Catholic. He moved to the U.S. in 1959, becoming a psychiatrist specializing in PTSD and child psychiatry. His memoir, The Lost Childhood, published in 1989, is used in high schools. Nir died in Manhattan in 2014, with his burial in Jerusalem.* **(59 tokens)**

Figure 16: An example of SWiPE (Laban et al., 2023) where the record contains fewer tokens than the expected average. This reduces the effectiveness of OCG and MG individually, but their combination could enhance performance.

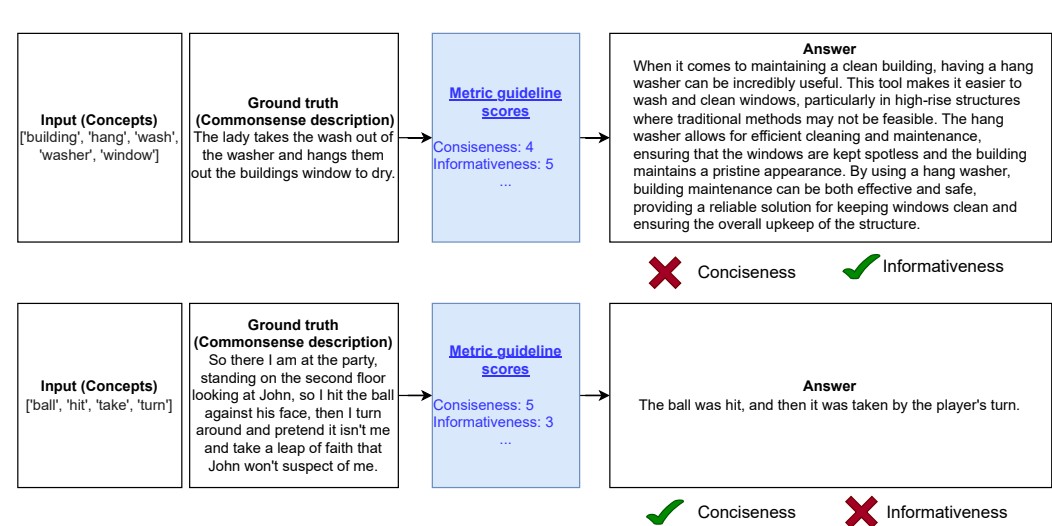

Figure 17: A CommonGen-Challenge example (Lin et al., 2020), where output with high Conciseness score could have low Informativeness score and vice versa

**Input (Dialogue)**
George: Hi Donna. I've been trying to catch you.
Donna: What about?
George: A rather delicate matter.
Donna: Did you catch AIDS?
George: Very funny!
Donna: It is, isn't it?
George: I don't think so.
Donna: Too bad. So what do you want?
George: Could we meet and discuss it somewhere?
Donna: Like where?
George: Like in a coffee shop or somewhere.
Donna: You sure a coffee shop is better to discuss delicate matters.
George: Come to think of it, you are right. It's not.
Donna: See? So what's up?
George: Couldn't you come to my place tonight?

**Ground truth answer (Summary)**
Donna will pay George a visit tonight to discuss a personal matter.

**Zero-shot answer (ROUGE-L = 16.22)**
George attempts to make arrangements to meet Donna to discuss a personal matter, but she is skeptical and makes a joke about the topic being AIDS. George finds her humor inappropriate and denies it. Donna presses for information on the discussion topic, and George suggests they meet at his place that night since a coffee shop may not be an appropriate setting.

**LongGuide (ROUGE-L = 21.62)**
(*MG metrics' scores*: Accuracy: 5, Brevity: 5, **Clarity: 4**, Correctness: 5, Relevance: 5, Understandability: 5, ***Informativeness: 4***)

George proposed discussing a private matter with Donna, suggesting a coffee shop but later acknowledging its unsuitability; he then invited her to his place instead.

**LongGuide without Step 2 (ROUGE-L = 17.24)**
(*MG metrics' scores w/o step 2*: Accuracy: 5, Brevity: 5, **Clarity: 5**, Correctness: 5, Relevance: 5, Understandability: 5, ***Informativeness: 5***)

George asked Donna to discuss a delicate matter with him, initially suggesting a coffee shop. However, Donna pointed out that a coffee shop might not be the best place for such a conversation, leading George to reconsider and propose coming to his place that night instead.

Figure 18: A SAMSum example, where skipping step 2 worsens the performance due to lack of clarity in metrics

**Input 1:**
Eddy: Audrey dear, it's about our meeting an Tuesday. Quite unexpectedly we'll be having guest in on that day, some important acquaintances of Marion's. Could we have our meeting on Monday? Otherwise on Wednesday?
Audrey: Oh I see. On Monday we're both in the gym in the afternoon and afterwards, you know, how one feels. And Wednesday we're going with the Meadows to see the Baltus exhibition in Riehen. How about Friday? Or weekend?
Eddy: I've already seen it. It is very much worth seeing! And don't believe the texts about teenage girls. Wait. I have to ask Marion about Friday.
Audrey: Or weekend.
Eddy: Just asked Marion. She thinks Friday will be perfect.
Audrey: The same place the same time?
Eddy: I should think so.
Audrey: Tat ta. Till Friday!
Eddy: Marion says on Fridays they're open as early as 4 pm. So one hours earlier?
Audrey: Would be great but I might come late. Am in the office till 4.
Eddy: How long do you take to cycle to Terra del Fuego?
Audrey: 10 mins?
Eddy: That's nothing! So see you in Terra d. F. at around 4 pm on Friday.
Audrey: Ta ta!
**Output 1**: Audrey, Eddy and Marion will meet at Terra del Fuego around 4 pm on Friday. Eddy has seen the Baltus exhibition in Riehen and thinks it is worth seeing. Audrey is planning to see the Balthus exhibition on Wednesday with the Meadows. *[clarity: 5, #sentence: 3]*

**Input 2:**
Beatrix: hi Carol, did someone get a flu with fever and dry cough ?
Carol: yes, Erik and Mark were ill this week. But it wasn't influenza, luckily
Beatrix: did you give them paracetamol?
Carol: yes of course, every 4 hours
Beatrix: and that's all?
Carol: no you have to drink a lot, tea and honey is good
Beatrix: Sarah doesn't like honey.
Carol: you may give her tea and lemon, that's good too for the throat
Beatrix: still i'm worried because the fever is high
Carol: try to alternate paracetamol and ibuprofen, every 4 hours
Beatrix: are you sure i can?
Carol: yes do that for 24 hours, if she's not better tomorrow, you may call your doc.
Beatrix: that's the point, he's on vacation till next week.
Carol: i'm sure Sarah will recover soon
Beatrix: hope so, i have a big meeting in two days in Dublin , i really have to go.
Carol: good luck and let me know if you need some help
Beatrix: sure, thanks
**Output 2**: Sarah has a flu with fever and dry cough, so Beatrix asks Carol for advice on what treatment to use. Erik and Mark were also ill last week and Carol gave them paracetamol and a lot of tea and honey. Beatrix's doctor is on vacation till next week and Beatrix has a big meeting in Dublin in two days. *[clarity: 5, #sentence: 3]*

**Input 3**:
Tiffany: Who's Elon Musk?
Blair: You know nothin', John Snow.
Dale: Srsly? Never heard of him?
Tiffany: Nope, sorry.
Dale: He set up PayPal, Tesla and SpaceX.
Blair: He wants to go to the moon. He's creating the first passenger shuttle to the moon.
Tiffany: Rly? :) How much is the ticket?
Dale: Depends. Currently around $ 35 mil.
Tiffany: What?
Blair: Rly. Unfortunately, that kinda cash doesn't fit into my piggy bank.
Dale: Bt the good news is that you'll be able to watch it using VR!
Tiffany: How?!
Blair: Apparently they're going to stream the whole thing in HD via their satellites.
Tiffany: Wow! Amazing! When?
Dale: No one really knows. When they finally build the shuttle and set off, but the date is not known for now.
Tiffany: Shame. Maybe until then I'll manage to buy myself this VR headset ;)
Blair: You really should! If not for work, you can always use it for pleasure :)
Dale: Just don't play horror games. You'll get a heart attack.
Tiffany: Really that scary?
Dale: Tried it once. Nevermind the money. I'm not doing it again.
Blair: Someone got scared? ;)
Dale: But all other games are fine. And the experiences are unspeakable!
Blair: I do confirm that. When you put on the headset it's a little awkward at first, but when you forget about it, virtual reality becomes reality.
Tiffany: Where can I buy this?
Dale: The easiest way? Online.
Blair: Remeber, it's not cheap.
Dale: <file_other> <file_other> <file_other> <file_other> <file_other> There you go.
Tiffany: Cheap is not the word that best describes it.
Blair: But it's worth it ;)
**Output 3**: Elon Musk set up PayPal, Tesla and SpaceX and is creating the first passenger shuttle to the moon. The tickets cost around $ 35 million, but the event is going to be streamed in HD via satelites. The date is yet unknown. Tiffany wants to buy a VR headset. *[clarity: 5, #sentence: 3]*

**Input:**
Morgan: Hey gorgeous, how's your day?
Suzanne: Nothing special, it's just one of many boring days at work. But… better now though!
Morgan: Are you working at all? 😊
Suzanne: I'm trying 😊 But you aren't helping me, at all
Suzanne: I'm just taking a well-deserved break 😊
Morgan: I miss you Suzie
Suzanne: I miss you too Morgan
Morgan: Do you feel like going to a concert next week? Maroon 5 is playing at the Hulu Theater at Madison Square Garden.
Morgan: As it happens, I've got two tickets 😊 Morgan: Do you want to go?
Suzanne: Really? OMG! That's wonderful!
Suzanne: Thank you sweetheart!
Morgan: Oh, nothing. I just want you to be happy

**Output:** Morgan and Suzanne discuss their day, with Morgan inviting Suzanne to a Maroon 5 concert. Suzanne excitedly accepts, and Morgan expresses his desire for her happiness. *[clarity: 4, #sentence: 2]*

Figure 19: Full text for an example in §2.

