# OpenReview forum: "Beyond In-Context Learning: Enhancing Long-form Generation of Large Language Models via Task-Inherent Attribute Guidelines"
_ICLR.cc/2025/Conference — Submitted to ICLR 2025_

### Official Review · Reviewer_yZpL · 2024-10-28

**Soundness:** 4
**Presentation:** 4
**Contribution:** 3
**Rating:** 6
**Confidence:** 4

**Summary:**

The paper proposes a study on LLMs' generation quality using in-context learning showing its ineffectiveness on long-context tasks, and proposed a new technique, LongGuide, to alleviate the problem.
LongGuide is an algorithm to generate customized guidelines for the LLM to optimize a set of imposed self-evaluated metrics. Overall, LongGuide collects a set of task-independent metrics, obtains the verbal descriptions of metric values via LLM self-evaluation and combines them with constraint-based guidelines which instruct the model on the numerical properties such as token/sentence count.

The technique is evaluated on a set of long-form generation tasks (summarization, text simplification, machine translation, generation) against SoTA prompt optimization algorithms such as APO and adv-ICL.
The experiments show that LongGuide improves LLMs' performance across the board, working both for medium-sized models (Mistral-7B-it) and large ones (ChatGPT 3.5 Turbo) - as shown via automatic metrics (BLEU, ROUGE-L) and human evaluation.

**Strengths:**

* a new method for automatically finding a set of LLM guidelines that improve longform generation is proposed
* it outperforms prompt optimization SoTA algorithms on a series of benchmarks, both for medium-sized and large models
* in addition to introducing the algorithm, the authors conduct an extensive study on how in-context learning is ineffective for longform generation

**Weaknesses:**

* metrics used as the core set of LongGuide are described insufficiently - the main thing explained about them is that they do not include LLM-based ones which sounds worrying since those are proved to have correlation with human judgements, at least for summarization tasks they're superior to the numeric ones like ROUGE-L
* with a combination of metric guidelines and output constraint guidelines evaluated at the last step of LongGuide, there arises the question on performance/cost aspects of LongGuide and how it compares to prompt optimization SoTA - I couldn't find it in the main paper content

**Questions:**

l. 208: "...and propose 12more metrics for a broader evaluation coverage" - where are they described?

---

> ### Author Response · Authors · 2024-11-20
> **Response to reviewer**
>
> Dear reviewer yZpL,
>
> We deeply thank you for your time and efforts in providing constructive reviews for our paper. We would like to address your concerns below and our updated changes in the paper are in blue.
>
> > metrics used as the core set of LongGuide are described insufficiently - the main thing explained about them is that they do not include LLM-based ones which sounds worrying since those are proved to have correlation with human judgements, at least for summarization tasks they're superior to the numeric ones like ROUGE-L.
>
> Thank you for your suggestion. We have added GPT-4o-Judge scores evaluating how aligned the generated answer is with the reference answer and its quality on criteria:
>
> - Format consistency: ensuring the generated response matches the length, structure and layout of the reference.
> - Content completeness: evaluating whether all key points present in the reference are included in the assistant's answer.
> - Factuality: checking for factual correctness of the assistant's answer.
> - Style adherence: ensuring that the tone, style, and level of detail of the assistant's answer match the reference.
> - Assistant's answer quality: assessing how well the response satisfies the user's requirements.
>
> Each criterion is scored on a scale of 10, and the final GPT-4o-Judge score is the average of them. We have included the evaluation scores in Table 2 and Figure 2. We summarize the results below:
>
> | Method | Format | Content | Factuality | Style | Quality |
> | -------- | ------- | ------- | ------- | ------- | ------- |
> | Baseline | 4.18 | 4.83 | 6.64 | 4.36 | 4.75 |
> | + APO  | 4.73 | 5.91 | 7.26 | 4.91| 5.39 |
> | + LongGuide | **5.72** | **6.01** | **8.25** | **5.78** | **6.04** |
>
> Among five GPT-4o-Judge criteria in Figure 2, LongGuide notably improves Format, Style, and Factuality, confirming its effectiveness in aligning model generation with ground-truth distributions. In addition, the significant gains in Quality criterion, together with the ROUGE-L scores from Table 2 further demonstrate that LongGuide also significantly enhances the generation quality.
>
> Our evaluation prompting template is heavily inspired by (https://openreview.net/forum?id=uccHPGDlao).
>
> > with a combination of metric guidelines and output constraint guidelines evaluated at the last step of LongGuide, there arises the question on performance/cost aspects of LongGuide and how it compares to prompt optimization SoTA - I couldn't find it in the main paper content
>
> Thank you for the constructive feedback. The prompting costs for generating guidelines we provided in Appendix F.3. Below we present the prompting costs for the last step of LongGuide compared to adv-ICL and APO on SAMSum using 3 demonstrations:
>
> | | Method | #Prompts Sampled | Cost |
>  |---------|------------------|------------------------------------------------------------|-------------------------------|
> | **ZS** | adv-ICL | (3 iterations) x (1 instruction) x (5 variants) | 15 x prompt validation cost |
> | | APO | (5 iterations) x (15 prompts sampled) x (1 instruction) | 75 x prompt validation cost |
> | | LongGuide | 4 prompts (MG, OCG, MG-OCG, No guideline) | **4** x prompt validation cost |
> | **FS** | adv-ICL | (3 iterations) x (3 demonstrations + 1 instruction) x (5 variants) | 60 x prompt validation cost |
> | | APO | (5 iterations) x (15 prompts sampled) x (3 demonstrations + 1 instruction) | 300 x prompt validation cost |
> | | LongGuide | 4 prompts (MG, OCG, MG-OCG, No guideline)  | **4** x prompt validation cost |
>
> LongGuide is approximately at least **3.75** times cheaper than PO algorithms in terms as it requires only four prompt variants to verify on the validation set. For SAMSum, the validation of one prompt using 50 samples involves approximately 22K tokens, which incurs a cost of $0.02 USD as of November 19, 2024.
>
> We have added these analyses in Appendix F.3. We have also added a sentence discussing the cost-efficiency of LongGuide in the Introduction L091-092.
>
> > l. 208: "...and propose 12more metrics for a broader evaluation coverage" - where are they described?
>
> They are described in Table 11 as we noted in L198: (Appx.-Table 11 for details).
>
> ## In summary
>
> We thank you for your time and constructive feedback. We hope our responses can sufficiently address your concern and improve your ratings. Thank you for your consideration.

---

> ### Comment · Reviewer_yZpL · 2024-11-25
> **Increased soundness upon rebuttal**
>
> Thanks to the authors for the comprehensive additions to the paper - I increased the Soundness rating by one point.

---

> ### Author Response · Authors · 2024-11-26
> **Thank you for reviewing our paper**
>
> Thank you reviewer yZpL for your feedback!

---

### Official Review · Reviewer_YJ28 · 2024-10-30

**Soundness:** 2
**Presentation:** 3
**Contribution:** 2
**Rating:** 6
**Confidence:** 3

**Summary:**

The paper introduces LongGuide, to efficiently generate two parallel streams of guidelines capturing task language and format properties. All the experiments are conducted on finetuned LLMs (Mistral/ ChatGPT), which is a major concern without isolating ICL capacity from Instruction following capacities. I don't get the point of mentioning In-Context Learning in the name.

I like the theoretical derivations given in Section 2.1. But it is not solid. I don't buy in that 4.1 is a strong proof of Hypothesis 2.1.

Specifically, the discussion on the weights of different objectives are pretty weak. Manual preference on different aspects of generated long-context are actually always highly unbalanced based on queries and contents. The theory basically ignores the unbalanced and dynamic weights on different objectives and simply treats them the same.

The metrics and datsets used for evaluating LongGuide are pretty weak.

Additionally, the work is pretty much only a prompt work with little solid science contribution. The prompt workflow itself is also only based on weak hypothesis and cannot intuitionally matches the manual work pattern. For manual workflow, I refer to human writer's working pattern. Human writer doesn't write based on a given metrics collection and keeps reviewing it. Intuitionally it is not solid for me. Take novel writing as an example, as a reader, sometimes I pay more attention to whether the writing is good, sometimes I pay more attention to storyline. I don't always assign 0.9 weight to storyline and 0.1 weight to writing.

**Strengths:**

1. Good writing.
2. Motivation is clear.
3. Their ablation studies on each elements of their method in the appendix are detailed and clear.

**Weaknesses:**

1. For the proof in Section 2.1, it is well-written but only a descriptive math. Remark 2.1 and Definition 2.1 only have weak relationship with hypothesis 2.1. And Hypothesis 2.1 only claims a simple thing: Task T can be optimized by optimizing several understandable aspects of the task, e.g. fluency, factuality, and etc. Noted that each aspect is with a fixed L in their setting. They don't give a solid proof for that. Intuitionally it is not solid for me. Take novel writing as an example, as a reader, sometimes I pay more attention to whether the writing is good, sometimes I pay more attention to storyline. I don't always assign 0.9 weight to storyline and 0.1 weight to writing.

2. The metrics reported in Section 4.1 is pretty weak, with only BLEU-1/ ROUGE-L given and without more clear and specific evaluation aspects related to human, like fluency, factuality, and etc. If human evaluation is not accessible, at least LLM-based evaluation should be given. Although they provide BERTScore, but it is actually similar to BLEU-1/ ROUGE-L on the evaluation aspects. It is mainly based on similarity. They should move more detailed analysis or some estimations of the manual evaluation in the appendix to the main context.

3. There is little direct takeaway from the paper. By direct takeaway, I mean that engineers and researchers can directly adopt the hyper-parameters and models given from a paper to their academic and industrial pipeline. The experiments conducted in the paper only cover SAMSum/ CNN/ SWiPE in the main text, which are not comprehensive and challenging at least for nowadays research.

**Questions:**

I don't get the point of mentioning ICL in the paper's name. Because they only adapt instruction tuned LLMs in their research. And a lot of existing research points that there is a trade-off between the instruction following capacity and ICL. Maybe more experiments on base models are needed.

---

> ### Author Response · Authors · 2024-11-20
> **Response to reviewer (1)**
>
> Dear reviewer YJ28,
>
> We deeply thank you for your time and efforts in providing constructive reviews for our paper. We would like to address your concerns below and our updated changes in the paper are in blue.
>
> > ...Specifically, the discussion on the weights of different objectives are pretty weak. Intuitionally it is not solid for me...I don't always assign 0.9 weight to storyline and 0.1 weight to writing.
>
> Thank you for your comment. We have revised Hypothesis 2.1 and Proof of Remark 2.2 to address the weighting of different objectives.
>
> We agree that human preferences can vary dynamically, with different temporal weights assigned to objectives based on context. However, modelling those is highly complex: accurately determining such weighting parameters typically requires careful empirical experiments or expert judgment.
>
> In this work, we have tried our best by (1) selecting the most important metrics for capturing task properties and (2) incorporating the specific levels of these metrics from the training data. Extending our work to model dynamic objective weights is a valuable direction for future work. We will add this discussion into the Generalization section of our paper.
>
> > The metrics reported in Section 4.1 is pretty weak, with only BLEU-1/ ROUGE-L given and without more clear and specific evaluation aspects related to human, like fluency, factuality, and etc....
>
> Thank you for your suggestion. We have added GPT-4o-Judge scores (Section 4) evaluating how aligned the generated answer is with the reference answer and its quality on criteria:
>
> - Format consistency: ensuring the generated response matches the length, structure and layout of the reference.
> - Content completeness: evaluating whether all key points present in the reference are included in the assistant's answer.
> - Factuality: checking for factual correctness of the assistant's answer.
> - Style adherence: ensuring that the tone, style, and level of detail of the assistant's answer match the reference.
> - Assistant's answer quality: assessing how well the response satisfies the user's requirements.
>
> Each criterion is scored on a scale of 10, and the final GPT-4o-Judge score is the average of them. We have included the evaluation scores in Table 2 and Figure 2. We summarize the results below:
>
> | Method | Format | Content | Factuality | Style | Quality |
> | -------- | ------- | ------- | ------- | ------- | ------- |
> | Baseline | 4.18 | 4.83 | 6.64 | 4.36 | 4.75 |
> | + APO  | 4.73 | 5.91 | 7.26 | 4.91| 5.39 |
> | + LongGuide | **5.72** | **6.01** | **8.25** | **5.78** | **6.04** |
>
> Among five GPT-4o-Judge criteria in Figure 2, LongGuide notably improves Format, Style, and Factuality, confirming its effectiveness in aligning model generation with ground-truth distributions. In addition, the significant gains in Quality criterion, together with the ROUGE-L scores from Table 2 further demonstrate that LongGuide also significantly enhances the generation quality.
>
> Our evaluation prompting template is heavily inspired by (https://openreview.net/forum?id=uccHPGDlao).
>
> > The experiments conducted in the paper only cover SAMSum/ CNN/ SWiPE in the main text, which are not comprehensive and challenging at least for nowadays research.
>
> Thank you for your comment. We conduct our main experiments across 7 diverse generation tasks, including summarization, text simplification, translation, dialogue generation, and table-to-text generation, see Table 2.

---

> > ### Author Response · Authors · 2024-11-20
> > **Response to reviewer (2)**
> >
> > > Additionally, the work is pretty much only a prompt work with little solid science contribution.
> >
> > Thank you for your feedback. While we respect your perspective, we would like to address your concerns and clarify the significance of our work.
> >
> > Although your comment contrasts with most of the other reviewers (3 out of 5 rated our contribution 3/5 and the other reviewer thought our work was not very novel so they gave 2), we understand that the value of prompt-based research may not be universally appreciated. However, it is important to recognize that prompting plays a crucial role in practical applications, particularly in business and real-world settings, where traditional benchmarks often do not align with user-centric outcomes. In these contexts, prompting to obtain optimized model performance is of paramount importance.
> >
> > Our work extends beyond mere prompting; it tackles the critical challenge of aligning LLM generation distribution with downstream task distribution via prompting, as highlighted in lines L086-087. While numerous fine-tuning methods exist to address this, there is a notable gap in research on non-fine-tuning approaches for LLMs, such as prompting and calibration. These methods are often more accessible and scalable for a wider audience compared to traditional fine-tuning, as fine-tuning LLMs to be successful and reliable can be impractical for many researchers, engineers, and institutions.
> >
> > We believe that solving LLM adaptation for long-form generation tasks through prompting represents a meaningful scientific contribution, as hopefully, you can agree with us. Our approach addresses a real-world problem and provides a practical solution that is both novel and widely applicable. We hope this clarification helps convey the value and importance of our work.
> >
> > > There is little direct takeaway from the paper. By direct takeaway, I mean that engineers and researchers can directly adopt the hyper-parameters and models given from a paper to their academic and industrial pipeline.
> >
> > Thank you for your comment. We appreciate your perspective and would like to highlight the key contributions of our paper and their practical implications:
> > - (C1) We identify a critical challenge: the misalignment between LLM generation and the distributions required for downstream long-form generation tasks. We demonstrate both empirical and theoretical intuitions that ICL demonstrations alone are insufficient to teach LLMs the task-specific language and format distributions (L016-019). This finding is meaningful, as ICL is currently the most widely used instructional method for adapting LLMs (L034-036).
> > - (C2) We propose LongGuide, an efficient guideline-learning algorithm designed to improve the distribution alignment of LLMs for downstream tasks. This method significantly addresses this fundamental challenge.
> > - (C3) We provide an in-depth analysis of LongGuide, revealing key insights into its properties and why it works: it can be used by weaker models to enhance stronger models, it boosts the performance of non-instruct models, it significantly improves ICL performance, and it integrates effectively with automatic prompt optimizers.
> >
> > We believe that engineers and researchers can takeaways findings from (C1), (C2), (C3), as hopefully you can agree with us. The framework we present is also highly efficient, highly adaptable, and generalizable and can be directly applied to any long-form generation task in both academic and industrial pipelines.
> >
> > > I don't get the point of mentioning ICL in the paper's name. Because they only adapt instruction tuned LLMs in their research. And a lot of existing research points that there is a trade-o" between the instruction following capacity and ICL. Maybe more experiments on base models are needed.
> >
> > Thank you for your comment. We actually experimented with one non-instruction-tuned model, Mistral-7B-v0.1 in Appendix C.1.  The results show that LongGuide improves more than half of the experiments, showing its potential effectiveness in enhancing even non-instruct models.

---

> ### Author Response · Authors · 2024-11-20
> **Response to reviewer (3)**
>
> > For the proof in Section 2.1, it is well-written but only a descriptive math. Remark 2.1 and Definition 2.1 only have weak relationship with hypothesis 2.1. And Hypothesis 2.1 only claims a simple thing: Task T can be optimized by optimizing several understandable aspects of the task, e.g. fluency, factuality, and etc. They don't give a solid proof for that.
>
> Thank you for your feedback. We would like to clarify that the purpose of this subsection is not to present a “rigorous” theory, but rather to provide an intuition that motivates our approach. We believe that a “solid” theoretical discussion should deserve the whole paper discussing details to address the epsilon distribution recovery. To address your concern, we have made the following modifications:
>
> - We move Subsection 2.1 into Appendix A, and we add a short paragraph theoretical intuitions in Section 2.
> - We change “Theoretical Derivations” to “Theoretical Intuitions”.
> - We shorten the definition of task property significantly for more conciseness.
> - We remove the Remark 2.2 as you suggested. We describe it in L180-181 and put the old Remark 2.2 into Appendix A.
>
> We hope these changes improve the clarity and coherence of the section. We also try our best to balance the opinions among reviewers. If you have further suggestions, we would be happy to consider them.
>
> ## In summary
>
> We thank you for your time and constructive feedback. We hope our responses can sufficiently address your concern and improve your ratings. Thank you for your consideration.

---

> ### Author Response · Authors · 2024-11-23
> **Request for your review**
>
> Dear Reviewer YJ28,
>
> As the author-reviewer discussion period is nearing its conclusion, we kindly request your consideration of our responses to your concerns.
>
> We deeply thank you for your time and careful review. Many of the comments in your review we have actually addressed in the revised paper, as hopefully you can agree as mentioned above. We thank you for taking the concern to provide very detailed critique of the paper, and trust that you find our clarifications appropriate and worthy of a higher score.
>
> Thank you for your attention and consideration.
>
> Best regards, The Authors

---

> ### Comment · Reviewer_YJ28 · 2024-11-26
>
> Thanks for the detailed response provided by the authors. However, I'd like to respectfully retain my score. Because the rebuttal doesn't address several key problems of the paper. :
>
> 1. One non-instruction-tuned model, Mistral-7B-v0.1 is definitely not enough for an ICL-related paper. You should provide a wide range of sota foundation models to verify your methods' efficiency. For Open-source models, you can select Qwen-2.5 / Yi / LLaMA-3.1 / Mistral-v0.3 / etc. For fully transparent models, you can select OLMo / MAP-Neo / Pythia/ etc. Honestly speaking, although I fully understand the LLM verision updating is crazy. Mistral-v0.1-7B is still too outdated to be convincing for a paper in the late 2024.
> 2. I still recommend you to include more diverse NLG tasks and benchmarks published after 2023. For the tasks in the paper, there is a really high possibility that they are included in the pretrain corpus and not convincing at all. The rebuttal about that is pretty weak.
> 3. For the claimed scientific contribution, a more detailed statistic analysis between the real downstream [i.e. from the true user, like what have been done in Chatbot Arena / WildBench] and the provided aspects should be deeply studied. Instead of a prompt engineering paper, a worse thing is overclaiming. The experiment results in the paper do not solidly solve LLM adaptation for long-form generation tasks or clearly reveal the relationship between different downstreams and the provided fine-grained metrics. If the detailed analysis can be provided, the analysis itself can be a very good paper then.

---

> ### Author Response · Authors · 2024-11-28
> **Thank you for your feedback (4)**
>
> Dear reviewer YJ28,
>
> Thank you for engaging with us in this conversation and your feedback. We appreciate the time and effort you have invested in evaluating our paper. We would like to address the new points you raised:
>
> > 1. One non-instruction-tuned model, Mistral-7B-v0.1 is definitely not enough for an ICL-related paper. You should provide a wide range of sota foundation models to verify your methods' efficiency...
>
> Thank you for sharing your thoughts. To the best of our knowledge, the ICL concept applies broadly to all language models (Brown et al., 2020; Dong et al., 2023). Our study focuses on instruction-tuned models. As noted in our Acknowledgement (L540-543), LongGuide requires models that *possess strong instruction-following capabilities and a certain level of task knowledge*. These attributes are essential for enabling self-evaluation and leveraging task-specific guidelines effectively. Models that are not instruction-tuned, such as Mistral-7B-v0.1, were included to demonstrate baseline capabilities only and they are not our primary focus since they can’t follow our method’s instructions to perform self-evaluation.
>
> *Perhaps, the reviewer meant our study of the limitations of ICL (Section 2) should cover more non-instruct LLMs?* For this perspective, we have added the experiments in Section 2 with 3 non-instruct models (Mistral-7B-v0.3, Llama-3.1-8B, Qwen2.5-7B) + 1 instruct-model (Llama-3.1-8B-it). The results are presented below and we supplemented them in Section 2.
>
> | **ICL w/ 5 demos**    | **(1) COV** | **(2) FAC**    | **(3) CON** | **(4) INF**    | **(5) COH** | **(6) REL**    | **(7) NT (mean)** | **(7) NT (std)**    |
> | -------- | ------- | -------- | ------- | -------- | ------- | -------- | ------- | -------- |
> | *Expected*    | *100*       | *100*       | *100*       | *100*       | *100*       | *100*       | *17.00*           | *0.00*            |
> | **Mistral-7B-v0.3**   | 12          | 27          | 28          | 8           | 20          | 35          | 87.74             | 144.91            |
> | **Llama-3.1-8B**      | 12          | 42          | 50          | 4           | 32          | 47          | 271.81            | 379.48            |
> | **Qwen2.5-7B**        | 43          | **90**      | **85**      | **40**      | 78          | **96**      | 281.38            | 264.59            |
> | **Mistral-7B-it-v0.2**| 38          | 80          | 78          | 17          | 75          | 88          | 50.25             | 55.54             |
> | **Llama-3.1-8B-it**   | **44**      | 86          | 82          | 26          | **81**      | 87          | **34.72**         | **45.29**         |
>
> We find almost the same observations as we had in Section 2: (i) ICL models do not achieve a 100% score of 5 on any metric; (ii) increasing # demonstrations does not rectify this issue; (iii) adding a simple guideline improves instruct models. Additionally, Qwen scored high on metrics (1)–(6) while failed on metric (7) (and (8)) because it copied the input dialogue as the summarization outcome and thus did not solve the task properly.
>
> We have added the above experiments to Section 2. We believe that exploring the extension of our work to non-instruct model adaptation is a promising direction for future work.

---

> ### Author Response · Authors · 2024-11-28
> **Thank you for your feedback, we have added AlpacaEval2 and been waiting for WildBench-V2 (5)**
>
> > 2. I still recommend you to include more diverse NLG tasks and benchmarks published after 2023. For the tasks in the paper, there is a really high possibility that they are included in the pretrain corpus and not convincing at all. The rebuttal about that is pretty weak.
>
> Thanks for this new feedback. We understand your concern about the data contamination of LLMs. We have added and summarized our AlpacaEval2 evaluations and been waiting for WildBench-V2 evaluations from AI2. We would like to clarify:
>
> - *Our selected tasks are widely used by the community:* The tasks and benchmarks used in our evaluation are widely adopted for assessing the generation capabilities of LLMs ([Jang et al., NeurIPS 2024](https://arxiv.org/pdf/2411.06710v1),
> [Feng et al., EMNLP 2024](https://aclanthology.org/2024.findings-emnlp.648.pdf),
> [Bai et al., ACL 2024](https://aclanthology.org/2024.acl-long.172.pdf)) similar to GSM8K/SVAMP for reasoning. These benchmarks are suitable for our method, as our method requires a small number of training samples available.
>
> - *The benchmarks are valuable as they challenge and expose LLM weaknesses*: The benchmarks are challenging and expose LLM weaknesses: As shown in Table 3 and Figure 5, none of the tested models scored above 8 on GPT-4o-Judge or surpassed 50% ROUGE-L against ground truth. Average quality and format ratings for answers remained below 5/10, highlighting LLM limitations.
>
> - *We prioritize selecting widely used test sets and their latest versions for evaluation:*
>    - CNN: latest version 3.0.0
>    - SWiPE: published May 2023
>    - Synthetic-Persona-Chat: released 2024
>    - CommonGen-Challenge: challenge test set of CommonGen
>    - SAMSum, XL-Sum, and IWSLT: widely cited and used in summarization and translation studies
>
> While we agree with you that we cannot control whether these test sets were used to pretrain LLMs, similar to the GSM8K/SVAMP benchmarks, our selected benchmarks remain reasonable and valuable for studying capabilities and weaknesses in LLMs, as hopefully, you can agree with us.
>
> We have also added an experiment with subsets of **AlpacaEval2 (Yann et al., 2024)** and been waiting for **WildBench-V2 (Lin et al., 2024)**. Due to the very limited resources and time constraints, our experiments are conducted on 203 random samples of AlpacaEval2 and 200 samples of WildBench with ChatGPT (gpt-3.5-turbo-1106). Since these benchmarks do not have training data, our setups are:
> - For AlpacaEval2, we train LongGuide on *only 5 random samples* from [alpaca-gpt4](https://huggingface.co/datasets/vicgalle/alpaca-gpt4). We also use those 5 samples as few-shot demonstrations. Note that [alpaca-gpt4](https://huggingface.co/datasets/vicgalle/alpaca-gpt4) is a quite OOD dataset compared to AlpacaEval2.
> - For WildBench, we train LongGuide on *only 5 random samples* from [WildBench-V2 GPT-4 outputs](https://huggingface.co/datasets/allenai/WildBench-V2-Model-Outputs/viewer/gpt-4-turbo-2024-04-09). We also use those 5 samples as few-shot demonstrations *and exclude them from our evaluation samples*.
>
> The results with AlpacaEval2 are summarized below.
>
> | Setting           | LC Win Rate | Win Rate |
> |-------------------|-------------|----------|
> | ZS               | 11.08%      | 3.17%    |
> | ZS + OCG         | 4.73%       | 2.44%    |
> | ZS + MG          | **19.13%**      | **7.07%**    |
> | ZS + MG-OCG      | 8.42%       | 3.90%    |
> | **ZS + LongGuide**   | **19.13%**      | **7.07%**    |
> |-------------------|-------------|----------|
> | FS               | 8.08%       | 2.68%    |
> | FS + MG          | **12.65%**      | **4.88%**    |
> | FS + OCG         | 7.73%       | 3.45%    |
> | FS + MG-OCG      | 12.63%      | 4.88%    |
> | **FS + LongGuide**     | **12.65%**      | **4.88%**    |
>
> With only 5 samples from [alpaca-gpt4](https://huggingface.co/datasets/vicgalle/alpaca-gpt4), LongGuide significantly improves ChatGPT on AlpacaEval2. OCG did not achieve good results because [alpaca-gpt4](https://huggingface.co/datasets/vicgalle/alpaca-gpt4) is a quite OOD dataset compared to AlpacaEval2.
>
> For WildBench, we are waiting for AI2 to get the results and *we are unsure if we can get the results to supplement here within the rebuttal period, but we will do so if we can*.
>
> Feel free to share your feedback, we are happy to take and discuss it.

---

> > ### Comment · Reviewer_YJ28 · 2024-11-28
> >
> > I will increase my score to 6. But I would still recomment you to:
> > 1. Seriously reconsider the name of including ICL, because it does not makes any sense. Since the paper does not seriously evaluate the role of ICL capacities in the Long-form Generation and your method also has little relationship with that. In context learning is an intrinsic capacity of LLM and the paper shows nothing beyond that. Besides that, for a strong enough and large enough LLM, like Qwen-2.5-72B(base), if a decent prompt and decent samples can be selected to activate the LLM's self-judge capacity, your work may also be adopted on non-instruction model. But, still, "Beyond" is still a bad word here.
> > 2. consider trying HelloBench: Evaluating Long Text Generation Capabilities of Large Language Models or selecting long-form generation tasks from WildBench / AlpacaEval like what they (HelloBench) have done. What is the real downstream needs from the users are the only important thing.
> >
> > I will look forward to a better version of the paper if accepted.

---

> > > ### Author Response · Authors · 2024-11-30
> > > **Thank you for your thoughtful feedback (6)**
> > >
> > > Dear reviewer YJ28,
> > >
> > > Thank you for your thoughtful feedback. We sincerely appreciate the time and effort you have dedicated to evaluating our work. Below, we provide short responses to your comments and suggestions.
> > >
> > > > 3. … a more detailed statistic analysis between the real downstream [i.e. from the true user, like what have been done in Chatbot Arena / WildBench] and the provided aspects should be deeply studied…
> > >
> > > Thank you for your thoughtful comment. We agree with you as you suggested that this will be an exciting (and possibly important?) future direction, as we believe aligning fine-grained metric properties is an exciting alternative alignment direction as our work follows. Below we provide the metrics chosen for AlpacaEval2 and WildBench-V2 for our reference:
> > >
> > > - AlpacaEval2: ```{'Accuracy': 5, 'Clarity': 4.8, 'Coherence': 5, 'Completeness': 5, 'Conciseness': 4.8, 'Engagement': 5, 'Relevance': 5}```
> > > - WildBench-V2: ```{'Accuracy': 5, 'Coherence': 4.8, 'Completeness': 5, 'Creativity': 5, 'Engagement': 5, 'Informativeness': 5, 'Naturalness': 5, 'Readability': 4.8}```
> > >
> > > While the model generally prioritizes ```Completeness, Relevance``` for AlpacaEval2, it notably prioritizes ```Creativity, Informativeness``` for  WildBench-V2, which are reasonable. While in this work we collected metrics widely used in prior studies (Section 3, Step 1), we also want to note that these metrics are flexibly customizable and generalizable (Generalizability Section). We strongly encourage task-specific customizations to further enhance the applicability and effectiveness of our approach (Generalizability Section).
> > >
> > > > Two comments about the name of including ICL and considering trying HelloBench…
> > >
> > > Thank you for your thoughtful suggestions; we greatly appreciate them. We named the paper "Beyond ICL" as it reflects our investigation into the limitations of in-context learning (ICL) in recovering the desired distribution for LLMs, though we acknowledge that this exploration could be more thorough. Nonetheless, we take your naming suggestion seriously and are actively considering it. Since the rebuttal period is closing soon, we are unable to give you a concrete response but we are committed that we will revise these points carefully in our revised manuscript. We sincerely thank you for your time and insightful feedback, and we hope our revised version will meet your expectations.

---

### Official Review · Reviewer_ku1W · 2024-11-04

**Soundness:** 3
**Presentation:** 3
**Contribution:** 2
**Rating:** 6
**Confidence:** 3

**Summary:**

This paper explores the ICL capabilities of LLMs, pointing out that relying solely on ICL is insufficient for effectively completing long-form generation tasks, and provides experimental validation and theoretical analysis to support this claim. Based on these findings, the authors propose a hypothesis: optimizing multiple text property tasks can approximate the overall goal of optimizing long text generation tasks. To this end, the authors design the LongGuide to generate two types of guidelines to enhance the performance of LLMs, and validate its effectiveness through a large amount of experiments.

**Strengths:**

1. The proposed method is grounded in a well-established conclusion supported by both experimental and theoretical evidence, providing a solid basis for its formulation.
2. The effectiveness of LongGuide has been confirmed through a number of experiments, demonstrating its significant performance improvement in multiple long-text generation tasks.
3. The article provides a solid theoretical analysis explaining why LongGuide can better achieve task objectives.

**Weaknesses:**

1. The LLM's selection of metrics is based on the distribution of its pre-training data, which may lead it to favor common or general metrics. Introducing human verification on top of this could be more effective.
2. I believe the novelty of this method is limited, as there are already some automated prompt designs for long-form generation [https://arxiv.org/html/2406.14449v1, https://arxiv.org/abs/2211.01910]. In certain cases, the LongGuide method can only choose not to use any guidelines in the final step.

**Questions:**

1. When calculating the JS.Avg metric, the authors used ChatGPT to score two responses, but the paper does not provide a specific display of the prompt used.
2.  In the final step of the method, there are only four choices—use MG, use OCG, use both, or use neither. Can a more sophisticated strategy be designed to leverage the advantages of both? For example, considering the requirements of MG and OCG in different stages rather than simultaneously.

---

> ### Author Response · Authors · 2024-11-20
> **Response to reviewer**
>
> Dear reviewer ku1W,
>
> We deeply thank you for your time and efforts in providing constructive reviews for our paper. We would like to address your concerns below and our updated changes in the paper are in blue.
>
> > The LLM's selection of metrics is based on the distribution of its pre-training data...
>
> Thank you for your suggestion. Appendix D.11 shows the metrics selected for each task by each model. After reviewing the metrics chosen by models like Mistral and ChatGPT, we find no clear bias in their selection process.
>
> Both models consistently choose key metrics like “Accuracy,” “Clarity,” “Relevance,” and “Understandability,” which are important for many language tasks. They also adjust their metric choices based on the tasks. For example, specific tasks like CNN and XL-Sum include additional metrics such as “Engagement” and “Semantic Coverage.” This suggests that the models select metrics reasonably, based on the needs of the task, rather than showing a preference for certain metrics. Overall, the variety and suitability of the selected metrics show that the process is fair and appropriate for the tasks.
>
> We have supplemented the discussion in Appendix D.11 where the selected metrics are presented.
>
> > I believe the novelty of this method is limited...
>
> Thank you for your perspective and for referencing APEER and APE. Both APEER and APE are automatic prompt generation methods: APEER uses a feedback-and-refine mechanism for prompt optimization and APE selects prompts based on validation performance.
>
> Our method fundamentally differs from APEER and APE as well as all prompt optimization (PO) methods in twofold:
> - Rather than focusing on refining, paraphrasing, or evolving prompts like PO methods, we generate task-specific guidelines to improve LLM alignment and performance in long-form generation tasks;
> - Our approach prioritizes task property and format distribution alignment over solely optimizing model performance like PO studies.
>
> As shown in Table 2 and Appendix C.2, LongGuide consistently outperforms advanced prompt optimization (PO) methods in long-form generation tasks. Current PO algorithms, even the most advanced ones, struggle to outperform LongGuide in certain long-form generation tasks because they typically rely on sampling new prompts through search, evolution, or paraphrasing methods, which rarely produce comprehensive guidelines like those generated by LongGuide. LongGuide has its own unique advantage.
>
> Our method has also been confirm novel and constructive by other reviewers (Reviewer V713 said our work is “novel” and 3/5 reviewers rated our contribution 3). We thank you for your feedback and we hope you appreciate the novelty of our work.
>
> > When calculating the JS.Avg metric, the authors used ChatGPT to score two responses, but the paper does not provide a specific display of the prompt used.
>
> Thank you for your comment. We have provided the ChatGPT property scorer prompt in Appendix F.2.
>
> > ...there are only four choices—use MG, use OCG, use both, or use neither. Can a more sophisticated strategy be designed to leverage the advantages of both?...
>
> Thank you for your feedback. We would like to clarify that in the inference stage, our approach employs LongGuide directly, which does not involve multiple stages.
>
> We extended our experiments to 2 baselines using 2 stages where we used:
>
> Baseline 1: MG to OCG
> - Stage 1: Instruction + Input + MG -> Output 1 (as usual MG baseline)
> - Stage 2: “Refine the following output from the task:\n” + Input + Output 1 + OCG -> Output 2
>
> Baseline 2: OCG to MG
> - Stage 1: Instruction + Input + OCG -> Output 1 (as usual OCG baseline)
> - Stage 2: “Refine the following output from the task:\n” + Input + Output 1 + MG -> Output 2
>
> The results are provided below with ChatGPT:
>
> | #shot | CNN (3.0.0) | SWiPE | Comm.-Chall. |
> | -------- | ------- | ------- | ------- |
> | Zero-shot | 20.12 / 7.44 | 45.09 / 7.28 | 24.21 / 6.53 |
> | Zero-shot + LongGuide | **22.19 / 7.67** | **45.09 / 7.28** | **34.41 / 7.23** |
> | Zero-shot + MG to OCG | 16.74 / 6.23 | 30.22 / 5.76 | 15.92 / 4.92 |
> | Zero-shot + OCG to MG | 9.62 / 4.18 | 20.34 / 4.82 | 8.86 / 3.97 |
>
> We observe that 2-stage baselines significantly degrade model performance, as the final generated answers deviate substantially from the ground truth. We attribute this to the model's inherent bias amplified by self-refining (https://aclanthology.org/2024.acl-long.826/).
>
> We agree that exploring more sophisticated strategies to leverage the complementary strengths of MG and OCG could be a promising direction for future research. We will incorporate the discussion of this potential extension into the Generalization section of our paper.
>
> ## In Summary
>
> We thank you for your time and constructive feedback. We hope our responses can sufficiently address your concern and improve your ratings. Thank you for your consideration.

---

> ### Author Response · Authors · 2024-11-23
> **Request for your review**
>
> Dear Reviewer ku1W,
>
> As the author-reviewer discussion period is nearing its conclusion, we kindly request your consideration of our responses to your concerns.
>
> We deeply thank you for your time and careful review. Many of the comments in your review we have actually addressed in the revised paper, as hopefully you can agree as mentioned above. We thank you for taking the concern to provide very detailed critique of the paper, and trust that you find our clarifications appropriate and worthy of a higher score.
>
> Thank you for your attention and consideration.
>
> Best regards, The Authors

---

> ### Comment · Reviewer_ku1W · 2024-11-28
> **Response to Rebuttal**
>
> Thanks for the response, I have increased my scores accordingly.

---

> > ### Author Response · Authors · 2024-11-30
> > **Thank you for reviewing our paper!**
> >
> > Thank you, reviewer ku1W, for your feedback!

---

### Official Review · Reviewer_V713 · 2024-11-05

**Soundness:** 2
**Presentation:** 3
**Contribution:** 3
**Rating:** 5
**Confidence:** 4

**Summary:**

In-context learning for generation tasks requires the model to capture some attributes of the desired output texts. The paper introduces a method, LongGuide, to more effectively use in-context examples for generation tasks by first using a small set of examples to develop guidelines about the desired output format, then providing these guidelines (generally in addition to the ICL examples) during inference. The guidelines are divided into two sets: metric-based guidelines, developed by using ChatGPT evaluation along axes of generation quality and selecting axes that human answers perform highly along; and statistics-based guidelines, generated from properties of the human answers (e.g. average length). LongGuide provides additional gains on top of ICL and prompt optimization.

**Strengths:**

S1. The idea of providing explicit task guidelines is well-motivated and clearly effective; I like the breakdown into metric-oriented and output-text-statistics oriented guidelines, which to the best of my knowledge is novel.

S2. The method is more effective on stronger models (likely because these models are better at instruction following). Surprisingly, it's also somewhat effective on weaker, non-instruction-tuned models (line 1036).

S3. The authors are thorough in their analysis, specification of hyperparameters, and description of the setting. I also appreciate the specification of annotator wages.

**Weaknesses:**

W1. I feel that the formalization in Section 2 is not rigorous and frankly distracting from the goals of the paper. While I can see some benefit to stating Remark 2.1 given that some of the literature makes different assumptions, I feel the assumptions provided to start the proof are not well-formed and the property as a whole does not feel terribly useful. Remark 2.2 seems wholly unnecessary, as it seems to essentially claim that you will consider your method better than the baseline if it outperforms the baseline on the metrics. This does not require a proof.

W2. While the attributes are computed using up to 50 train set examples, the model is not evaluated on using 50-shot ICL, which feels like a natural baseline to consider. Recent works on long-context ICL have demonstrated improved performance with many demonstrations.

W3. The use of ROUGE and BLEU as the final downstream metrics seems ill-advised. These are both very simple ngram metrics, without the expressiveness of other metrics; the fact that they show near-identical trends (line 368) is unsurprising because they measure very similar things.

**Questions:**

Q1: Can you elaborate on the aims of the formalization / theoretical claims in section 2? I am not clear on the reasoning for this section.

Q2: The point about the guidelines not being useful for tasks the models are trained on is an interesting claim (line 556 onwards). Could you verify this with a model with open training data?

Q3: In regards to the title: are these texts really "long-form"? Certainly they are generation rather than classification, but the outputs for most of these tasks are quite short.

Q4. The evaluation in 4.1 establishes the goal as minimal Jensen-Shannon divergence between the score distributions of gold summaries and model answers. Is this a good goal? Is it possible that the model answers are better on some axes than human answers, and this fails to account for this setting?

Minor presentation notes:
* the jump to notation in lines 36-45 was abrupt and it's not clear at this point in the paper why establishing this notation is useful; it might be better to introduce this in section 2.
* the use of "metrics" in line 101 reads a bit strange; I would refer to these as properties, which you evaluate using ChatGPT.
* showing the % gain is not super helpful in table 2, and it clutters the table.
* lines 478-485 about the ablations performed would have been useful to know earlier, since earlier tables reference these settings.
* formatting typo in the citation of Krippendorff's alpha (line 431)
* unclear what "the mode of 17 response tokens" means in line 123
* typo in line 1015: "second not the sam"
* in Figure 17, skipping step 2 seems to improve the performance, not worsen it?
* the formatting of links to appendix figures is a bit non-standard

---

> ### Author Response · Authors · 2024-11-20
> **Response to reviewer (1)**
>
> Dear reviewer V713,
>
> We deeply thank you for your time and efforts in providing reviews for our paper. We would like to address your concerns below and our updated changes in the paper are in blue.
>
> > W1. I feel that the formalization in Section 2 is not rigorous and frankly distracting from the goals of the paper… Remark 2.2 seems wholly unnecessary… Can you elaborate on the aims of the formalization / theoretical claims in section 2?…
>
> Thank you for your feedback. While this contradicts the rest of reviewers (YJ28 commented “I like the theoretical derivations given in Section 2.1” and ku1W commented “the article provides a solid theoretical analysis explaining why LongGuide can better achieve task objectives.”), we appreciate your comment and allow us to clarify your questions.
>
> Our goal of Subsection 2.1 is twofold:
> - It shows that ICL, the most common prompting-based alignment method, can’t help LLM to recover the desired alignment in the limit if initially the model fails to capture the task language distribution.
> - It provides the formalizations for our Hypothesis 2.1 and the (exact) definition of task property, which serves as the basis for our method.
>
> It is important to note that the purpose of this subsection is not to present a “rigorous” theory, but rather to provide an intuition that motivates our approach. We believe that a “solid” theoretical discussion should deserve the whole paper discussing details such as addressing epsilon distribution recovery. However, this is not our main focus in this work.
>
> We value your feedback and that of other reviewers, and we have worked hard to balance differing perspectives. Specifically, we have made the following changes in response to your suggestions:
> - We change “Theoretical Derivations” to “Theoretical Intuitions”.
> - We shorten the definition of task property significantly for more conciseness.
> - We remove the Remark 2.2 as you suggested. We describe it in L180-181 and put the old Remark 2.2 into Appendix A.
>
> We hope these changes address your concerns and demonstrate our efforts to balance reviewers.
>
> > W2. While the attributes are computed using up to 50 train set examples, the model is not evaluated on using 50-shot ICL, which feels like a natural baseline to consider.
>
> Thank you for your comment. While this may seem “natural”, it is important to note that for long-form generation, prompt optimization (PO) studies typically do not follow this method, please see APO (Pryzant et al., 2023) and adv-ICL (Do et al., 2024). We believe the reason is twofold: (1) Few-shot prompting with an excessive number of examples, such as 50 shots is unnatural in practice; (2) for long-form generation tasks, such as CNN, on average the #tokens for 1 shot is 798.29, thus 50 shots is 40K which exceeds the window size of most current commonly used LLMs such as Mistral-7B-it-v0.2 (limit 4096), Llama 3 (limit 8K) and gpt-3.5-turbo-1106 (limit 16K).
>
> Nevertheless, we still supplement the results for CNN (3.0.0), SWiPE, and Comm.-Chall. below where we use 10 shots for CNN, 40 shots for SWiPE, and Comm.-Chall **up to the limit of gpt-3.5-turbo-1106** evaluated by ROUGE-L / GPT-4o-Judge scores:
>
> | #shot | CNN (3.0.0) | SWiPE | Comm.-Chall. |
> | -------- | ------- | ------- | ------- |
> | 3-5 shots | 14.51 / 4.38 | 33.72 / 5.07 | 22.08 / 4.19 |
> | 3-5 shots + LongGuide | **18.17 / 4.42** | **37.60 / 5.25** | **38.21 / 7.21** |
> | 10-50 shots  | 20.55 / 6.67 | 44.04 / 6.07 | 28.18 / 4.85 |
> | 10-50 shots + LongGuide | **21.69 / 6.82** | **46.17 / 6.67** | **42.55 / 7.72** |
>
> We observe that while supplementing more shots to ChatGPT improves the model’s performance, LongGuide further boosts the ICL performance significantly for all three benchmarks.
>
> We have supplemented these results in Appendix D.1 and added one description in Section 4 describing that we also compare our method with many-shot prompting in L297.

---

> > ### Author Response · Authors · 2024-11-20
> > **Response to reviewer (2)**
> >
> > > W3. The use of ROUGE and BLEU as the final downstream metrics seems ill-advised.
> >
> > Thank you for your suggestion. We have added GPT-4o-Judge scores (Section 4) evaluating how aligned the generated answer is with the reference answer and its quality on criteria:
> >
> > - Format consistency: ensuring the generated response matches the length, structure, and layout of the reference.
> > - Content completeness: evaluating whether all key points present in the reference are included in the assistant's answer.
> > - Factuality: checking for factual correctness of the assistant's answer.
> > - Style adherence: ensuring that the tone, style, and level of detail of the assistant's answer match the reference.
> > - Assistant's answer quality: assessing how well the response satisfies the user's requirements.
> >
> > Each criterion is scored on a scale of 10, and the final GPT-4o-Judge score is the average of them. We have included the evaluation scores in Table 2 and Figure 2. We summarize the results below:
> >
> > | Method | Format | Content | Factuality | Style | Quality |
> > | -------- | ------- | ------- | ------- | ------- | ------- |
> > | Baseline | 4.18 | 4.83 | 6.64 | 4.36 | 4.75 |
> > | + APO  | 4.73 | 5.91 | 7.26 | 4.91| 5.39 |
> > | + LongGuide | **5.72** | **6.01** | **8.25** | **5.78** | **6.04** |
> >
> > Among five GPT-4o-Judge criteria in Figure 2, LongGuide notably improves Format, Style, and Factuality, confirming its effectiveness in aligning model generation with ground-truth distributions. In addition, the significant gains in Quality criterion, together with the ROUGE-L scores from Table 2 further demonstrate that LongGuide also significantly enhances the generation quality.
> >
> > Our evaluation prompting template is heavily inspired by (https://openreview.net/forum?id=uccHPGDlao).
> >
> > > The point about the guidelines not being useful for tasks the models are trained on is an interesting claim (line 556 onwards). Could you verify this with a model with open training data?
> >
> > Thank you for your feedback. We would like to clarify that this is a hypothesis we proposed, as stated in the manuscript: “LongGuide may not be useful for the tasks the models are trained on…”. Our hypothesis comes from our observation that “while we see notable enhancements on the CommonGen-Challenge dataset (Lin et al., 2020), it’s intriguing that we don’t observe any improvements on the WebNLG (Gardent et al., 2017) and E2E NLG (Puzikov & Gurevych, 2018) datasets. Given the popularity of these datasets, we suspect the models we tested may have been previously trained on them.” written in L556-562.
> >
> > Testing this hypothesis directly is challenging due to the opaque nature of training data for most large language models. Even for models with open training data, identifying specific overlaps or pretraining exposure remains complex.
> >
> > That said, we acknowledge that this is not a central claim of our work but rather an observation to guide future research. We appreciate your suggestion and agree that further studies could delve deeper into verifying this hypothesis.
> >
> > > In regards to the title: are these texts really "long-form"?
> >
> > Thank you for your question. We follow the ELI5 definition (Fan et al., 2019) of long-form generation as generating sentence- or paragraph-length answers (L053). All our tasks fall within this scope, requiring sentence- or paragraph-level answers. This distinguishes long-form generation from factoid question answering involving single-word answers, and multiple-choice question answering.
> >
> > > Q4. The evaluation in 4.1 establishes the goal as minimal Jensen-Shannon divergence between the score distributions of gold summaries and model answers. Is this a good goal? Is it possible that the model answers are better on some axes than human answers, and this fails to account for this setting?
> >
> > Thank you for your question. The primary goal of our work and the LongGuide method is to improve the alignment between the LLM generation distribution and ground-truth distribution (L086-087). For that purpose, minimizing the Jensen-Shannon divergence between the score distributions of gold summaries and model answers is a suitable objective.
> >
> > It is possible that the model answers are better on some axes than human answers. However, this is not our paper’s goal. Our work’s goal is to align LLM responses with human responses. Note that our goal aligns with most of the current alignment techniques where we all try to optimize towards human answers/preferences.
> >
> > > Minor presentation notes.
> >
> > Thank you for pointing out the writing advice. We have revised our manuscript accordingly. There is one question “in Figure 17, skipping step 2 seems to improve the performance, not worsen it?”. Skipping step 2 improves model performance from 17.24 to 21.62.

---

> > > ### Author Response · Authors · 2024-11-20
> > > **Response to reviewer (3)**
> > >
> > > ## In summary
> > >
> > > We thank you for taking the concern to provide very detailed critique of the paper. Many of the comments in your review we have actually addressed in the rebuttal and the updated manuscript, as hopefully you can agree as mentioned above. We trust that you find our paper, updates, and clarifications appropriate and worthy of a higher score.

---

> > > > ### Comment · Reviewer_V713 · 2024-11-21
> > > >
> > > > **On the formalization in 2.1:**
> > > > > While this contradicts the rest of reviewers
> > > >
> > > > I do think `ku1W` and I disagree on this, but I'd like to note this is a somewhat willful misquoting of `YJ28`, who wrote “I like the theoretical derivations given in Section 2.1. *But it is not solid. I don't buy in that 4.1 is a strong proof of Hypothesis 2.1.*" I share this concern about Hypothesis 2.1.
> > > >
> > > > > It is important to note that the purpose of this subsection is not to present a “rigorous” theory, but rather to provide an intuition that motivates our approach. We believe that a “solid” theoretical discussion should deserve the whole paper discussing details such as addressing epsilon distribution recovery. However, this is not our main focus in this work.
> > > >
> > > > While I understand this is not a theory paper (and I agree that "intuitions" is more appropriate framing than "derivations," I still feel the revised section 2 is not solid enough for acceptance. If you are using the form and verbiage of formal mathematics for your claims, you must also adopt the same level of rigor. I think the current use of notation does more to obstruct than aid the claims in the paper.
> > > >
> > > >
> > > > > We remove the Remark 2.2 as you suggested. We describe it in L180-181 and put the old Remark 2.2 into Appendix A.
> > > >
> > > > My issue with Remark 2.2 wasn't really that it was taking up main text space-- it's that it isn't a meaningful statement at all. The remark states that if the method meets/exceeds the baseline performance in all measured metrics and "text generation quality," it exceeds the baseline in "task performance." Most ML papers take "exceeds the baseline in measured metrics" as a proxy for task performance, so this is not necessary to state. And "text generation quality" is not defined (and, the phrasing implies, is something separate from the $f_i$s).
> > > >
> > > >
> > > > **On 50-shot ICL:**
> > > >  I'm glad to see that LongGuide outperforms ICL in a setting with more demonstrations in-context. I do disagree that this is an "unnatural" setting-- several recent works (e.g. [1](https://arxiv.org/abs/2405.00200), [2](https://arxiv.org/abs/2404.11018)) have demonstrated that ICL is effective up to several thousand examples in context, and many example selection methods are less effective in the higher-demonstration-regimes.
> > > >
> > > > **Downstream metrics**
> > > > Thank you for adding additional metrics here! This addresses my concern on this point.

---

> > > > > ### Author Response · Authors · 2024-11-22
> > > > > **Reponse to reviewer (4)**
> > > > >
> > > > > Dear reviewer V713,
> > > > >
> > > > > Thank you for engaging with us in this conversation and your constructive feedback. We would like to address your concerns below and our updated changes in the paper are in blue.
> > > > >
> > > > > > …this is a somewhat willful misquoting of YJ28…
> > > > >
> > > > > Thank you for sharing your thoughts. We believe the *main* reason why our hypothesis was “not solid” to ```YJ28``` was we did not cover the different weightings for different objectives as he elaborated on later and provided an illustrative example. Nevertheless, we thank you for your insight and agree with you and we made the below modifications, see our response below.
> > > > >
> > > > > > While I understand this is not a theory paper (and I agree that "intuitions" is more appropriate framing than "derivations," I still feel the revised section 2 is not solid enough for acceptance. If you are using the form and verbiage of formal mathematics for your claims, you must also adopt the same level of rigor. I think the current use of notation does more to obstruct than aid the claims in the paper.
> > > > >
> > > > > Thank you for your constructive feedback. We agree with the reviewer that this section might make readers obstructed. As a result, we have moved Subsection 2.1 into Appendix A, and we have added a short paragraph **Theoretical intuitions** in Section 2.
> > > > >
> > > > > > My issue with Remark 2.2 wasn't really that it was taking up main text space-- it's that it isn't a meaningful statement at all. The remark states that if the method meets/exceeds the baseline performance in all measured metrics and "text generation quality," it exceeds the baseline in "task performance." Most ML papers take "exceeds the baseline in measured metrics" as a proxy for task performance, so this is not necessary to state.
> > > > >
> > > > > Thank you for sharing your point in detail. We want to note that “measured metrics” in our case are not proxies for task performance as conventional ML papers. They are **text properties** as we defined in Section 2 (now Appendix A) such as “Fluency, Context Coverage, Informativeness, etc”. For example, not always optimizing “Context Coverage” in all summarization circumstances leads to better summarization performance.
> > > > >
> > > > > We believe the value of Remark 2.2 (now Remark A.1) is not trivial. Without Hypothesis 2.1 (now Hypothesis A.1), Remark 2.2 (now Remark A.1) can’t be proven. This is because there is no guarantee that text properties are the proxies for task performance. Remark 2.2 (now Remark A.1) emphasizes that text properties must be “well-chosen” following Hypothesis 2.1 and we need to optimize their levels in the task data. Only in that case, they all in once will be the proxy with task performance.
> > > > >
> > > > > Please feel free to share more of your thoughts. Thank you!
> > > > >
> > > > > > On 50-shot ICL, Downstream metrics
> > > > >
> > > > > We are glad that our responses addressed your two concerns. Thank you for your feedback.
> > > > >
> > > > > We appreciate your feedback and the references regarding many-shot prompting. As noted in our “Response to Reviewer (1),” prior PO studies did not incorporate such baseline, and we previously followed them. We already supplemented your suggested baseline in the paper, see L293 and Appendix D.1.
> > > > >
> > > > > ## In Summary
> > > > > In summary, we thank you for your time, effort, and constructive feedback. We hope you will recognize the empirical, observational, and constructive contributions presented in this work, which we believe benefit the field as a whole. With our modifications, we trust that you find our paper, updates, and clarifications appropriate and worthy of a higher score.

---

> > > > > > ### Comment · Reviewer_V713 · 2024-11-28
> > > > > >
> > > > > > > Without Hypothesis 2.1 (now Hypothesis A.1), Remark 2.2 (now Remark A.1) can’t be proven. This is because there is no guarantee that text properties are the proxies for task performance.
> > > > > >
> > > > > > There's no guarantee that any metric is a proxy for task performance!  You could write a similar formulation to Hypothesis 2.1 about just about any method for self-refinement or output reranking.
> > > > > >
> > > > > > > Remark 2.2 (now Remark A.1) emphasizes that text properties must be “well-chosen” following Hypothesis 2.1
> > > > > >
> > > > > > I also think this is an issue in the proof, for what it's worth -- in the proof of Remark 2.2, you assume that the set of text properties you are measuring are _well chosen_, but Hypothesis 2.1 only claims that such a set _exists_.
> > > > > > (Also I believe in the current version the original Remark 2.2 is Remark A.2. I refer to it here as 2.2 for clarity.)
> > > > > >
> > > > > >
> > > > > > Overall, I still don't think your mathematical formulation is meaningful, because the remarks leverage text descriptions of concepts that do not have a mathematically rigorous definition, like "text quality" in Remark 2.2. At best, it adds nothing to your empirical results; at worst, I worry it could be misleading to the reader.

---

> > > > > > > ### Author Response · Authors · 2024-11-29
> > > > > > > **Thank you for your thoughful comments**
> > > > > > >
> > > > > > > Dear reviewer V713,
> > > > > > >
> > > > > > > Thank you for your getting back to us and for your thoughtful comments. We understand your concern about concepts that lack mathematical definitions. We would like to address your concerns below.
> > > > > > >
> > > > > > > > I also think this is an issue in the proof, for what it's worth -- in the proof of Remark 2.2, you assume that the set of text properties you are measuring are well chosen, but Hypothesis 2.1 only claims that such a set exists.
> > > > > > >
> > > > > > > Thank you for your feedback. We want to clarify that Remark A.2 builds directly on Hypothesis A.1. Under the assumption that Hypothesis A.1 holds, we obtain two things: (1) the existence of $\{f_1,\dots,f_r\}$ and (2) optimizing these functions during generation ensures a lower overall loss.
> > > > > > >
> > > > > > > > Overall, I still don't think your mathematical formulation is meaningful, because the remarks leverage text descriptions of concepts that do not have a mathematically rigorous definition, like "text quality" in Remark 2.2. At best, it adds nothing to your empirical results; at worst, I worry it could be misleading to the reader.
> > > > > > >
> > > > > > > Thank you for your thoughtful comment. We appreciate your concern regarding the clarity of the “text generation quality” concept. We propose this concept primarily covering fundamental linguistic properties of generated responses (Grammatical correctness, Readability, etc) and importantly, also the task-specific alignment.
> > > > > > >
> > > > > > > This concept itself is subjective: each person may have their own “text quality” like we can write the same idea very differently. Nevertheless, this concept is necessary. Specifically, the set of "well-chosen" text properties may not cover all subjective aspects, and lower task loss alone may not always indicate a better solution. For instance, a generated answer that diverges from the ground truth can still be valid.
> > > > > > >
> > > > > > > To address your concern, we propose a simple modification. We define the (subjective) text generation property as a function $f_P: (\mathcal{X}, \mathcal{Y}) \to \mathbb{R}$. The modifications to the current Appendix A are also simple: we briefly define this function in L1049-1051, include it in Remark A.2 L1053, and briefly talk about it in L1112-1114. We believe now the theoretical intuition should be more solid, as hopefully you agree with us, given that all concepts are defined by functions, even though we did not specify how they are computed. We will further clarify this “text generation property” concept better in the Appendix A.
> > > > > > >
> > > > > > > Overall, we very much understand your concern and your suggestion. As we have noted, these formulations are intended to provide a theoretical intuition, and we have placed them in the Appendix to support audiences who find them helpful. We hope you will recognize the empirical, observational, and constructive contributions of our work
> > > > > > >
> > > > > > > We are very open to suggestions and discussions, and happy to take them. Feel free to share your feedback and thoughts.

---

> > > > > > > > ### Author Response · Authors · 2024-12-02
> > > > > > > > **Request for your review**
> > > > > > > >
> > > > > > > > Dear reviewer V713,
> > > > > > > >
> > > > > > > > As the ICLR discussion phase is closing soon, we respectfully request your consideration of our responses to your concerns. We have carefully addressed your concerns and made the noted changes that the theoretical intuition section is now in the Appendix.
> > > > > > > >
> > > > > > > > Thank you for your valuable time and feedback, and thank you for your attention.
> > > > > > > >
> > > > > > > > Best regards,
> > > > > > > > Authors

---

> > > > > > > > > ### Comment · Reviewer_V713 · 2024-12-02
> > > > > > > > >
> > > > > > > > > > We want to clarify that Remark A.2 builds directly on Hypothesis A.1. Under the assumption that Hypothesis A.1 holds, we obtain two things: (1) the existence of
> > > > > > > > >  and (2) optimizing these functions during generation ensures a lower overall loss.
> > > > > > > > >
> > > > > > > > > While I see why you introduced Hypothesis A.1 to use as a stepping stone to Remark A.2, the concern here stands. Even when you assume Hypothesis A.1 holds, you can only use the fact that a set of functions that satisfies these properties _exists_. The proof (in line 1112 of the current pdf) assumes that the set of the text properties you are currently optimizing _is the set that you proved exists_, but there's no guarantee of this. Hypothesis A.1 says that you believe this set of properties exists; section 4/LongGuide is based on the idea that if you can discover these properties, you can approximately optimize the objective; your results show that you can discover a set of properties that serve as a better approximation of true quality than taking the maximum likelihood conditioned on a set of in-context examples. I don't think you need Remark A.2 at all for the claims you are making, and I don't think there is a way to prove it in the framework you've established.
> > > > > > > > >
> > > > > > > > >
> > > > > > > > > > We believe now the theoretical intuition should be more solid, as hopefully you agree with us, given that all concepts are defined by functions, even though we did not specify how they are computed.
> > > > > > > > >
> > > > > > > > > Writing the text generation quality as a function (without defining how it is computed) is not any more rigorous than writing it in natural language, it just looks more math-y. I think this is the crux of my critique of appendix A: it _looks_ like math, but it contains remarks that make claims involving subjective criteria and proofs that are not sound. While I think the paper could be fine without a theoretical intuition section at all, I think Remark/Definition/Hypothesis A.1 are all fine; it's really Remark A.2 that I take serious issue with.
> > > > > > > > >
> > > > > > > > >
> > > > > > > > > Finally, I want to thank the authors for engaging so consistently throughout the rebuttal period, especially as we've continued to disagree. I was quite conflicted on my final score here; while I still do not agree with the authors on the formalization and find parts of it remain fundamentally flawed, I do recognize that this is not the main point of the paper. After consideration, I chose to raise my score 3->5. If the paper is accepted, I urge the authors to carefully rethink the mathematical sections of the paper for the final version.

---

> > > > > > > > > > ### Author Response · Authors · 2024-12-03
> > > > > > > > > > **Thank you for your thoughful comments**
> > > > > > > > > >
> > > > > > > > > > Dear reviewer V713,
> > > > > > > > > >
> > > > > > > > > > Thank you for getting back to us with your very thoughtful feedback; we sincerely appreciate your explanations and feedback and the time and effort you have dedicated to evaluating our work. We would like to address your concerns below.
> > > > > > > > > >
> > > > > > > > > > > While I see why you introduced Hypothesis A.1 to use as a stepping stone to Remark A.2, the concern here stands...
> > > > > > > > > >
> > > > > > > > > > > Writing the text generation quality as a function (without defining how it is computed) is not any more rigorous than writing it in natural language, it just looks more math-y. I think this is the crux of my critique of appendix A…
> > > > > > > > > >
> > > > > > > > > > Thank you for your thoughtful comments; we understand your concerns, and you are right. Our initial motivation for Hypothesis A.1 was to establish two key points: first, that the set of text properties exists, and second, optimizing this set during generation leads to improved alignment. However, upon reflection, we agree with your observation regarding the gap between Hypothesis A.1 and Remark 2.2.
> > > > > > > > > >
> > > > > > > > > > *To address your two concerns, we will remove Remark A.2 and its proof for two reasons all pointed out by you: the identified gap and the lack of a rigorous definition of text quality.* We are glad to receive your feedback that Remark/Definition/Hypothesis A.1 are fine, so we will keep them in the revised manuscript.
> > > > > > > > > >
> > > > > > > > > > If possible, we would greatly appreciate your confirmation that the proposed modifications fully address your concerns and bring us into an agreement.
> > > > > > > > > >
> > > > > > > > > > Thank you once again for your consistently constructive feedback, engagement, and attention. It has been a pleasure working with you on this rebuttal. Thank you for reviewing our paper!

---

### Official Review · Reviewer_9VDd · 2024-11-05

**Soundness:** 3
**Presentation:** 4
**Contribution:** 3
**Rating:** 6
**Confidence:** 3

**Summary:**

ICL usually use demonstration examples to improve the LLM's performance.  In this paper, the authors propose LongGuide (§3), a guideline- learning algorithm that efficiently1 generates two types of guidelines concurrently from limited task training data as supplementary instructions to enhance LLMs: metrics guidelines and output constraint guidelines.

**Strengths:**

- the paper demonstrate the complementary values of guidelines; and propose two kinds of guidelines: metrics guidelines and output constraint guidelines.
- the paper proposed a full algorithm to learn guidelines from task training dataset. and show improved performance on multiple datasets of different tasks
- The paper did a through ablation study and investigation/evaluations to study the impact of guidelines.

**Weaknesses:**

- There exist many advanced automatic prompt optimization algorithm based on a training dataset, the authors only include APO, and I think they should include some more recent methods as baselines. So that we can have a better understanding the usefulness of this method.
- the automatic metrics are out-of-dated e.g. using Rouge scores for summarization. The authors can use LLM as judges to show the improvement.
- human evaluate datasets in Figure 5 is small where only 50 examples.

**Questions:**

- Please add some more recent algorithms in automatic prompt improvement as baselines.
- Please include some LLM-based evaluation method to compare the long-form generation instead of rouge and bleu scores.

---

> ### Author Response · Authors · 2024-11-20
> **Response to reviewer**
>
> Dear reviewer 9VDd,
>
> We deeply thank you for your time and efforts in providing constructive reviews for our paper. We would like to address your concerns below and our updated changes in the paper are in blue.
>
> > There exist many advanced automatic prompt optimization algorithm based on a training dataset, the authors only include APO, and I think they should include some more recent methods as baselines. Please add some more recent algorithms in automatic prompt improvement as baselines.
>
> Thank you for your suggestion. As discussed in line 302, we have compared LongGuide with **adv-ICL** (Do et al., 2024), a strong prompt optimization (PO) method at the time, in Appendix C.3 across three representative datasets: CNN, IWSLT17, and CommGen. We have also incorporated **EvolPrompt** (Guo et al., 2024) in our experimental analysis in Appendix C.3.
>
> We also acknowledge other recent PO algorithms such as **PromptAgent** (Wang et al., 2023) and **Promptbreeder** (Fernando et al., 2023). However, these PO methods are less applicable to long-form generation tasks due to the ambiguity of error feedback, compared to reasoning/MCQ tasks.
>
> Current PO algorithms, even the most advanced ones, struggle to outperform LongGuide in certain long-form generation tasks because they typically rely on sampling new prompts through search, evolution, or paraphrasing methods, which rarely produce comprehensive guidelines like those generated by LongGuide. LongGuide has its own unique advantage. Additionally, LongGuide can be combined with PO algorithms to further enhance its guidelines, as noted in Appendix C.3.
>
> > Please include some LLM-based evaluation method to compare the long-form generation instead of rouge and bleu scores.
>
> Thank you for your suggestion. We have added GPT-4o-Judge scores (Section 4) evaluating how aligned the generated answer is with the reference answer and its quality on criteria:
>
> - Format consistency: ensuring the generated response matches the length, structure, and layout of the reference.
> - Content completeness: evaluating whether all key points present in the reference are included in the assistant's answer.
> - Factuality: checking for factual correctness of the assistant's answer.
> - Style adherence: ensuring that the tone, style, and level of detail of the assistant's answer match the reference.
> - Assistant's answer quality: assessing how well the response satisfies the user's requirements.
>
> Each criterion is scored on a scale of 10, and the final GPT-4o-Judge score is the average of them. We have included the evaluation scores in Table 2 and Figure 2. We summarize the results below:
>
> | Method | Format | Content | Factuality | Style | Quality |
> | -------- | ------- | ------- | ------- | ------- | ------- |
> | Baseline | 4.18 | 4.83 | 6.64 | 4.36 | 4.75 |
> | + APO  | 4.73 | 5.91 | 7.26 | 4.91| 5.39 |
> | + LongGuide | **5.72** | **6.01** | **8.25** | **5.78** | **6.04** |
>
> Among five GPT-4o-Judge criteria, LongGuide notably improves Format, Style, and Factuality, confirming its effectiveness in aligning model generation with ground-truth distributions. In addition, the significant gains in the Quality criterion, together with the ROUGE-L scores from Table 2 further demonstrate that LongGuide also significantly enhances the generation quality.
>
> Our evaluation prompting template is heavily inspired by (https://openreview.net/forum?id=uccHPGDlao).
>
> > human evaluate datasets in Figure 5 is small where only 50 examples.
>
> Thank you for your feedback. For each sample, each annotator was asked to rate 7 metrics in total (5 random MG metrics and 2 OCG metrics). Due to resource constraints, we were only able to hire them for 50 samples, resulting in 350 ratings per annotator.
>
> ## In summary
>
> We thank you for your time and constructive feedback. We hope our responses can sufficiently address your concern and improve your ratings. Thank you for your consideration.

---

> > ### Author Response · Authors · 2024-11-30
> > **Request for your review**
> >
> > Dear reviewer 9VDd,
> >
> > We hope this message finds you well. As the author-reviewer discussion period is nearing its conclusion, we kindly request your consideration of our responses to your concerns. We sincerely thank you for your thoughtful reviews and the time and effort you have dedicated to evaluating our work.
> >
> > Thank you for your attention.
> >
> > Best regards,
> > Authors

---

### Author Response · Authors · 2024-11-22
**Summarize major changes to all reviewers and invite them to review and discuss**

Dear the Reviewers,

We sincerely appreciate and thank you for the time and effort you have dedicated to providing detailed feedback on our paper. We would like to bring your attention to the major changes we have made in our paper:

- We have added GPT-4o-Judge as an LLM evaluation method to address the concerns from ```9VDd, V713, YJ28, yZpL```.
- We have shortened the theoretical intuition section in our paper into a paragraph in Section 2, and moved the full section into the Appendix to address the concerns from ```V713, YJ28```. We have retained the section's content in the Appendix to support reviewers who recognized its benefits.
- We have added new baselines to address all baseline-related concerns, including more recent PO baselines (```9VDd```), a many-shot prompting baseline (```V713```), and several-stage baselines (```ku1W```).
- We have added ```AlpacaEval2``` evaluation to further verify the effectiveness of our method on real-life LLM chat, following the suggestion of ```YJ28```.
- Prompting cost analyses comparing our method with PO algorithms have been also added, addressing the concern raised by ```yZpL```.

For their details, we invite you to review our responses, we have carefully responded to all the concerns raised by each of you, which are ready for your consideration.

Thank you once again for your invaluable feedback, dedication, and attention. We are looking forward to discussing with you.

Best regards,
The authors

---

### Author Response · Authors · 2024-12-03
**Summary of reviews, contributions, and changes**

Dear Reviewers and Chairs,

We sincerely thank all the reviewers again for their insightful and constructive reviews. We are grateful that they found our paper has good writing (```YJ28```) and recognized our method to be novel (```V713, yZpL```), well-motivated (```YJ28, ku1W```) and effective (```V713, yZpL, ku1W, 9VDd```) supported by comprehensive experiments (```9VDd, V713, yZpL, YJ28, ku1W```). And we are delighted that subsequent discussions have successfully addressed your major concerns (```ku1W, YJ28, yZpL```), and reviewers ```V713, ku1W, YJ28, yZpL``` have raised their scores.

Our key contributions compared to automatic prompt engineering (PE)/optimization (PO) algorithms are summarized as follows:

1. We show that ICL demonstrations alone fail to enable pre-trained LLMs to consistently maintain their language and format properties during generation, as the ICL demonstrations alone can’t fully align the LLM-induced distribution to the desired task distribution in the limit.

2. We then propose a novel alignment method (LongGuide) by automatically selecting and capturing important task-specific language and format properties and explicitly instructing LLMs to optimize them during generation. To the best of our knowledge, LongGuide is the first to explore enhancing the generation by optimizing task properties during this process.

3. LongGuide significantly outperforms baselines and PO algorithms in long-form generation tasks. It is also efficient (>= 3.5x cheaper than PO methods), generalizable, transferrable, and can be synergistically combined with PO/PE algorithms.

Following the insightful suggestions of the reviewers, we have made the following revisions:

1. We have added GPT-4o-Judge as an LLM evaluation method for our main experiments (Table 3, Figure 5, Table 6) to address the major concerns from ```9VDd, V713, YJ28, yZpL```.

2. We have added four new LLMs to our experiments in Section 2 verifying the limitations of the ICL method following the suggestion of ```YJ28```.

3. We have shortened the theoretical intuition section into two short paragraphs in Section 2, and moved the full section into the Appendix to address the concerns from ```V713, YJ28```. We will also remove Remark A.2 and its proof to fully address the concern from ```V713```. We have retained the section's core content to support reviewers who recognized its benefits.

4. We have added new baselines including more recent PO baselines (```9VDd```), a many-shot prompting baseline (```V713```), and several-stage baselines (```ku1W```).

5. We have added AlpacaEval2 evaluation in Section 5.3 to further verify the effectiveness of our method on real-life LLM chat, following the suggestion of ```YJ28```.

6. Prompting cost analyses comparing our method with PO algorithms have been also added, addressing the concern raised by ```yZpL```.

7. We have revised the discussions of Sections 5.2 and 5.4 to provide more insights. We have also revised some minor details suggested by reviewers and provided implementation details for the new experiments added.

Thank you all once again for your valuable feedback, dedication, engagement and attention; we greatly appreciate them.

Best Regards,

Authors

---

### Meta-Review · Area_Chair_HmFA · 2024-12-16

**Metareview:**

Here’s the revised text maintaining the original style while fixing grammar and improving clarity:

This paper introduces LongGuide, a novel guideline-learning algorithm designed to enhance in-context learning (ICL) for long-form generation tasks. By concurrently generating metric-based and constraint-based guidelines from limited training data, LongGuide supplements standard ICL examples to improve adherence to desired text properties such as format and length. Experimental results demonstrate its effectiveness across various generation tasks, outperforming prompt optimization techniques on models of varying sizes, as evidenced by improvements in both automatic metrics (e.g., BLEU, ROUGE-L) and human evaluations.

Strengths:

* The paper introduces a relatively novel approach, and the idea of providing explicit task guidelines is well-motivated.
* The paper shows improved performance on multiple datasets and long-text generation tasks. In particular, it outperforms prompt optimization SoTA algorithms.
* The article provides a theoretical analysis of LongGuide, suggesting why it can better achieve task objectives.

Weaknesses:

* Novelty: The proposed method lacks substantial originality, as automated prompt design techniques for long-form generation already exist in recent literature. Furthermore, the method sometimes defaults to not using any guidelines, which may limit its practical impact.
* Presentation: The theoretical formalization in Section 2 is not rigorous, with seemingly unnecessary remarks and weakly connected hypotheses that detract from the clarity and utility of the paper. The proofs provided do not substantively enhance understanding or support the claims made about the method.
* Evaluation: The evaluation relies heavily on outdated metrics like ROUGE and BLEU, which lack the expressiveness of modern alternatives and fail to align with human judgment. While BERTScore is included, it does not sufficiently address the limitations of n-gram-based metrics. Human evaluation is conducted on a small dataset of only 50 examples, limiting its reliability.
* Comprehensiveness: The scope of the experiments is narrow, focusing only on simpler datasets such as SAMSum, CNN, and SWiPE. More challenging and diverse benchmarks, along with a deeper exploration of performance/cost trade-offs compared to state-of-the-art prompt optimization methods, would strengthen the practical relevance of the results.

To the authors' credit, several of the issues mentioned above were addressed during the discussion period—for example, through numerous additional experiments and a reframing of the theoretical section—which led some reviewers to increase their ratings. However, despite these significant changes, the paper remains very much borderline, making me hesitant to recommend acceptance, especially since none of the five reviewers was willing to give the updated paper a clear endorsement. During the reviewer-AC discussion, most reviewers maintained that the novelty of the work is limited. Consequently, I recommend rejecting the paper.

**Additional Comments On Reviewer Discussion:**

During the discussions, the authors addressed most of the weaknesses listed in the meta-reviews, except the one on novetly. During the reviewer–AC discussion, several reviewers expressed satisfaction with some of these changes. However, despite reading the authors' rebuttal, four reviewers stated that they still feel this work does not provide enough novel ideas or meaningful insights, leading me to recommend rejecting this paper.

---

### Decision · Program_Chairs · 2025-01-22

Reject